# Twisted gauging and topological sectors in (2+1)d abelian lattice gauge theories

Bram Vancraeynest-De Cuiper,$^{\heartsuit}$ Clement Delcamp$^{\varheartsuit}$

$^{\heartsuit}$*Department of Physics and Astronomy, Ghent University, Krijgslaan 281, 9000 Gent, Belgium*
$^{\varheartsuit}$*Institut des Hautes Études Scientifiques, Bures-sur-Yvette, France*

bram.vancraeynestdecuiper@ugent.be, delcamp@ihes.fr

Given a two-dimensional quantum lattice model with an abelian gauge theory interpretation, we investigate a duality operation that amounts to gauging its invertible 1-form symmetry, followed by gauging the resulting 0-form symmetry in a twisted way via a choice of discrete torsion. Using tensor networks, we introduce explicit lattice realisations of the so-called condensation defects, which are obtained by gauging the 1-form symmetry along submanifolds of spacetime, and employ the same calculus to realise the duality operators. By leveraging these tensor network operators, we compute the non-trivial interplay between symmetry-twisted boundary conditions and charge sectors under the duality operation, enabling us to construct isometries relating the dual Hamiltonians. Whenever a lattice gauge theory is left invariant under the duality operation, we explore the possibility of promoting the self-duality to an internal symmetry. We argue that this results in a symmetry structure that encodes the 2-representations of a 2-group.

# Introduction

Given a quantum theory, a modern viewpoint identifies the existence of a collection of *topological defects* as the definition of a global *symmetry* [GKSW14]. Aside from producing notions of symmetry that go beyond the ordinary one—and its axiomatisation in terms of abstract *groups*—this viewpoint has led to a calculus of symmetries that takes full advantage of well-established methods from topological quantum field theory [FMT22]. In particular, this calculus greatly facilitates the *gauging* of finite symmetries and the study of the consequences of such an operation, both in the continuum [BT17, GK20, ABGE⁺21, KOZ22, DLWW23, CJ24] and in the discrete [LDOV21, MMT22, DT23, BBSNT24].

In this manuscript, we are interested in two-dimensional quantum lattice models that result from gauging an ordinary finite abelian 0-form symmetry. These are defined on a constrained Hilbert space of states satisfying magnetic *Gauß constraints*. As such, these can be interpreted as abelian *lattice gauge theories*. It is well established that theories obtained in this way can be coupled to topological lines labelled by the *Pontrjagin dual* of the initial abelian group, which define a 1-form symmetry. But, it was recently understood that in addition to these topological lines, such theories also host topological surfaces referred to as *condensation defects* [RSS22, LRS22], which have the peculiarity of being typically non-invertible as linear maps on the constrained Hilbert space [CCH⁺22, DT23, CSSZ24]. The resulting complete symmetry structure is axiomatised in terms of a higher mathematical structure encoding the so-called *2-representations* of the group [Del21, DT23, BBSNT22, BSNW22, BBFP22].

Given a (2+1)d abelian lattice gauge theory, we consider an operation that amounts to ungauging its original 0-form symmetry, before gauging it back in a twisted way via a choice of *discrete torsion*, which is classified by the third cohomology of the group. The (untwisted) gauging operation alone is intimately related to the celebrated *Kramers–Wannier* transformation [KW41, Weg71], which is known to be a *non-invertible* operation on a fixed Hilbert space. Indeed, assuming closed periodic boundary conditions, gauging projects both theories onto the *singlet* sectors of their respective symmetries. Therefore, turning the Kramers–Wannier transformation into a duality requires addressing the non-trivial interplay between closed boundary conditions and charge sectors. An isometry acting on the resulting total Hilbert space that relate the spectra of the dual models can then be found by promoting the choice of boundary condition to a dynamical degree of freedom [LTL⁺20, MMT22, LDV22, SSS24]. Calculating the mapping of topological sectors under the combination of untwisted and twisted gaugings described above is the main objective of this work.

One merit of the dualities investigated in this manuscript is the ability to define them in any spacetime dimension. More precisely, one can apply the same strategy to any quantum theory that results from gauging the finite invertible 0-form symmetry of another theory. For instance, it was demonstrated in ref. [LDOV21] that in (1+1)d the so-called *Kennedy–Tasaki* transformation was precisely of this type [KT92, Osh92]. Another attractive trait of these dualities is that they preserve the symmetry structure. This is especially relevant in (2+1)d where most gauging procedures are found to modify the symmetry structure. As a matter of fact, it was conjectured in ref. [DT23] that most non-trivial dualities in (2+1)d sharing this feature are of the type investigated in this manuscript, later corroborated in ref. [DY23]. Since the symmetry structure is preserved under the duality, one can ask for models that are left invariant under the transformation. Whenever this is the case, one can further ask to promote the resulting self-duality to an internal symmetry, which mathematically amounts to performing a *group extension* of the corresponding symmetry structure [TY98, ENO10, Dé24]. In (2+1)d, we argue that this results into the emergence of lattice *higher*

*gauge theories* [BH10, KT13, BCK+17, DT18, DT19, BD19, ZLW19] whose symmetry operators are organised into higher mathematical structures encoding the 2-representations of 2-groups.

Concretely, given a finite symmetry acting on a one-dimensional quantum lattice model, a framework was developed in ref. [LDOV21] in order to systematically gauge any of its subsymmetries, effectively realising the approach of ref. [FFRS09, BT17] in the discrete. This framework relies on a generalisation of the *anyonic chain* construction [FTL+06, GAT+13, BG17], which, in addition to clearly distinguishing an abstract symmetry from its explicit realisation in a physical system, makes *tube algebra* techniques [Ocn94, Ocn01] amenable to the computation of *topological sectors* [AFM20, LOST22, LDV22]. In this formulation, the action of a finite symmetry, which is encoded into a so-called *fusion category* [ENO10], requires in particular a choice of *module category* [EGNO16]. Two physical systems that only differ in such a choice of module category are related by some gauging procedure. Then, the type of duality considered in the present manuscript relates two physical systems that differ in a choice of module category that is equivalent to the category of complex vector spaces. A generalisation of this framework to (2+1)d was initiated in ref. [DT23]. In this higher-dimensional context, the action of a finite symmetry, which is now encoded into a so-called *fusion 2-category* [DR18, GJF19], requires in particular a choice of *module 2-category* [Dé21, Del21, Dé23, DY23]. The type of duality we are interested in then relates two abelian lattice gauge theories that only differ in a choice of module 2-category that is equivalent to the 2-category of 2-vector spaces. A general strategy to realise on the lattice the various symmetry operators evoked above was presented in ref. [DT23, IO23]. Following ref. [LDV22], the same strategy can be employed to compute the lattice operators that transmute the dual Hamiltonians onto one another. Here, we realise these various operators very explicitly in the form of *tensor network operators* using the calculus outlined in ref. [Del21], partially extending to higher dimensions the results of ref. [BAV09, SCPG10, cWB+14, WBV17, LFH+21]. Using these operators, we are then able to compute the mapping of topological sectors under the dualities.

**Organisation of the manuscript:** We begin by reviewing in sec. 2 the one-dimensional scenario. After introducing a tensor network operator implementing the duality that amounts to the gauging of the abelian invertible symmetry followed by the twisted gauging of the dual symmetry, we derive the interplay between symmetry-twisted boundary conditions and charge sectors. This section is concluded by exploring concrete examples. In sec. 3, we begin by introducing families of two-dimensional quantum lattice models with a lattice gauge theory interpretation. We then unpack the symmetry encoded into the fusion 2-category of 2-representations of the gauge group, realising in particular the condensation defects in the form of tensor networks. We then employ the same calculus to realise the duality operator, which we leverage to compute the permutation of topological sectors under the duality. Finally, we discuss lifting the self-duality to a genuine internal symmetry through the lens of a simple theory built from the so-called toric code and the double semion model.

**Acknowledgements:** The authors are grateful to Laurens Lootens for collaboration at an early stage of this work, and Frank Verstraete for inspiring discussions. This work has received funding from the Research Foundation Flanders (FWO) through doctoral fellowship No. 11O2423N awarded to BVDC.

# Twisted gauging of (1+1)d spin chains

*Given an arbitrary one-dimensional quantum lattice model with an abelian invertible symmetry, we review the non-trivial interplay between closed symmetry-twisted boundary conditions and charge sectors under dualities that amount to the (untwisted) gauging of the symmetry, followed by the twisted gauging of the dual symmetry. Given pairs of compatible boundary conditions and charge sectors, the unitary operators relating the spectra of dual Hamiltonians are explicitly constructed in the form of tensor network operators.*

## 2.1 Symmetric Hamiltonians and closed symmetry-twisted boundary conditions

Let $G$ be a finite *abelian* group $G$ and $\psi$ a normalised representative 2-cocycle of a cohomology class $[\psi]$ in $H^2(G, \mathrm{U}(1))$. We begin by constructing a family of Hamiltonians on a *closed* one-dimensional lattice that commutes with symmetry operators labelled by characters in the Pontrjagin dual $\widehat{G} = \mathrm{Hom}(G, \mathrm{U}(1))$ of $G$. Given a closed one-dimensional lattice, we denote the sets of vertices and oriented edges by $\mathsf{V}$ and $\mathsf{E}$, respectively. Notice that $|\mathsf{V}| = |\mathsf{E}|$. Choosing a total ordering of the vertices induces an orientation of the edges. For any edge $\mathsf{e} \in \mathsf{E}$ we write its source and target vertex as $\partial_-\mathsf{e}$ and $\partial_+\mathsf{e}$, respectively, so that we identify $\mathsf{e} \equiv (\partial_-\mathsf{e}\,\partial_+\mathsf{e})$. Throughout this section, we assume without loss of generality that all the edges share the same orientation. To every edge, we assign a copy of $\mathbb{C}[G] = \mathrm{Span}_{\mathbb{C}}\{|g\rangle \,|\, g \in G\}$, with $\langle g_1|g_2\rangle = \delta_{g_1,g_2}$, so that the microscopic Hilbert space of the model is given by $\bigotimes_{\mathsf{e}\in\mathsf{E}} \mathbb{C}[G]$. Given a function $\mathfrak{g} : \mathsf{E} \to G$, we notate via $|\mathfrak{g}\rangle = \bigotimes_{\mathsf{e}\in\mathsf{E}} |\mathfrak{g}(\mathsf{e})\rangle$ the function $\mathfrak{g}$ regarded as a basis element of the microscopic Hilbert space. Moreover, for every group element $x \in G$ and every vertex $\mathsf{v} \in \mathsf{V}$, we introduce a function $\mathfrak{x}_\mathsf{v} : \mathsf{V} \to G$ such that $\mathfrak{x}_\mathsf{v}(\mathsf{v}_1) = \mathbb{1}$, whenever $\mathsf{v}_1 \neq \mathsf{v}$, and $\mathfrak{x}_\mathsf{v}(\mathsf{v}) = x$. In other words, given a vertex $\mathsf{v} \in \mathsf{V}$, $\mathfrak{x}_\mathsf{v}$ denotes the group element $x \in G$ regarded as a map $\mathsf{V} \to G$.[1] The set of such functions is naturally isomorphic to $G$. Finally, to every $\mathsf{v} \in \mathsf{V}$, we assign the local operators

$$\mathbb{h}_{\mathsf{v},n}^\psi := \sum_{\mathfrak{g},\mathfrak{x}_\mathsf{v}} h_n(\mathfrak{g}, \mathfrak{x}_\mathsf{v})\, \psi(\mathfrak{g}, \mathfrak{x}_\mathsf{v}) \, |\mathfrak{g}\, \mathrm{d}\mathfrak{x}_\mathsf{v}\rangle\!\langle\mathfrak{g}| \,, \tag{2.1}$$

wherein $n$ is some label, $h_n(\mathfrak{g}, \mathfrak{x}_\mathsf{v})$ are complex coefficients, $\mathfrak{g}\, \mathrm{d}\mathfrak{x}_\mathsf{v} : \mathsf{E} \to G$ is the function such that

$$(\mathfrak{g}\, \mathrm{d}\mathfrak{x}_\mathsf{v})(\mathsf{e}) := \begin{cases} x\mathfrak{g}(\mathsf{e}) & \text{if } \mathsf{e} \equiv (\mathsf{v}\,\mathsf{v}_2) \\ \mathfrak{g}(\mathsf{e})x^{-1} & \text{if } \mathsf{e} \equiv (\mathsf{v}_1\mathsf{v}) \\ \mathfrak{g}(\mathsf{e}) & \text{otherwise} \end{cases} \quad \text{with} \quad \mathfrak{x}_\mathsf{v}(\mathsf{v}) \equiv x \,, \tag{2.2}$$

and $\psi(\mathfrak{g}, \mathfrak{x}_\mathsf{v})$ is the phase factor

$$\psi(\mathfrak{g}, \mathfrak{x}_\mathsf{v}) := \frac{\psi\big(\mathfrak{g}(\mathsf{v}_1\mathsf{v})x^{-1}, x\big)}{\psi(x, \mathfrak{g}(\mathsf{v}\,\mathsf{v}_2))} \,, \tag{2.3}$$

where $\mathsf{v}_1 < \mathsf{v} < \mathsf{v}_2$, so that $(\mathsf{v}_1\mathsf{v})$ and $(\mathsf{v}\,\mathsf{v}_2)$ are the edges whose target and source vertices are $\mathsf{v}$, respectively.

For any possible choice of complex coefficients $h_n(\mathfrak{g}, \mathfrak{x}_\mathsf{v})$, for every $\mathfrak{g} : \mathsf{E} \to G$ and $\mathfrak{x}_\mathsf{v}$, and cohomology class $[\psi]$, it follows from the defining property of group characters that local operators (2.1) commute with (0-form) symmetry operators of the form

$$\mathbb{U}^\chi := \sum_{\mathfrak{g}} \left( \prod_{\mathsf{e}\in\mathsf{E}} \chi\big(\mathfrak{g}(\mathsf{e})\big) \right) |\mathfrak{g}\rangle\!\langle\mathfrak{g}| \,, \tag{2.4}$$

---

[1] Although this notation may seem excessive at this point, we introduce it here in preparation for the higher-dimensional study.

for any $\chi \in \widehat{G}$. From $(\chi_1 \otimes \chi_2)(g) = \chi_1(g)\chi_2(g)$ we readily infer that the symmetry operators compose according to $\mathbb{U}^{\chi_1} \circ \mathbb{U}^{\chi_2} = \mathbb{U}^{\chi_1 \otimes \chi_2}$. Although it is somewhat superfluous for an abelian group, we can state that symmetry operators are organised into the fusion (1-)category $\mathsf{Rep}(G)$ of (finite-dimensional complex) representations of $G$. Finally, we construct arbitrary $\widehat{G}$-symmetric models as $\mathbb{H}^\psi = \sum_{\mathsf{v} \in \mathsf{V}} \sum_n \mathbb{h}^\psi_{\mathsf{v},n}$. Notice that we restricted to local operators acting on nearest neighbours for simplicity but nothing prevents us from defining local operators acting on additional sites.

Up until this point we have implicitly assumed periodic boundary conditions. Symmetry-twisted closed boundary conditions are then obtained by inserting symmetry twists, which—in contrast to symmetry operators that extend over the whole space—are localised at one spatial point and extend in the time direction. The resulting symmetry-twisted boundary conditions preserve the translation invariance of the system, in the sense that there is an isomorphism between Hilbert spaces obtained by moving the defect by one site [AFM20, LDV22, LOST22, Sei23]. More specifically, we promote symmetry-twisted boundary conditions to genuine physical degrees of freedom by tensoring the microscopic Hilbert space $\mathrm{Span}_\mathbb{C}\{|\mathfrak{g}\rangle \,|\, \mathfrak{g} : \mathsf{E} \to G\}$ with a copy of $\mathbb{C}[\widehat{G}]$ associated with a vertex conventionally denoted by $\mathsf{v}_0$. Fixing the boundary condition to be that labelled by $\eta \in \widehat{G}$ then amounts to setting the $\mathbb{C}[\widehat{G}]$ degree of freedom to $|\eta\rangle$, and we denote the resulting kinematical Hilbert space by $\mathcal{H}^\eta$. Local operators across the symmetry defect labelled by $\eta \in \widehat{G}$ are acted upon by the symmetry action as follows:

$$\mathbb{h}^{\psi,\eta}_{\mathsf{v}_0,n} := \sum_{\mathfrak{g}, \mathfrak{x}_{\mathsf{v}_0}} h_n(\mathfrak{g}, \mathfrak{x}_{\mathsf{v}_0}) \, \psi(\mathfrak{g}, \mathfrak{x}_{\mathsf{v}_0}) \, \eta(\mathfrak{x}_{\mathsf{v}_0}) \, |\mathfrak{g}\, \mathrm{d}\mathfrak{x}_{\mathsf{v}_0}\rangle\langle\mathfrak{g}| \,, \tag{2.5}$$

with $\eta(\mathfrak{x}_{\mathsf{v}_0}) \equiv \eta(\mathfrak{x}_{\mathsf{v}_0}(\mathsf{v}_0))$. The Hamiltonian in the presence of this symmetry twist hence reads

$$\mathbb{H}^{\psi,\eta} := \sum_{\substack{\mathsf{v} \in \mathsf{V} \\ \mathsf{v} \neq \mathsf{v}_0}} \sum_n \mathbb{h}^\psi_{\mathsf{v},n} + \sum_n \mathbb{h}^{\psi,\eta}_{\mathsf{v}_0,n} \,. \tag{2.6}$$

In order to accommodate the presence of a symmetry twist labelled by $\eta \in \widehat{G}$, the symmetry operators (2.4) need to be modified as follows:

$$\mathbb{U}^\chi_\eta := \sum_{\mathfrak{g}} \left( \prod_{\mathsf{e} \in \mathsf{E}} \chi\big(\mathfrak{g}(\mathsf{e})\big) \right) |\mathfrak{g}\rangle\langle\mathfrak{g}| \otimes |\eta\rangle\langle\eta| \,. \tag{2.7}$$

Again, it readily follows from the defining property of group characters that $[\mathbb{h}^{\psi,\eta}_{\mathsf{v}_0,n}, \mathbb{U}^\chi_\eta] = 0$. Given a choice of symmetry-twisted closed boundary condition $\eta \in \widehat{G}$, it follows from the $\widehat{G}$-symmetry that a Hamiltonian $\mathbb{H}^{\psi,\eta}$ decomposes into charge sectors labelled by group elements $g \in G$ corresponding to the holonomy of $\mathfrak{g}$ around the spatial manifold. Pairs $(g, \eta) \in G \times \widehat{G}$ label the (simple) *topological sectors* of the theory that decompose the action of the algebra of operators (2.7) onto the microscopic Hilbert space. The projector onto the topological sector $(g, \eta)$ is then given by $\sum_{\chi \in \widehat{G}} \chi^\vee(g) \, \mathbb{U}^\chi_\eta$, where $\chi^\vee$ denotes the complex conjugate character of $\chi$ and where we are using the fact

$$\sum_{\chi \in \widehat{G}} \chi^\vee(g) \prod_{\mathsf{e} \in \mathsf{E}} \chi\big(\mathfrak{g}(\mathsf{e})\big) = \sum_{\chi \in \widehat{G}} \chi^\vee(g) \, \chi\big(\mathrm{hol}(\mathfrak{g})\big) = \delta_{g, \mathrm{hol}(\mathfrak{g})} \,, \tag{2.8}$$

with $\mathrm{hol}(\mathfrak{g}) := \prod_{\mathsf{e} \in \mathsf{E}} \mathfrak{g}(\mathsf{e})$.

## 2.2 Duality operators

Given a finite abelian group $G$, we constructed in sec. 2.1 families of one-dimensional quantum lattice models with $\mathsf{Rep}(G)$ symmetry, which are parameterised by normalised representatives $\psi$ of a cohomology class $[\psi]$ in $H^2(G, \mathrm{U}(1))$ as well as sets of complex coefficients $\{h_n\}_n$. We claim that models

constructed in this way that only differ in the choice of cohomology class $[\psi]$ are dual to one another. One way to confirm this statement would be to verify that (von Neumann) algebras of local operators generated by the Hamiltonian terms are isomorphic, which follows from the 2-cocycle condition of $\psi$. Another possibility is to explicitly compute the lattice operators transmuting Hamiltonians associated with distinct choices of $[\psi]$ onto one another. Without loss of generality, let us focus on the relation between Hamiltonian models $\mathbb{H}^\psi$ and $\mathbb{H}$ with and without non-trivial 2-cocycle. We know from the results of ref. [LDOV21] that operators transmuting local symmetric operators of two Hamiltonians that only differ in such a choice of 2-cocycle $\psi$ onto one another are labelled by projective representations of $G$ with Schur's multiplier $\psi$. Formally, this is the statement that these operators are encoded into the category $\mathsf{Rep}^\psi(G)$ of $\psi$-projective representations of $G$.

Let us verify this statement explicitly. To this end, let $\pi$ denote the representative of an isomorphism class of simple objects in $\mathsf{Rep}^\psi(G)$. The operator transmuting the Hamiltonian $\mathbb{H}$ into $\mathbb{H}^\psi$ with periodic boundary conditions explicitly reads

$$\mathbb{D}^\pi := \sum_{\mathfrak{g}} \mathrm{tr}\left( \prod_{\mathsf{e}\in\mathsf{E}} \pi\big(\mathfrak{g}(\mathsf{e})\big)\right) |\mathfrak{g}\rangle\langle\mathfrak{g}|\,, \tag{2.9}$$

where the product over edges $\mathsf{e}\in\mathsf{E}$ is ordered. The operator $\mathbb{D}^\pi$ acts diagonally in the computational basis. We claim that $\mathbb{D}^\pi$ transmutes $\mathbb{H}$ into $\mathbb{H}^\psi$, i.e., $\mathbb{D}^\pi \circ \mathbb{H} = \mathbb{H}^\psi \circ \mathbb{D}^\pi$. Indeed letting $\mathsf{v}\in\mathsf{V}$ and $\mathfrak{r}_\mathsf{v}: \mathsf{V}\to G$ such that $\mathfrak{r}_\mathsf{v}(\mathsf{v}_1) = \mathbb{1}$, whenever $\mathsf{v}_1 \neq \mathsf{v}$, it follows from the definition of a projective representation that

$$\pi\big((\mathfrak{g}\,\mathrm{d}\mathfrak{r}_\mathsf{v})(\mathsf{v}_1\mathsf{v})\big) = \pi\big(\mathfrak{g}(\mathsf{v}_1\mathsf{v})x^{-1}\big) = \frac{1}{\psi\big(\mathfrak{g}(\mathsf{v}_1\mathsf{v}),x^{-1}\big)}\,\pi\big(\mathfrak{g}(\mathsf{v}_1\mathsf{v})\big)\,\pi(x^{-1})\,,$$
$$\pi\big((\mathfrak{g}\,\mathrm{d}\mathfrak{r}_\mathsf{v})(\mathsf{v}\,\mathsf{v}_2)\big) = \pi\big(x\mathfrak{g}(\mathsf{v}\,\mathsf{v}_2)\big) = \frac{1}{\psi\big(x,\mathfrak{g}(\mathsf{v}\,\mathsf{v}_2)\big)}\,\pi(x)\,\pi\big(\mathfrak{g}(\mathsf{v}\,\mathsf{v}_2)\big)\,, \tag{2.10}$$

where $\mathfrak{r}_\mathsf{v}(\mathsf{v}) \equiv x$. Bringing everything together, one obtains

$$\prod_{(\mathsf{v}_1\mathsf{v}_2)} \pi\big((\mathfrak{g}\,\mathrm{d}\mathfrak{r}_\mathsf{v})(\mathsf{v}_1\mathsf{v}_2)\big) = \psi(\mathfrak{g},\mathfrak{r}_\mathsf{v})\cdot \prod_{(\mathsf{v}_1\mathsf{v}_2)} \pi\big(\mathfrak{g}(\mathsf{v}_1\mathsf{v}_2)\big)\,, \tag{2.11}$$

where $\psi(\mathfrak{g},\mathfrak{r}_\mathsf{v})$ was defined in eq. (2.3). This guarantees that $\mathbb{D}^\pi \circ \mathbb{H} = \mathbb{H}^\psi \circ \mathbb{D}^\pi$, as desired. Notice how this operation only depends on $\psi$ and not the specific choice of simple object in $\mathsf{Rep}^\psi(G)$.

Before proceeding further, let us introduce a tensor network representation of the duality operator (2.9). Given the tensor

$$\begin{array}{c}\text{[tensor diagram]}\end{array} \equiv \sum_{g_1,g_2\in G}\sum_{i_1,i_2=1}^{\dim_{\mathbb{C}}\pi} \begin{array}{c}\text{[tensor diagram]}\end{array} |g_2\rangle\langle g_1| \otimes |i_1\rangle\langle i_2| \equiv \sum_{\substack{g\in G \\ i_1,i_2}} \pi(g)_{i_1 i_2}\, |g\rangle\langle g|\otimes |i_1\rangle\langle i_2|\,, \tag{2.12}$$

one can realise the duality operator $\mathbb{D}^\pi$ with periodic boundary conditions as a tensor network of the form

$$\mathbb{D}^\pi \equiv \begin{array}{c}\text{[tensor network diagram]}\end{array}\,, \tag{2.13}$$

where opposite indices in the horizontal direction are contracted so as to implement periodic boundary conditions. Restricting to the charge sector labelled by $g\in G$, it follows from the defining property of

the projective representation $\pi$ that the duality operator explicitly depends on the complex number $\operatorname{tr}\pi(g) = \operatorname{tr}\pi\big(\mathrm{hol}(\mathfrak{g})\big)$. Indeed, repeatedly using the defining property of $\psi$-projective representation $\pi$ one finds

$$\mathbb{D}^{\pi} = \sum_{\mathfrak{g}} \psi\big(\mathfrak{g}(\mathsf{v}_1\mathsf{v}_2), \mathfrak{g}(\mathsf{v}_2\mathsf{v}_3)\big)\, \psi\big(\mathfrak{g}(\mathsf{v}_1\mathsf{v}_2)\mathfrak{g}(\mathsf{v}_2\mathsf{v}_3), \mathfrak{g}(\mathsf{v}_3\mathsf{v}_4)\big) \cdots$$
$$\cdots \psi\big(\mathfrak{g}(\mathsf{v}_1\mathsf{v}_2)\mathfrak{g}(\mathsf{v}_2\mathsf{v}_3)\cdots\mathfrak{g}(\mathsf{v}_{|\mathsf{V}|-1}\mathsf{v}_{|\mathsf{V}|}), \mathfrak{g}(\mathsf{v}_{|\mathsf{V}|}\mathsf{v}_1)\big)\, \operatorname{tr}\pi\big(\mathrm{hol}(\mathfrak{g})\big)\, |\mathfrak{g}\rangle\!\langle\mathfrak{g}|\,, \tag{2.14}$$

where $\mathsf{v}_1 < \mathsf{v}_2 < \ldots < \mathsf{v}_{|\mathsf{V}|}$. Graphically, this follows from

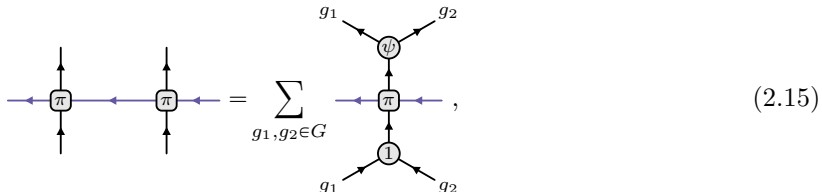

$$\tag{2.15}$$

where we introduced the tensors

$$\tag{2.16}$$

for any normalised representative $\psi$ of $[\psi] \in H^2(G, \mathrm{U}(1))$. Besides, we can intuitively infer from this graphical identity that duality operators transmuting Hamiltonians $\mathbb{H}^{\psi'}$ into $\mathbb{H}^{\psi}$ would be labelled by simple objects in $\mathsf{Rep}^{\psi/\psi'}(G)$.

The duality operator acting diagonally in the computational basis, any charge sector $g \in G$ is preserved as we transmute $\mathbb{H}$ into $\mathbb{H}^{\psi}$. But, notice that

$$\operatorname{tr}\pi(g) = \operatorname{tr}\pi(gxx^{-1}) = \frac{\psi(g,x)}{\psi(x,g)}\,\operatorname{tr}\pi(g)\,, \tag{2.17}$$

for all $x \in G$. This implies that the duality operator forces both $\mathbb{H}$ and $\mathbb{H}^{\psi}$ onto charge sectors $g \in G$ for which $\frac{\psi(g,-)}{\psi(-,g)} = 1$. This follows from the fact that we are implicitly assuming periodic boundary conditions for both models. Accessing other charge sectors requires modifying the duality operator, which in turn forces one of the models to have a different symmetry-twisted boundary condition. This is the statement that topological sectors are not mapped identically under the duality. We compute the corresponding permutation of topological sectors in the following.

### 2.3 Permutation of topological sectors and non-invertibility

In order to lift the duality operator $\mathbb{D}^{\pi}$ to a unitary transformation, we need to characterise the interplay between symmetry-twisted closed boundary conditions and charge sectors. First, we need to modify the duality operator so as to accommodate the presence of a symmetry twist. Let $\pi$ be a (unitary) irreducible $\psi$-projective representation and $\chi \in \widehat{G}$. Naturally, the tensor product $\chi \otimes \pi$ is

isomorphic to a $\psi$-projective representation $\tilde{\pi}$ of the same dimension as $\pi$ via an intertwining unitary map

$$\begin{bmatrix} \chi & \pi & | & \tilde{\pi} \\ 1 & - & | & - \end{bmatrix} \in \mathrm{Hom}_{\mathsf{Rep}^\psi(G)}(\chi \otimes \pi, \tilde{\pi}) \tag{2.18}$$

satisfying

$$(\chi \otimes \pi)(g)_{i_1 i_2} = \sum_{i_3, i_4} \begin{bmatrix} \chi & \pi & | & \tilde{\pi} \\ 1 & i_1 & | & i_3 \end{bmatrix}^* \tilde{\pi}(g)_{i_3 i_4} \begin{bmatrix} \chi & \pi & | & \tilde{\pi} \\ 1 & i_2 & | & i_4 \end{bmatrix}. \tag{2.19}$$

A similar definition holds for $\pi \otimes \chi$. From these intertwining maps, we construct the following tensor

$$\equiv \sum_{i_1, i_2} \quad \cdots \quad \equiv \sum_{i_1, i_2, i_3} \begin{bmatrix} \pi & \eta_2 & | & \tilde{\pi} \\ i_1 & 1 & | & i_3 \end{bmatrix}^* \begin{bmatrix} \eta_1 & \pi & | & \tilde{\pi} \\ 1 & i_2 & | & i_3 \end{bmatrix} |\eta_2\rangle\langle\eta_1| \otimes |i_1\rangle\langle i_2|\,, \tag{2.20}$$

for any $\eta_1, \eta_2 \in \widehat{G}$, out of which we construct the duality operator $\mathbb{D}^\pi_{\eta_1 \to \eta_2} : \mathcal{H}^{\eta_1} \to \mathcal{H}^{\eta_2}$ depicted as

$$\mathbb{D}^\pi_{\eta_1 \to \eta_2} \equiv \quad \cdots \quad . \tag{2.21}$$

By definition of the map (2.18) implementing an isomorphism of projective representations, the rank 4 tensor (2.20) satisfies the following invariance property:

$$\cdots = \cdots , \tag{2.22}$$

for all $x \in G$, where we introduced the matrices

$$\cdots \equiv \sum_{\eta \in \widehat{G}} \cdots |\eta\rangle\langle\eta| = \sum_{\eta \in \widehat{G}} \eta(x)\,|\eta\rangle\langle\eta|\,, \qquad \cdots \equiv \sum_{i_1, i_2 = 1}^{\dim_{\mathbb{C}} \pi} \cdots |i_1\rangle\langle i_2| \equiv \sum_{i_1, i_2 = 1}^{\dim_{\mathbb{C}} \pi} \pi(x)_{i_1 i_2}\,|i_1\rangle\langle i_2|\,. \tag{2.23}$$

It follows from eq. (2.10) in combination with the invariance property (2.22) that $\mathbb{D}^\pi_{\eta_1 \to \eta_2} \circ \mathbb{H}^{1, \eta_1} = \mathbb{H}^{\psi, \eta_2} \circ \mathbb{D}^\pi_{\eta_1 \to \eta_2}$. As suggested above, such a duality operator mapping boundary condition $\eta_1 \in \widehat{G}$ onto $\eta_2 \in \widehat{G}$ puts some constraints on the charge sectors allowed, which must be the same on both side of the duality. Conversely, a choice of boundary condition $\eta_1 \in \widehat{G}$ and charge sector $g \in G$, fully specifies the boundary condition $\eta_2 \in \widehat{G}$. Concretely, invoking eq. (2.14), one can consider the duality operator effectively acting on a single site whose degree of freedom is labelled by the charge sector $g$, which combined with the invariance property (2.22) results for every $x \in G$ in

$$\cdots = \frac{\psi(x, g)}{\psi(g, x)} \cdots , \tag{2.24}$$

<p align="center">~ 8 ~</p>

where in the spirit of eq. (2.15), the diagrams should here be interpreted as tensor networks whose opposite indices in the horizontal direction are contracted so as to impose periodic boundary conditions. In obtaining this identity, we made use of $\pi(x)\pi(g) = \frac{\psi(x,g)}{\psi(g,x)}\pi(g)\pi(x)$. Since the previous derivation holds for any $x \in G$, we find that the boundary condition $\eta_2$ is related to $\eta_1$ via

$$\eta_2(-) \stackrel{!}{=} \frac{\psi(-,g)}{\psi(g,-)}\,\eta_1(-)\,. \tag{2.25}$$

Putting everything together, upon a duality that amounts to performing a twisted gauging of the trivial sub-symmetry, the group of topological sectors undergo the following automorphism in $\mathrm{Aut}(G \times \widehat{G})$ [NR14]:

$$(g,\eta) \mapsto (g,\eta_g^\psi) \quad \text{with} \quad \eta_g^\psi(-) := \frac{\psi(-,g)}{\psi(g,-)}\,\eta(-)\,. \tag{2.26}$$

In particular, it follows from the normalisation of $\psi$ that in the singlet sector $g = \mathbb{1}$ the boundary condition is always preserved.

Before considering some examples, let us comment on the *non-invertibility* of the duality operations studied in this section. We already noticed that fixing periodic boundary conditions on both sides of the duality, the operator $\mathbb{D}_{\mathbb{1}\to\mathbb{1}}^\pi$ projects onto certain charge sectors, and as such it is non-invertible. Nevertheless, restricting to such a sector, acting with the corresponding projector on $\mathbb{D}^\pi \equiv \mathbb{D}_{\mathbb{1}\to\mathbb{1}}^\pi$ produces the unitary map relating the spectra of the dual Hamiltonians. More generally, given compatible sectors as per eq. (2.26), projecting the duality operator performing the mapping $\eta \mapsto \eta_g^\psi$ onto the charge sector $g \in G$ produces a unitary map. More specifically about the non-invertibility, instead of transmuting $\mathbb{H}$ into $\mathbb{H}^\psi$, we can transmute $\mathbb{H}^\psi$ into $\mathbb{H}$ via a duality operator labelled by a projective representation in $\mathsf{Rep}^{1/\psi}(G)$. Given a projective representation $\pi$ with Schur's multiplier $\psi$, its contragredient representation $\pi^\vee$ is a projective representation with Schur's multiplier $1/\psi$. Indeed, $\pi^\vee$ is defined as $\pi^\vee(g) := ((\pi(g)^{-1})^\vee : V^\vee \to V^\vee)$, where $\pi(g)^\vee$ is the transpose of $\pi(g)$ and $V^\vee := \mathrm{Hom}(V,\mathbb{C})$. It follows from $\pi(g)^{-1} = \frac{1}{\psi(g,g^{-1})}\pi(g^{-1})$ that

$$\pi^\vee(g_1)\,\pi^\vee(g_2) = \frac{1}{\psi(g_1,g_2)}\,\pi^\vee(g_1 g_2)\,, \tag{2.27}$$

for all $g_1, g_2 \in G$, as expected. In particular, we have that $\pi^\vee \otimes \pi$ is a linear representation of $G$, where $\otimes$ is the usual tensor product, and thus labels a symmetry operator of $\mathbb{H}$. Defining duality operators $\mathbb{D}_{\eta_2\to\eta_1}^{\pi^\vee}$ in the same vein as $\mathbb{D}_{\eta_1\to\eta_2}^\pi$, it is clear for instance that $\mathbb{D}_{\eta_2\to\eta_1}^{\pi^\vee} \circ \mathbb{D}_{\eta_1\to\eta_2}^\pi = \mathbb{D}_{\eta_1\to\eta_1}^{\pi^\vee\otimes\pi}$ does not act as the trivial symmetry operator in general. Nevertheless, it is possible to find a combination of duality operators that boils down to the trivial symmetry operator. For every $\eta \in \widehat{G}$ such that $\pi \otimes \eta \cong \pi$, consider for instance duality operators $\mathbb{D}_{\mathbb{1}\to\eta}^\pi$. But,

$$\dim_\mathbb{C} \mathrm{Hom}_{\mathsf{Rep}^\psi(G)}(\pi \otimes \eta, \pi) = \dim_\mathbb{C} \mathrm{Hom}_{\mathsf{Rep}(G)}(\eta, \pi^\vee \otimes \pi) \leqslant 1\,, \tag{2.28}$$

for every $\eta \in \widehat{G}$. In particular, characters in $\widehat{G}^\pi := \{\eta \in \widehat{G}\,|\,\dim_\mathbb{C} \mathrm{Hom}_{\mathsf{Rep}(G)}(\eta, \pi^\vee \otimes \pi) = 1\}$ form a subgroup of $\widehat{G}$ such that $|\widehat{G}^\pi| = (\dim_\mathbb{C} \pi)^2$. It follows from the unitarity of the intertwining map

$$\left[\begin{smallmatrix} \pi^\vee & \pi & \\ - & - & 1 \end{smallmatrix}\,\middle|\,\begin{smallmatrix} - \\ 1 \end{smallmatrix}\right] \in \mathrm{Hom}_{\mathsf{Rep}(G)}\Big(\pi^\vee \otimes \pi, \bigoplus_{\eta\in\widehat{G}^\pi} \eta\Big) \tag{2.29}$$

that the operator $\sum_{\eta\in\widehat{G}^\pi} \mathbb{D}_{\eta\to\mathbb{1}}^{\pi^\vee} \circ \mathbb{D}_{\mathbb{1}\to\eta}^\pi$ does act like the trivial symmetry operator on the microscopic Hilbert space associated with periodic boundary conditions.

## 2.4 Examples

Let us illustrate our formalism by specialising to the group $G = \mathbb{Z}_p \times \mathbb{Z}_p$, with $p$ prime. We begin by introducing some notations: Throughout this part, we denote group elements of $\mathbb{Z}_p \times \mathbb{Z}_p$ by $g \equiv (g^{(1)}, g^{(2)})$ where $g^{(1)}, g^{(2)} \in \{0, 1, \ldots, p-1\}$ and we define the multiplication rule of $\mathbb{Z}_p$ as addition modulo $p$. Introducing the primitive $p^{\text{th}}$ root of unity $\omega^{(p)} := \exp(\frac{2\pi i}{p})$, a normalised representative 2-cocycle $\psi$ generating the second cohomology group $H^2(\mathbb{Z}_p \times \mathbb{Z}_p, \mathrm{U}(1)) \cong \mathbb{Z}_p$ is given by[2]

$$\psi(g_1, g_2) := (\omega^{(p)})^{g_1^{(2)} g_2^{(1)}}. \tag{2.30}$$

It turns out that for any cohomology class in $H^2(\mathbb{Z}_p \times \mathbb{Z}_p, \mathrm{U}(1))$, there is exactly one irreducible projective representation, up to equivalence, which is of dimension $p$ [Che15]. In other words, the category $\mathsf{Rep}^\psi(\mathbb{Z}_p \times \mathbb{Z}_p)$ admits one isomorphism class of simple objects. Given the generalised Pauli matrices

$$X^{(p)} := \sum_{g=0}^{p-1} |g+1\rangle\langle g| \quad \text{and} \quad Z^{(p)} := \sum_{g=0}^{p-1} (\omega^{(p)})^g |g\rangle\langle g|, \tag{2.31}$$

which satisfy the commutation relation $Z^{(p)} X^{(p)} = \omega^{(p)} X^{(p)} Z^{(p)}$, the projective representation $\pi_k$ associated with the cocycle $\psi^k(g_1, g_2)$ with $k = 0, \ldots, p-1$ explicitly reads

$$\pi_k(g) := (X^{(p)})^{g^{(1)}} (Z^{(p)})^{k g^{(2)}}, \tag{2.32}$$

for all $g \equiv (g^{(1)}, g^{(2)}) \in \mathbb{Z}_p \times \mathbb{Z}_p$. Indeed, one can readily check that $\pi_k(g_1)\pi_k(g_2) = (\omega^{(p)})^{k g_1^{(2)} g_2^{(1)}} \pi_k(g_1 + g_2)$. Elements of the character group of $\mathbb{Z}_p \times \mathbb{Z}_p$ are denoted by $\chi_l \equiv \chi_{(l^{(1)}, l^{(2)})}$ with $l^{(1)}, l^{(2)} \in \{0, 1, \ldots, p-1\}$ and are defined via

$$\chi_l : g \mapsto (\omega^{(p)})^{g^{(1)} l^{(1)} + g^{(2)} l^{(2)}}. \tag{2.33}$$

The tensor product of two characters simply reads

$$\chi_{l_1} \otimes \chi_{l_2} \cong \chi_{(l_1^{(1)} + l_2^{(1)}, l_1^{(2)} + l_2^{(2)})}, \tag{2.34}$$

where the sums are again modulo $p$. Finally, bases of intertwiners realising the isomorphisms $\chi_l \otimes \pi_k \xrightarrow{\sim} \pi_k$ and $\pi_k \otimes \chi_l \xrightarrow{\sim} \pi_k$ are provided by coefficients

$$\begin{bmatrix} \pi_k & \chi_l & | & \pi_k \\ i_1 & 1 & | & i_2 \end{bmatrix} = \begin{bmatrix} \chi_l & \pi_k & | & \pi_k \\ 1 & i_1 & | & i_2 \end{bmatrix} = \left[ (X^{(p)})^m (Z^{(p)})^{-l^{(1)}} \right]_{i_2 i_1}, \tag{2.35}$$

for all $k, l^{(1)}, l^{(2)} \in \{0, \ldots, p-1\}$ and $i_1, i_2 \in \{1, \ldots, p\}$, where $m$ is a solution of $mk = l^{(2)} \pmod{p}$, which is guaranteed to exist since $p$ is prime. Let us confirm the mapping (2.26) of topological sectors using the explicit formula (2.35) for the intertwiners. For concreteness, let us assume periodic boundary conditions for the Hamiltonian $\mathbb{H}$. Given the charge sector $g \in \mathbb{Z}_p \times \mathbb{Z}_p$, let us compute the dual symmetry-twisted boundary condition for the Hamiltonian $\mathbb{H}^{\psi^k}$. We find that the duality operator explicitly depends on the complex number

$$\mathrm{tr}\left[ (Z^{(p)})^{l^{(1)}} (X^{(p)})^{-m} (X^{(p)})^{g^{(1)}} (Z^{(p)})^{k g^{(2)}} \right] = \mathrm{tr}\left[ (X^{(p)})^{g^{(1)} - m} (Z^{(p)})^{k g^{(2)} + l^{(1)}} \right], \tag{2.36}$$

where $mk = l^{(2)} \pmod{p}$. But, since $p$ is prime, this factor is non-vanishing if and only if $m = g^{(1)}$ and $l^{(1)} = -k g^{(2)} \pmod{p}$. This indicates that the boundary condition of the dual theory must be provided by the character

$$g \mapsto (\omega^{(p)})^{-g^{(1)} k g^{(2)} + g^{(2)} g^{(1)} k}, \tag{2.37}$$

---

[2]In defining $\psi(g_1, g_2)$, we are implicitly promoting $\mathbb{Z}_p$ to the corresponding Galois field $\mathbb{F}_p$.

which indeed coincides with $\psi^k(-, g)/\psi^k(g, -)$.

It is instructive to further specialise to the case $p = 2$. Let us consider a representative of the gapped phase where the $\widehat{\mathbb{Z}_2} \times \widehat{\mathbb{Z}_2}$ is spontaneously broken in the ground state subspace for periodic boundary conditions. To make contact with the literature, we think of the edges of the lattice as the sites of a chain of length $|\mathsf{E}|$, which we enumerate $\mathsf{i} = 1, \ldots, |\mathsf{E}|$. The microscopic Hilbert space is then given by $\bigotimes_{\mathsf{i}=1}^{|\mathsf{E}|} \mathbb{C}[\mathbb{Z}_2 \times \mathbb{Z}_2]$ and the Hamiltonian encodes a pure ferromagnet:

$$\mathbb{H} = -\sum_{\mathsf{i}=1}^{|\mathsf{E}|} \left[ (X\mathbb{1})_\mathsf{i} (X\mathbb{1})_{\mathsf{i}+1} + (\mathbb{1}X)_\mathsf{i} (\mathbb{1}X)_{\mathsf{i}+1} \right], \tag{2.38}$$

where $X \equiv X^{(2)}$ is the usual Pauli matrix. In the notation of eq. (2.1), this is the Hamiltonian $\mathbb{H} = \sum_\mathsf{v} \mathbb{h}_{\mathsf{v},1}$ obtained by choosing non-vanishing coefficients $h_1(\mathfrak{g}, \mathfrak{x}_\mathsf{v}) = -1$ for any $\mathfrak{g} : \mathsf{E} \to G$ and $\mathfrak{x}_\mathsf{v}$ such that $\mathfrak{x}_\mathsf{v}(\mathsf{v}) \equiv (1,0)$ or $(0,1)$. Replacing the implicit trivial 2-cocycle in $H^2(\mathbb{Z}_2 \times \mathbb{Z}_2, \mathrm{U}(1)) \cong \mathbb{Z}_2$ by the normalised representative $\psi : (g_1, g_2) \mapsto (-1)^{g_1^{(2)} g_2^{(1)}}$ in the unique non-trivial class yields the dual model

$$\mathbb{H}^\psi = -\sum_{\mathsf{i}=1}^{|\mathsf{E}|} \left[ (XZ)_\mathsf{i} (X\mathbb{1})_{\mathsf{i}+1} + (\mathbb{1}X)_\mathsf{i} (ZX)_{\mathsf{i}+1} \right], \tag{2.39}$$

which we recognise as the (1+1)d *cluster state* model [BR01]. This model is the stable renormalisation group fixed point of the non-trivial $\widehat{\mathbb{Z}_2} \times \widehat{\mathbb{Z}_2}$ symmetry protected topological (SPT) phase. Within this context, the duality operator $\mathbb{D}^\pi$ implements the celebrated *Kennedy–Tasaki* duality transformation, which was first introduced in [KT92] as a means to elucidate the Haldane phase of the antiferromagnetic spin-1 Heisenberg XXZ model.[3] We count four charge sectors and four boundary conditions that amount to imposing periodic/antiperiodic boundary conditions with respect to each copy of $\mathbb{Z}_2$. The mapping of the 16 topological sectors given by eq. 2.26 coincides with that obtained in ref. [LOZ23a].[4]

As another application of our formalism, let us determine the fate of the $\widehat{\mathbb{Z}_2} \times \widehat{\mathbb{Z}_2}$ ferromagnetic order parameters under the duality. The symmetric ferromagnetic order parameter acting on the first $\mathbb{Z}_2$ component on sites $\mathsf{i} < \mathsf{j}$ away from the boundary condition reads $(X\mathbb{1})_\mathsf{i} (X\mathbb{1})_\mathsf{j}$. The duality operator $\mathbb{D}^\pi$ transmutes this local order parameter into the string order parameter $(XZ)_\mathsf{i} \left( \prod_{\mathsf{k}=\mathsf{i}+1}^{\mathsf{j}-1} (\mathbb{1}Z)_\mathsf{k} \right) (X\mathbb{1})_\mathsf{j}$, where the product is over all sites between $\mathsf{i}$ and $\mathsf{j}$. One proceeds similarly for the second $\mathbb{Z}_2$ component. Graphically, the mapping simply follows from

$$\boxed{x} - \boxed{\pi} - \boxed{x} \quad = \quad \frac{\psi(g, x)}{\psi(x, g)} \quad \boxed{\pi} \quad , \tag{2.40}$$

---

[3] The fact that the duality as formulated in this manuscript coincides with the Kennedy–Tasaki duality in the context of the spin-1 Heisenberg XXZ model was demonstrated in ref. [LDOV21].

[4] Note that we can add to the Hamiltonian $\mathbb{H}$ symmetric terms of the form $-\sum_\mathsf{i} (Z\mathbb{1})_\mathsf{i}$. The resulting model would describe a symmetry-breaking ferromagnetic interaction supported on the second copy of $\mathbb{Z}_2$ and a decoupled critical transverse-field Ising chain supported on the first copy of $\mathbb{Z}_2$. The corresponding dual Hamiltonian would then be given by $\mathbb{H}^\psi - \sum_\mathsf{i} (Z\mathbb{1})_\mathsf{i}$. An effective model describing the low-energy physics of this Hamiltonian was first examined in ref. [SPV17] and was termed a *gapless SPT phase*. In ref. [LOZ23b], this and other gapless SPT phases were constructed by making use of the Kennedy–Tasaki duality. It is plausible that our approach would facilitate the construction of other examples of gapless SPT phases as well as the study of their properties.

as well as the symmetry

$$
\vcenter{\hbox{\includegraphics{fig}}} = \psi(g,x) \quad \vcenter{\hbox{\includegraphics{fig}}} = \psi(x,g) \quad \vcenter{\hbox{\includegraphics{fig}}} \quad \text{with} \quad \boxed{x} \equiv \sum_{g \in \mathbb{Z}_2} |g\rangle\langle g + x|\,, \tag{2.41}
$$

for any $x \in \mathbb{Z}_2 \times \mathbb{Z}_2$.

## 2.5 Category theoretic underpinnings

In preparation for the following section, let us delve a little bit into the mathematical formalism underlying the results presented above, following the approach of ref. [LDOV21, LDV22, LDWV23]. Consider a one-dimensional quantum lattice model with a non-anomalous abelian symmetry group $G$. In modern parlance, the model is said to admit topological lines valued in $G$ [GKSW14]. The requirement that we should be able to construct junctions (in spacetime) of such topological lines invites us to promote the group $G$ to a higher mathematical structure [BT17], namely the *fusion category* $\mathsf{Vec}_G$ of $G$-graded vector spaces. In particular, simple objects in $\mathsf{Vec}_G$ are one-dimensional vector spaces $\mathbb{C}_g$, for every $g \in G$, such that $(\mathbb{C}_{g_1})_{g_2} = \delta_{g_1,g_2}\mathbb{C}$, $\mathrm{Hom}_{\mathsf{Vec}_G}(\mathbb{C}_{g_1}, \mathbb{C}_{g_2}) \cong \delta_{g_1,g_2}\mathbb{C}$ and $\mathbb{C}_{g_1} \otimes \mathbb{C}_{g_2} = \mathbb{C}_{g_1 g_2}$. Moreover, the *monoidal associator* evaluates to the identity so that $1_{\mathbb{C}_{g_1 g_2 g_3}} : (\mathbb{C}_{g_1} \otimes \mathbb{C}_{g_2}) \otimes \mathbb{C}_{g_3} \xrightarrow{\sim} \mathbb{C}_{g_1} \otimes (\mathbb{C}_{g_2} \otimes \mathbb{C}_{g_3})$. Given a non-anomalous symmetry $G$—or rather, a symmetry $\mathsf{Vec}_G$—it is always possible to gauge a subsymmetry $A$, where $A$ is a subgroup of $G$. There are several ways to perform such a gauging, classified by the second cohomology group $H^2(A, \mathrm{U}(1))$. Whenever the whole symmetry is gauged, the resulting theory possesses a $\mathsf{Rep}(G)$-symmetry, i.e., a symmetry whose operators are labelled by representations of the group $G$ and compose according to the tensor product of representations. Since the group $G$ is assumed to be abelian, we further have the equivalence $\mathsf{Rep}(G) \simeq \mathsf{Vec}_{\hat{G}}$.

Mathematically, a pair $(A, \psi)$ consisting of a subgroup $A \subseteq G$ and a normalised representative $\psi$ of a cohomology $[\psi] \in H^2(A, \mathrm{U}(1))$ specifies a so-called (finite semisimple indecomposable) *module category* over $\mathsf{Vec}_G$ [Ost02]. Let $\mathcal{M}(A, \psi)$ be the category whose simple objects are one-dimensional vector spaces $\mathbb{C}_{rA}$, for every $rA \in G/A$, equipped with the (left) $\mathsf{Vec}_G$-module structure provided by the $G$-action of left cosets in $G/A$, i.e. $\mathbb{C}_g \triangleright \mathbb{C}_{rA} := \mathbb{C}_{g \triangleright rA} = \mathbb{C}_{(gr)A}$, and *module associator* $\alpha_\psi^\triangleright$ specified by isomorphisms

$$
\alpha_\psi^\triangleright(g_1, g_2)(rA) \cdot 1_{\mathbb{C}_{(g_1 g_2) \triangleright rA}} : (\mathbb{C}_{g_1} \otimes \mathbb{C}_{g_2}) \triangleright \mathbb{C}_{rA} \xrightarrow{\sim} \mathbb{C}_{g_1} \triangleright (\mathbb{C}_{g_2} \triangleright \mathbb{C}_{rA})\,, \tag{2.42}
$$

for every $g_1, g_2 \in G$ and $rA \in G/A$. The 2-cochain $\alpha_\psi^\triangleright \in C^2(G, \mathrm{Fun}(G/A, \mathrm{U}(1)))$ is defined in terms of the 2-cocycle $\psi$ via

$$
\alpha_\psi^\triangleright(g_1, g_2)(rA) := \psi(a_{g_1, g_2 \triangleright rA}, a_{g_2, rA})\,, \tag{2.43}
$$

where given $g \in G$ and $rA, sA \in G/A$ such that $g \triangleright rA = sA$, $a_{g,rA}$ is the unique group element in $A$ satisfying $g = s\, a_{g,rA}\, r^{-1}$. It follows from the 2-cocycle condition satisfied by $\psi$ and the property $a_{g_1, g_2 \triangleright rA}\, a_{g_2, rA} = a_{g_1 g_2, rA}$, for every $g_1, g_2 \in G$ and $rA \in G/A$, that

$$
\frac{\alpha_\psi^\triangleright(g_2, g_3)(rA)\, \alpha_\psi^\triangleright(g_1, g_2 g_3)(rA)}{\alpha_\psi^\triangleright(g_1 g_2, g_3)(rA)\, \alpha_\psi^\triangleright(g_1, g_2)(g_3 \triangleright rA)} = 1\,, \tag{2.44}
$$

for every $g_1, g_2, g_3 \in G$ and $rA \in G/A$. Identity (2.44) guarantees the so-called *pentagon axiom* of $\mathcal{M}(A, \psi)$.

Performing the $\psi$-twisted gauging of the symmetry $G$ amounts to picking the $\mathsf{Vec}_G$-module category $\mathcal{M}(G,\psi) \equiv \mathsf{Vec}^\psi$, which is equivalent to $\mathsf{Vec}$ as a category but whose $\mathsf{Vec}_G$-module structure depends on $\psi$. The symmetry structure of the theory resulting from the $\psi$-twisted gauging of the sub-symmetry $A$ is provided by the fusion category $(\mathsf{Vec}_G)^\vee_{\mathcal{M}(A,\psi)} := \mathsf{Fun}_{\mathsf{Vec}_G}(\mathcal{M}(A,\psi),\mathcal{M}(A,\psi))$ defined as the category of $\mathsf{Vec}_G$-module functors from $\mathcal{M}(A,\psi)$ to itself, with fusion structure provided by the composition of $\mathsf{Vec}_G$-module functors [EGNO16]. In particular, we have $(\mathsf{Vec}_G)^\vee_{\mathsf{Vec}^\psi} \simeq \mathsf{Rep}(G)$, for any $[\psi] \in H^2(G,\mathrm{U}(1))$, as expected.

We can think of the $\widehat{G}$-symmetric models defined in sec. 2.1 as resulting from the $\psi$-twisted gauging of $G$-symmetric models. Therefore, two models with Hamiltonians $\mathbb{H}$ and $\mathbb{H}^\psi$—only differing in a choice of 2-cocycle $\psi$—are associated with the $\mathsf{Vec}_G$-module categories $\mathsf{Vec}$ and $\mathsf{Vec}^\psi$, respectively. This can be made very explicit employing the formalism of ref. [LDOV21, LDV22], from which it also follows that a duality operator transmuting $\mathbb{H}$ into $\mathbb{H}^\psi$ is associated with a simple object in $\mathsf{Fun}_{\mathsf{Vec}_G}(\mathsf{Vec},\mathsf{Vec}^\psi) \simeq \mathsf{Rep}^\psi(G)$, as expected. For future reference, it is useful to observe that the category $\mathsf{Rep}^\psi(G)$ of $\psi$-projective representations of $G$ is equivalent to the category $\mathsf{Mod}(\mathbb{C}[G]^\psi)$ of modules over the $\psi$-twisted group algebra $\mathbb{C}[G]^\psi$. Explicitly, $\mathbb{C}[G]^\psi$ is the algebra with underlying vector space $\mathrm{Span}_\mathbb{C}\{|g\rangle \,|\, g \in G\}$ and multiplication $|g_1\rangle \cdot |g_2\rangle = \psi(g_1,g_2)\,|g_1 g_2\rangle$, for every $g_1, g_2 \in G$. In this context, topological sectors of a $\widehat{G}$-symmetric model correspond to simple objects in the so-called *Drinfel'd center* $\mathcal{Z}(\mathsf{Vec}_{\widehat{G}})$ of the symmetry fusion category $\mathsf{Vec}_{\widehat{G}}$ [LDV22, LOST22], which are indeed labelled by group elements in $G \times \widehat{G}$.

## 2.6   Self-duality

As announced in the introduction, an appealing feature of the dualities we consider is that they strictly preserve the symmetry structure. Therefore, it is sensible to ask for (1+1)d models that would be left invariant under such duality transformations. Supposing we have such a model, one can further ask to promote the self-duality to a genuine internal symmetry, which amounts to performing a group extension of the symmetry fusion category.

Let us illustrate this procedure for the case $G = \mathbb{Z}_2 \times \mathbb{Z}_2$. The Hamiltonian $\mathbb{H} + \mathbb{H}^\psi$ constructed from the Hamiltonians (2.38) and (2.39), respectively, is left invariant under the action of the duality operator (2.9). Out of the possible extensions of $\mathsf{Rep}(\mathbb{Z}_2 \times \mathbb{Z}_2)$ by $\mathbb{Z}_2$ computed in ref. [TY98], promoting the self-duality operator to a symmetry operator results in the fusion category $\mathsf{Rep}(\mathbb{D}_8)$ of representations of the dihedral group $\mathbb{D}_8$ of order eight, which, as a category, is equivalent to $\mathsf{Rep}(\mathbb{Z}_2 \times \mathbb{Z}_2) \boxplus \mathsf{Rep}^\psi(\mathbb{Z}_2 \times \mathbb{Z}_2)$. One could verify this fact by computing tensors implementing junctions between the symmetry operators in $\mathsf{Rep}(\mathbb{Z}_2 \times \mathbb{Z}_2)$ and the self-duality operator, and show that these correspond to morphisms in $\mathsf{Rep}(\mathbb{D}_8)$. Instead, we will show that the Hamiltonian $\mathbb{H} + \mathbb{H}^\psi$ can be constructed from local operators that are manifestly $\mathsf{Rep}(\mathbb{D}_8)$ symmetric.

We begin by noticing that the dihedral group $\mathbb{D}_8$ with presentation $\langle r, s \,|\, r^4 = s^2 = (rs)^2 = \mathbb{1}\rangle$ can be realised as the *central extension*

$$1 \to \mathbb{Z}_2 \to \mathbb{D}_8 \cong \mathbb{Z}_2 \times_\phi (\mathbb{Z}_2 \times \mathbb{Z}_2) \to \mathbb{Z}_2 \times \mathbb{Z}_2 \to 1 \tag{2.45}$$

specified by the non-trivial cohomology class in $H^2(\mathbb{Z}_2 \times \mathbb{Z}_2, \mathbb{Z}_2) \cong \mathbb{Z}_2$ with normalised representative $\phi$ such that $\phi(g_1,g_2) = g_1^{(2)} + g_2^{(1)} \in \mathbb{Z}_2$ for every $g_1, g_2 \in \mathbb{Z}_2 \times \mathbb{Z}_2$. In particular, we notice the close similarity between the definitions of the 2-cocycles $\psi$ and $\phi$. Identifying $\mathbb{1} \equiv (0,(0,0))$, $s \equiv (0,(0,1))$ and $r \equiv (0,(1,1))$, one finds that the set $\mathbb{Z}_2 \times (\mathbb{Z}_2 \times \mathbb{Z}_2)$ equipped with the multiplication rule

$$(n_1, q_1) \cdot (n_2, q_2) := (n_1 + n_2 + \phi(q_1, q_2), q_1 + q_2)\,, \tag{2.46}$$

for every $q_1, q_2 \in \mathbb{Z}_2 \times \mathbb{Z}_2$ and $n_1, n_2 \in \mathbb{Z}_2$, is indeed isomorphic to $\mathbb{D}_8$. Families of one-dimensional quantum lattice models that possess a $\mathsf{Rep}(\mathbb{D}_8)$ symmetry can be easily constructed mimicking the definitions of sec. 2.1. Consider the microscopic Hilbert space $\bigotimes_{\mathsf{e}\in\mathsf{E}} \mathbb{C}[\mathbb{D}_8]$ spanned by states $|\mathfrak{g}\rangle$ where $\mathfrak{g} : \mathsf{E} \to \mathbb{D}_8$. Defining functions $\mathfrak{x}_\mathsf{v} : \mathsf{V} \to \mathbb{D}_8$ as in sec. 2.1, for every $\mathsf{v} \in \mathsf{V}$, consider the following local operators acting on the the kinematical Hilbert space:

$$\mathbb{h}_{\mathsf{v},n} := \sum_{\mathfrak{g},\mathfrak{x}_\mathsf{v}} h_n(\mathfrak{g}, \mathfrak{x}_\mathsf{v}) \, |\mathfrak{g} \, d\mathfrak{x}_\mathsf{v}\rangle\langle\mathfrak{g}| \,, \tag{2.47}$$

wherein $n$ is some label, $h_n(\mathfrak{g}, \mathfrak{x}_\mathsf{v})$ are complex coefficients, and $\mathfrak{g} \, d\mathfrak{x}_\mathsf{v}$ was defined in eq. (2.2). For any possible choice of complex coefficients $h_n(\mathfrak{g}, \mathfrak{x}_\mathsf{v})$, for every $\mathfrak{g} : \mathsf{E} \to \mathbb{D}_8$ and $\mathfrak{x}_\mathsf{v}$, it follows from the defining property of group representations that local operators (2.47) commute with symmetry operators of the form

$$\mathbb{U}^\rho := \sum_{\mathfrak{g}} \mathrm{tr}\left( \prod_{\mathsf{e}\in\mathsf{E}} \rho\big(\mathfrak{g}(\mathsf{e})\big) \right) |\mathfrak{g}\rangle\langle\mathfrak{g}| \,, \tag{2.48}$$

where the product over edges $\mathsf{e} \in \mathsf{E}$ is ordered, for any simple object $\rho$ in $\mathsf{Rep}(\mathbb{D}_8)$. By virtue of $(\rho_1 \otimes \rho_2)(g) = \rho_1(g)\rho_2(g)$, for every $g \in \mathbb{D}_8$, the symmetry operators compose according to $\mathbb{U}^{\rho_1} \circ \mathbb{U}^{\rho_2} = \sum_{\rho_3 \in \rho_1 \otimes \rho_2} \mathbb{U}^{\rho_3}$. In particular, symmetry operators associated with higher-dimensional representations are non-invertible. Finally, we construct arbitrary $\mathsf{Rep}(\mathbb{D}_8)$-symmetric models as $\sum_{\mathsf{v}\in\mathsf{V}} \sum_n \mathbb{h}_{\mathsf{v},n}$.

Let us now reveal how the Hamiltonian $\mathbb{H} + \mathbb{H}^\psi$ is effectively constructed from local operators of the form (2.47). By definition, the Hamiltonian $\mathbb{H} + \mathbb{H}^\psi$, which acts on the kinematical Hilbert space $\bigotimes_i \mathbb{C}[\mathbb{Z}_2 \times \mathbb{Z}_2]$, is made of local operators that act as $-(\mathbb{1}X)_i(\mathbb{1}X)_{i+1} - (\mathbb{1}X)_i(ZX)_{i+1}$ or $-(X\mathbb{1})_i(X\mathbb{1})_{i+1} - (XZ)_i(X\mathbb{1})_{i+1}$. We argue that there is a choice of complex coefficients $h_1(\mathfrak{g}, \mathfrak{x}_\mathsf{v})$ such that local operators of the form (2.47) effectively act in the same way. The crux is to restrict the microscopic Hilbert space to states $|\mathfrak{g}\rangle \equiv |(\mathfrak{n}, \mathfrak{q})\rangle$, with $\mathfrak{q} : \mathsf{E} \to \mathbb{Z}_2 \times \mathbb{Z}_2$ and $\mathfrak{n} : \mathsf{E} \to \mathbb{Z}_2$, such that $\mathfrak{n}$ assigns the identity group element to every edge/site, effectively reducing the microscopic Hilbert space to $\bigotimes_i \mathbb{C}[\mathbb{Z}_2 \times \mathbb{Z}_2]$. This can be enforced via a choice of complex coefficients $h_1(\mathfrak{g}, \mathfrak{x}_\mathsf{v}) \equiv h_1((\mathfrak{n}, \mathfrak{q}), \mathfrak{x}_\mathsf{v})$. One can further choose these coefficients such that the only non-vanishing ones are such that $\mathfrak{x}_\mathsf{v}(\mathsf{v}) = (0, (0,1)) \equiv s$ or $\mathfrak{x}_\mathsf{v}(\mathsf{v}) = (0, (1,0)) \equiv rs$ and evaluate to $-2$. Proceeding as such, it follows from $\phi\big((0,1),(1,0)\big) = 1$ that the local operator acting at site $i$ is a sum of two operators, effectively acting as $-2(\mathbb{1}X)_i(\mathbb{1}X)_{i+1}$ and $-2(X\mathbb{1})_i(X\mathbb{1})_{i+1}$, respectively, while projecting out states $|\mathfrak{g}\rangle$ such that $\mathfrak{g}(i+1) \notin \{\mathbb{1}, s\}$ and $\mathfrak{g}(i) \notin \{\mathbb{1}, rs\}$, respectively. On the effective Hilbert space, this recovers the local operators constitutive of $\mathbb{H} + \mathbb{H}^\psi$. One can now confirm that the two-dimensional representation $\rho$ of $\mathbb{D}_8$, which is such that $\rho(r) = XZ$ and $\rho(s) = Z$ reproduces the action of the self-duality operator.

# Twisted gauging of (2+1)d lattice gauge theories

*Given an arbitrary two-dimensional lattice gauge theory with an abelian invertible 1-form symmetry, we study the non-trivial interplay between symmetry-twisted closed boundary conditions and charge sectors under the duality that amounts to the (untwisted) gauging of the 1-form symmetry, followed by the twisted gauging of the dual 0-form symmetry. Given pairs of compatible boundary conditions and charge sectors, the unitary operators relating the spectra of dual Hamiltonians are explicitly constructed in the form of tensor network operators.*

## 3.1  Local Hamiltonians

Let us begin by constructing a family of two-dimensional lattice gauge theories on the two-torus $\mathbb{T}^2$. Let $\mathbb{T}^2_\triangle$ be a triangulation of $\mathbb{T}^2$, whose sets of vertices, edges and plaquettes are denoted by $\mathsf{V}(\mathbb{T}^2_\triangle)$, $\mathsf{E}(\mathbb{T}^2_\triangle)$ and $\mathsf{P}(\mathbb{T}^2_\triangle)$, respectively. Given an edge $\mathsf{e} \in \mathsf{E}(\mathbb{T}^2_\triangle)$, $-\mathsf{e}$ denotes the edge with opposite orientation. For simplicity, we shall often assume that $\mathbb{T}^2_\triangle$ is the triangular lattice, but our results typically hold more generally. We equip $\mathbb{T}^2_\triangle$ with a total ordering of its vertices, which induces a relative orientation for its edges and plaquettes. Given an oriented edge $\mathsf{e} \in \mathsf{E}(\mathbb{T}^2_\triangle)$, $\partial_-\mathsf{e}$ and $\partial_+\mathsf{e}$ still denote its source and target vertices, respectively, and we identify $\mathsf{e} \equiv (\partial_-\mathsf{e}\, \partial_+\mathsf{e})$. Given a finite *abelian* group $G$,[5] we consider a differential on the spaces of simplicial $G$-valued forms:

$$C^0(\mathbb{T}^2_\triangle, G) \xrightarrow{\mathrm{d}^{(0)}} C^1(\mathbb{T}^2_\triangle, G) \xrightarrow{\mathrm{d}^{(1)}} C^2(\mathbb{T}^2_\triangle, G) \tag{3.1}$$

such that for any $\mathsf{e} \in \mathsf{E}(\mathbb{T}^2_\triangle)$ and $\mathsf{p} \in \mathsf{P}(\mathbb{T}^2_\triangle)$

$$(\mathrm{d}^{(0)}\mathfrak{x})(\mathsf{e}) := \mathfrak{x}(\partial_-\mathsf{e})\mathfrak{x}(\partial_+\mathsf{e})^{-1} \quad \text{and} \quad (\mathrm{d}^{(1)}\mathfrak{g})(\mathsf{p}) := \prod_{\mathsf{e} \subset \partial\mathsf{p}} \mathfrak{g}(\mathsf{e}), \tag{3.2}$$

where the product $\prod_{\mathsf{e} \subset \partial\mathsf{p}}$ is over edges in the boundary of $\mathsf{p}$, whose orientations are assumed to be induced by that of $\mathsf{p}$. The differential is clearly nilpotent, i.e., $\mathrm{d} \circ \mathrm{d} = 0$, and we define the groups $Z^n(\mathbb{T}^2_\triangle, G)$ of *closed* $G$-valued forms and $B^n(\mathbb{T}^2_\triangle, G)$ of *exact* $G$-valued forms as $\ker \mathrm{d}^{(n)}$ and $\operatorname{im} \mathrm{d}^{(n-1)}$, respectively.

To every edge $\mathsf{e} \in \mathsf{E}(\mathbb{T}^2_\triangle)$, we assign a microscopic degree of freedom valued in $\mathbb{C}[G]$. Given $\mathfrak{g} \in C^1(\mathbb{T}^2_\triangle, G)$ such that $\mathfrak{g}(-\mathsf{e}) = \mathfrak{g}(\mathsf{e})^{-1}$, for every $\mathsf{e} \in \mathsf{E}(\mathbb{T}^2_\triangle)$, we notate via $|\mathfrak{g}\rangle$, the function $\mathfrak{g}$ regarded as an element of the tensor product Hilbert space $\bigotimes_\mathsf{e} \mathbb{C}[G]$. The kinematical Hilbert space of the two-dimensional lattice gauge theories on the two-torus is chosen to be the subspace of the microscopic Hilbert space $\bigotimes_\mathsf{e} \mathbb{C}[G]$ that is spanned by vectors $|\mathfrak{g}\rangle$, where $\mathfrak{g} \in Z^1(\mathbb{T}^2_\triangle, G)$. In other words, at every plaquette $\mathsf{p} \in \mathsf{P}(\mathbb{T}^2_\triangle)$, we impose the kinematical constraint that $\prod_{\mathsf{e} \subset \partial\mathsf{p}} \mathfrak{g}(\mathsf{e}) = \mathbb{1}$, promoting $\mathfrak{g}$ to a (1-form) flat gauge field. Since the group $G$ is abelian, we can view these kinematical constraints as Gauß constraints for a 'magnetic' gauge field defined on the Poincaré dual lattice, so that the resulting theory does have the interpretation of a lattice gauge theory.

Let $\lambda$ be a normalised representative 3-cocycle of a cohomology class $[\lambda] \in H^3(G, \mathrm{U}(1))$. Given a vertex $\mathsf{v} \in \mathsf{V}(\mathbb{T}^2_\triangle)$, we denote by $\mathfrak{x}_\mathsf{v} \in C^0(\mathbb{T}^2_\triangle, G)$ a function in $\mathsf{V}(\mathbb{T}^2_\triangle) \to G$ such that $\mathfrak{x}_\mathsf{v}(\mathsf{v}_1) = \mathbb{1}$, whenever $\mathsf{v}_1 \neq \mathsf{v}$, and $\mathfrak{x}_\mathsf{v}(\mathsf{v}) = x$, for some $x \in G$. We then consider the following local operators acting

---

[5] Although we focus on finite abelian groups throughout this manuscript, and occasionally use formalism specific to abelian groups, most of our results readily generalise to the non-abelian case, sometimes requiring little to no modification in the derivations.

on the kinematical Hilbert space:

$$\mathbb{h}_{\mathsf{v},n}^{\lambda} := \sum_{\mathfrak{g}\in Z^1(\mathbb{T}_{\triangle}^2,G)} \sum_{\mathfrak{x}_{\mathsf{v}}} h_n(\mathfrak{g},\mathfrak{x}_{\mathsf{v}})\,\lambda(\mathfrak{g},\mathfrak{x}_{\mathsf{v}})\,|\mathfrak{g}\,\mathrm{d}^{(0)}\mathfrak{x}_{\mathsf{v}}\rangle\langle\mathfrak{g}|\,, \tag{3.3}$$

where $n$ is some label, $h_n(\mathfrak{g},\mathfrak{x}_{\mathsf{v}})$ are arbitrary complex coefficients, $\mathfrak{g}\,\mathrm{d}^{(0)}\mathfrak{x}_{\mathsf{v}}$ is by definition the function $\mathsf{E}[\mathbb{T}_{\triangle}^2]\to G$ such that

$$(\mathfrak{g}\,\mathrm{d}^{(0)}\mathfrak{x}_{\mathsf{v}})(\mathsf{e}) = \begin{cases} x\mathfrak{g}(\mathsf{e}) & \text{if } \mathsf{e}\equiv(\mathsf{v}\,\mathsf{v}_1) \\ \mathfrak{g}(\mathsf{e})x^{-1} & \text{if } \mathsf{e}\equiv(\mathsf{v}_1\mathsf{v}) \\ \mathfrak{g}(\mathsf{e}) & \text{otherwise} \end{cases} \quad\text{with}\quad \mathfrak{x}_{\mathsf{v}}(\mathsf{v})\equiv x\,, \tag{3.4}$$

and $\lambda(\mathfrak{g},\mathfrak{x}_{\mathsf{v}})$ is the phase factor

$$\lambda(\mathfrak{g},\mathfrak{x}_{\mathsf{v}}) := \prod_{(\mathsf{v}_1\mathsf{v}_2\mathsf{v})} \lambda\big(\mathfrak{g}(\mathsf{v}_1\mathsf{v}_2),\mathfrak{g}(\mathsf{v}_2\mathsf{v})x^{-1},x\big)^{\epsilon(\mathsf{v}_1\mathsf{v}_2\mathsf{v})} \prod_{(\mathsf{v}_1\mathsf{v}\mathsf{v}_2)} \lambda\big(\mathfrak{g}(\mathsf{v}_1\mathsf{v})x^{-1},x,\mathfrak{g}(\mathsf{v}\,\mathsf{v}_2)\big)^{-\epsilon(\mathsf{v}_1\mathsf{v}\mathsf{v}_2)}$$
$$\cdot \prod_{(\mathsf{v}\,\mathsf{v}_1\mathsf{v}_2)} \lambda\big(x,\mathfrak{g}(\mathsf{v}\,\mathsf{v}_1),\mathfrak{g}(\mathsf{v}_1\mathsf{v}_2)\big)^{\epsilon(\mathsf{v}\,\mathsf{v}_1\mathsf{v}_2)}\,. \tag{3.5}$$

In the definition (3.5) of the phase factor $\lambda(\mathfrak{g},\mathfrak{x}_{\mathsf{v}})$, the products are over plaquettes adjacent to $\mathsf{v}$ such that $\mathsf{v}_1 < \mathsf{v}_2 < \mathsf{v}$, $\mathsf{v}_1 < \mathsf{v} < \mathsf{v}_2$ and $\mathsf{v} < \mathsf{v}_1 < \mathsf{v}_2$, respectively, while $\epsilon(\mathsf{p}) = \pm 1$ depends on the relative orientation of the plaquette $\mathsf{p}$. Notice that $\mathbb{h}_{\mathsf{v},n}^{\lambda}$ only acts non-trivially within a neighbourhood of the vertex $\mathsf{v}$. Concretely, suppose $\mathbb{T}_{\triangle}^2$ is the triangular lattice, and choose a total ordering of the vertices such that edges of the hexagonal subcomplex centered around $\mathsf{v}$ are oriented as follows:

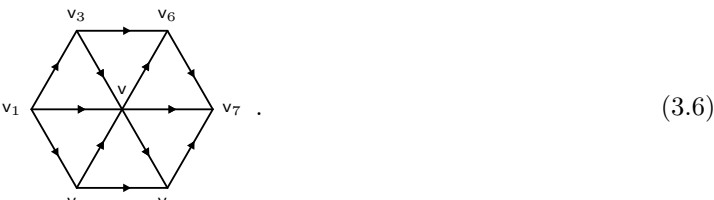

$$\tag{3.6}$$

Orientations above are compatible for instance with a total ordering such that $\mathsf{v}_1 < \mathsf{v}_2 < \mathsf{v}_3 < \mathsf{v} < \mathsf{v}_5 < \mathsf{v}_6 < \mathsf{v}_7$. Under these assumptions, the phase factor (3.5) explicitly reads

$$\lambda(\mathfrak{g},\mathfrak{x}_{\mathsf{v}}) = \frac{\lambda\big(\mathfrak{g}(\mathsf{v}_1\mathsf{v}_2),\mathfrak{g}(\mathsf{v}_2\mathsf{v})x^{-1},x\big)\,\lambda\big(\mathfrak{g}(\mathsf{v}_2\mathsf{v})x^{-1},x,\mathfrak{g}(\mathsf{v}\,\mathsf{v}_5)\big)\,\lambda\big(x,\mathfrak{g}(\mathsf{v}\,\mathsf{v}_5),\mathfrak{g}(\mathsf{v}_5\mathsf{v}_7)\big)}{\lambda\big(\mathfrak{g}(\mathsf{v}_1\mathsf{v}_3),\mathfrak{g}(\mathsf{v}_3\mathsf{v})x^{-1},x\big)\,\lambda\big(\mathfrak{g}(\mathsf{v}_3\mathsf{v})x^{-1},x,\mathfrak{g}(\mathsf{v}\,\mathsf{v}_6)\big)\,\lambda\big(x,\mathfrak{g}(\mathsf{v}\,\mathsf{v}_6),\mathfrak{g}(\mathsf{v}_6\mathsf{v}_7)\big)}\,. \tag{3.7}$$

Finally, arbitrary combinations of local operators (3.3) can be organised into translation invariant local Hamiltonians $\mathbb{H}^{\lambda} = \sum_{\mathsf{v}\in\mathsf{V}(\mathbb{T}_{\triangle}^2)} \mathbb{h}_{\mathsf{v}}^{\lambda} \equiv \sum_{\mathsf{v}\in\mathsf{V}(\mathbb{T}_{\triangle}^2)} \sum_n \mathbb{h}_{\mathsf{v},n}^{\lambda}$. By construction, the Hamiltonian $\mathbb{H}^{\lambda}$ assumes periodic boundary conditions.

### 3.2   Symmetry operators

Given a choice of normalised representative 3-cocycle in a cohomology class $[\lambda] \in H^3(G,\mathrm{U}(1))$, consider a local Hamiltonian $\mathbb{H}^{\lambda}$ defined in terms of local operators (3.3). Regardless of the choices of complex coefficients $h_n(\mathfrak{g},\mathfrak{x}_{\mathsf{v}})$, for every $\mathfrak{g} \in Z^1(\mathbb{T}_{\triangle}^2,G)$ and $\mathfrak{x}_{\mathsf{v}}$, and cohomology class $[\lambda]$, Hamiltonians $\mathbb{H}^{\lambda}$ share the same symmetry structure. First of all, the model with Hamiltonian $\mathbb{H}^{\lambda}$ possesses a 1-form $\widehat{G}$ symmetry—or rather, $\mathsf{Rep}(G)$ symmetry—generated by the *Wilson loop* operators

$$\chi(\ell) \equiv \bigotimes_{\mathsf{e}\subset\ell} \chi(\mathsf{e}) := \sum_{\mathfrak{g}\in Z^1(\mathbb{T}_{\triangle}^2,G)} \left(\prod_{\mathsf{e}\subset\ell} \chi\big(\mathfrak{g}(\mathsf{e})\big)\right) |\mathfrak{g}\rangle\langle\mathfrak{g}|\,, \tag{3.8}$$

where $\chi \in \widehat{G}$, $\ell \in Z_1(\mathbb{T}^2_\triangle, \mathbb{Z})$ is any oriented (simplicial) 1-cycle of $\mathbb{T}^2_\triangle$ and the product $\prod_{\mathsf{e} \subset \ell}$ is over edges along $\ell$ whose orientations are assumed to be induced by that of $\ell$. In particular, whenever the orientation of $\mathsf{e} \subset \ell$ does not agree with that of $\ell$, it contributes a factor $\chi(\mathfrak{g}(-\mathsf{e})) = \chi(\mathfrak{g}(\mathsf{e})^{-1}) = \chi^\vee(\mathfrak{g}(\mathsf{e}))$ to the product. Invoking that $\chi_1(g)\chi_2(g) = (\chi_1 \otimes \chi_2)(g)$, for every $g \in G$ and $\chi_1, \chi_2 \in \widehat{G}$, the composition of operators $\chi_1(\ell) \circ \chi_2(\ell)$ equals $(\chi_1 \otimes \chi_2)(\ell)$. Clearly, it follows from the definition of the kinematical constraints that $\chi(\ell)$ is invariant under continuous deformations of $\ell$, making it a topological line operator. In particular, it implies that $\chi(\ell) = 1$, whenever $\ell$ is a contractible 1-cycle, i.e., $\ell$ is homologically trivial. Whenever $\ell$ is a non-contractible 1-cycle, the commutation relation $[\mathbb{H}^\lambda, \chi(\ell)] = 0$ follows from the fact that

$$\prod_{\mathsf{e} \subset \ell} \chi\big((\mathfrak{g}\, d^{(0)} \mathfrak{r}_\mathsf{v})(\mathsf{e})\big) = \prod_{\mathsf{e} \subset \ell} \chi\big(\mathfrak{g}(\mathsf{e})\big)\,, \tag{3.9}$$

for every vertex $\mathsf{v} \in \mathsf{V}(\mathbb{T}^2_\triangle)$ and function $\mathfrak{r}_\mathsf{v} \in C^0(\mathbb{T}^2_\triangle, G)$ as defined above. This is essentially the 'gauge invariance' of Wilson loop operators. It follows from the 1-form $\mathsf{Rep}(G)$ symmetry that the Hamiltonian $\mathbb{H}^\lambda$ decomposes into charge sectors labelled by $(g_1, g_2) \in G \times G$ corresponding to holonomies of the flat gauge field $\mathfrak{g}$ along both non-contractible 1-cycles of the two-torus.

   Importantly, we can also construct topological junctions of topological Wilson lines.[6] Consider any number of oriented Wilson line operators meeting at any given vertex $\mathsf{v} \in \mathsf{V}(\mathbb{T}^2_\triangle)$. As long as the Wilson lines are chosen so that the tensor product of their corresponding representations, or their duals depending on the orientations, is isomorphic to the trivial representation, then it is a valid symmetry operator. As a matter of fact, since $G$ is abelian, it simply boils down to a composition of Wilson loop operators. Therefore, any network of topological Wilson lines thus constructed is a valid symmetry operator. More formally, consider the non-degenerate pairing

$$\begin{aligned} \langle -, - \rangle : \widehat{G} \times G &\longrightarrow \mathrm{U}(1) \\ : (\chi, g) &\longmapsto \langle \chi, g \rangle := \chi(g) \end{aligned}\,. \tag{3.10}$$

It enables us to define the following *cup product* [BCH18]:

$$\begin{aligned} \langle -, \smile - \rangle : C^1(\mathbb{T}^2_\triangle, \widehat{G}) \times C^1(\mathbb{T}^2_\triangle, G) &\longrightarrow C^2(\mathbb{T}^2_\triangle, \mathrm{U}(1)) \\ : (\mathfrak{X}, \mathfrak{g}) &\longmapsto \big(\langle \mathfrak{X}, \smile \mathfrak{g} \rangle : (\mathsf{v}_1 \mathsf{v}_2 \mathsf{v}_3) \longmapsto \langle \mathfrak{X}(\mathsf{v}_1 \mathsf{v}_2), \mathfrak{g}(\mathsf{v}_2 \mathsf{v}_3) \rangle\big) \end{aligned}\,. \tag{3.11}$$

Given the differential on the spaces of simplicial $\widehat{G}$-valued forms defined as in eq. (3.1), we can consider groups $Z^n(\mathbb{T}^2_\triangle, \widehat{G})$ and $B^n(\mathbb{T}^2_\triangle, \widehat{G})$. Let $\mathfrak{X} \in Z^1(\mathbb{T}^2_\triangle, \widehat{G})$ and consider the operator

$$\mathfrak{X}(\mathbb{T}^2_\triangle) := \sum_{\mathfrak{g} \in Z^1(\mathbb{T}^2_\triangle, G)} \int_{\mathbb{T}^2_\triangle} \langle \mathfrak{X}, \smile \mathfrak{g} \rangle\, |\mathfrak{g}\rangle\langle\mathfrak{g}| \equiv \sum_{\mathfrak{g} \in Z^1(\mathbb{T}^2_\triangle, G)} \left( \prod_{\mathsf{p} \in \mathsf{P}(\mathbb{T}^2_\triangle)} \langle \mathfrak{X}, \smile \mathfrak{g} \rangle(\mathsf{p}) \right) |\mathfrak{g}\rangle\langle\mathfrak{g}|\,. \tag{3.12}$$

It follows from the definition of the cup product that $\mathfrak{X}(\mathbb{T}^2_\triangle)$ precisely amounts to inserting a certain combination of Wilson loops as defined in eq. (3.8). Let us consider a concrete example in the case where $\mathbb{T}^2_\triangle$ is the triangular lattice. Let $\chi(\ell)$ be a Wilson loop operator with $\chi \in \widehat{G}$ and $\ell \in Z_1(\mathbb{T}^2_\triangle, \mathbb{Z})$. One can always associate to it a 1-cocycle $\mathfrak{X} \in Z^1(\mathbb{T}^2_\triangle, \widehat{G})$ such that the action of $\mathfrak{X}(\mathbb{T}^2_\triangle)$ coincides with

---

[6]This ability to construct junctions of topological lines is precisely why it is more accurate to talk about a 1-form $\mathsf{Rep}(G)$ symmetry rather than a 1-form $\widehat{G}$ symmetry, as a higher mathematical structure is required to encompass both lines and their junctions.

that of $\chi(\ell)$. We depict such a configuration below:[7]

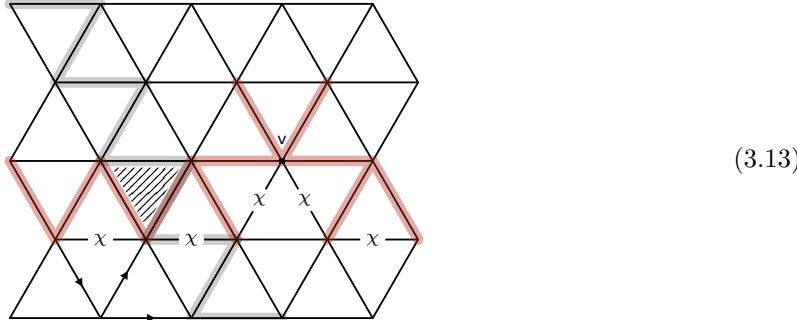

$$(3.13)$$

where the bold red lines represent the only edges to which $\mathfrak{X}$ assigns non-trivial elements in $\widehat{G}$, to wit $\chi$ or $\chi^\vee$ depending on the orientations. We also depicted in bold gray lines a choice of gauge field $\mathfrak{g}$. Finally, the hashed plaquette is the only one that contributes non-trivially to $\langle \mathfrak{X}, \smile \mathfrak{g} \rangle$. It follows from the properties of the cup product that $\langle \mathfrak{X}, \smile \mathfrak{g} \rangle$ yields the same result regardless of the choices of representatives in $[\mathfrak{g}]$ and $[\mathfrak{X}]$, respectively. In particular, picking a different representative in $[\mathfrak{X}]$ simply amounts to performing a continuous deformation of $\ell$.

It was recently pointed out that the 1-form $\mathsf{Rep}(G)$ symmetry is only a component of the total symmetry structure [Del21, BSNW22, BBFP22]. In particular, topological surface operators commuting with the Hamiltonian can also be defined. These are often referred to as *condensation operators* in the literature [RSS22, LRS22]. A lattice implementation of condensation operators was first presented in ref. [DT23], employing the tensor network calculus developed in ref. [Del21]. We shall provide here particularly explicit expressions for these operators in the form of tensor networks. First of all, we define the topological surface operator $\mathbb{U}^{\{\mathbb{1}\}}(\mathbb{T}_\triangle^2)$, which acts on the whole kinematical Hilbert space, by summing over all possible insertions of Wilson loops. Another way to describe this operator, is that we insert a network of Wilson line operators associated with the representation $\bigoplus_{\chi \in \widehat{G}} \chi$, in such a way that every junction implements the tensor product of representations—a process that amounts to gauging the $\mathsf{Rep}(G)$ symmetry along the spacetime submanifold $\mathbb{T}^2$ [RSS22]. The notation $\mathbb{U}^{\{\mathbb{1}\}}(\mathbb{T}_\triangle^2)$ then stems from the fact that the induced representation $\mathrm{Ind}_{\{\mathbb{1}\}}^G(\underline{0}_{\{\mathbb{1}\}})$ of the trivial representation $\underline{0}_{\{\mathbb{1}\}}$ of $\{\mathbb{1}\} \subset G$ in $G$ is isomorphic to the regular representation $\mathbb{C}^G \cong \bigoplus_{\chi \in \widehat{G}} \chi$. Invoking eq. (3.12), we can equivalently express this operator as follows:

$$\mathbb{U}^{\{\mathbb{1}\}}(\mathbb{T}_\triangle^2) := \frac{|Z^0(\mathbb{T}_\triangle^2, \widehat{G})|}{|Z^1(\mathbb{T}_\triangle^2, \widehat{G})|} \sum_{\mathfrak{X} \in Z^1(\mathbb{T}_\triangle^2, \widehat{G})} \mathfrak{X}(\mathbb{T}_\triangle^2) \,. \tag{3.14}$$

But, whenever $\mathfrak{X}$ is cohomologically trivial, $\mathfrak{X}(\mathbb{T}_\triangle^2) = 1$. Decomposing the sum over $Z^1(\mathbb{T}_\triangle^2, \widehat{G})$ into a sum over $H^1(\mathbb{T}_\triangle^2, \widehat{G}) = Z^1(\mathbb{T}_\triangle^2, \widehat{G})/B^1(\mathbb{T}_\triangle^2, \widehat{G}) \cong \widehat{G} \times \widehat{G}$ and a sum over $B^1(\mathbb{T}_\triangle^2, \widehat{G})$ yields the alternative formula

$$\mathbb{U}^{\{\mathbb{1}\}}(\mathbb{T}_\triangle^2) = \frac{1}{|G|} \sum_{[\mathfrak{X}] \in H^1(\mathbb{T}_\triangle^2, \widehat{G})} \mathfrak{X}(\mathbb{T}_\triangle^2) \,, \tag{3.15}$$

---

[7]Notice how deforming the Wilson loop so as to 'straighten' it amounts to multiplying $\mathfrak{X}$ by a 1-coboundary. Specifically, the coboundary is given by $\mathfrak{b}_\mathsf{v}$ wherein $\mathsf{v}$ stands for the vertex highlighted in eq. (3.13) and which is defined as $\mathfrak{b}_\mathsf{v}(\mathsf{v}_1) = \underline{0}$ if $\mathsf{v}_1 \neq \mathsf{v}$, $\underline{0}$ being the trivial representation of $G$, and $\mathfrak{b}_\mathsf{v}(\mathsf{v}) = \chi$.

where we used the fact that $|\widehat{G}| = |G|$, together with $|Z^0(\mathbb{T}_\triangle^2, \widehat{G})| = |\widehat{G}|$. What is the effect of such a condensation operator? It acts as a projector onto the singlet sector of the 1-form $\mathsf{Rep}(G)$-symmetry. Indeed, denoting by $(\gamma_1, \gamma_2)$ a generator of $H_1(\mathbb{T}_\triangle^2, \mathbb{Z}) \cong \mathbb{Z}_2 \times \mathbb{Z}_2$, i.e. two inequivalent non-contractible 1-cycles along the one-skeleton of $\mathbb{T}_\triangle^2$, it follows from $\sum_{\chi \in \widehat{G}} \chi(g) = |G| \, \delta_{g,\mathbb{1}}$, for every $g \in G$, that

$$\mathbb{U}^{\{\mathbb{1}\}}(\mathbb{T}_\triangle^2) = \frac{1}{|G|} \left( \sum_{\chi_1 \in \widehat{G}} \chi_1(\gamma_1) \right) \circ \left( \sum_{\chi_2 \in \widehat{G}} \chi_2(\gamma_2) \right), \tag{3.16}$$

enforces holonomies $\mathrm{hol}_{\gamma_1}(\mathfrak{g})$ and $\mathrm{hol}_{\gamma_2}(\mathfrak{g})$ of the gauge field $\mathfrak{g} \in Z^1(\mathbb{T}_\triangle^2, G)$ along $\gamma_1$ and $\gamma_2$, respectively, to evaluate to the identity element in $G$. In other words, it projects onto the subspace of the kinematical Hilbert space spanned by vectors $|\mathrm{d}^{(0)}\mathfrak{m}\rangle$, where $\mathfrak{m} \in C^0(\mathbb{T}_\triangle^2, G)$. Accordingly, we have $\mathbb{U}^{\{\mathbb{1}\}}(\mathbb{T}_\triangle^2) \circ \chi(\ell) = \chi(\ell) \circ \mathbb{U}^{\{\mathbb{1}\}}(\mathbb{T}_\triangle^2) = \mathbb{U}^{\{\mathbb{1}\}}(\mathbb{T}_\triangle^2)$, for every $\chi \in \widehat{G}$ and $\ell \in Z_1(\mathbb{T}_\triangle^2, \mathbb{Z})$. Moreover, it is immediate from eq. (3.16) that acting twice with $\mathbb{U}^{\{\mathbb{1}\}}(\mathbb{T}_\triangle^2)$ amounts to acting with $|G| \cdot \mathbb{U}^{\{\mathbb{1}\}}(\mathbb{T}_\triangle^2)$.

In a similar vein, for every subgroup $A \subseteq G$, one defines a condensation operator $\mathbb{U}^A(\mathbb{T}_\triangle^2)$ obtained by introducing a network of Wilson line operators associated with the representation $\mathrm{Ind}_A^G(\underline{0}_A) \cong \mathbb{C}^{G/A}$. But, by Frobenius reciprocity, the representations that appear in $\mathrm{Ind}_A^G(\underline{0}_A)$ are precisely the representations in $G$ that are trivial when restricted to $A$, so that the set of such representations is isomorphic to $\widehat{G/A}$.[8] Bringing everything together, one defines

$$\mathbb{U}^A(\mathbb{T}_\triangle^2) := \frac{1}{|G/A|} \left( \sum_{\chi_1 \in \mathrm{Ind}_A^G(\underline{0}_A)} \chi_1(\gamma_1) \right) \circ \left( \sum_{\chi_2 \in \mathrm{Ind}_A^G(\underline{0}_A)} \chi_2(\gamma_2) \right). \tag{3.17}$$

It follows from $\sum_{\chi \in \mathrm{Ind}_A^G(\underline{0}_A)} \chi(g)$ being equal to $|G/A|$ whenever $g \in A$, and zero otherwise,[9] that $\mathbb{U}^A(\mathbb{T}_\triangle^2)$ has the effect of projecting onto charge sectors where holonomies along both $\gamma_1$ and $\gamma_2$ evaluate to group elements in $A$. Moreover, we have

$$\mathbb{U}^A(\mathbb{T}_\triangle^2) \circ \mathbb{U}^A(\mathbb{T}_\triangle^2) = |G/A| \cdot \mathbb{U}^A(\mathbb{T}_\triangle^2). \tag{3.18}$$

Naturally, $\mathbb{U}^A(\mathbb{T}_\triangle^2)$ boils down to $\mathbb{U}^{\{\mathbb{1}\}}(\mathbb{T}_\triangle^2)$ when specialising to $A = \{\mathbb{1}\}$. By construction, Wilson lines labelled by representations in $\mathrm{Ind}_A^G(\underline{0}_A)$ condense on this condensation operator in the sense that $\mathbb{U}^A(\mathbb{T}_\triangle^2) \circ \chi(\ell) = \chi(\ell) \circ \mathbb{U}^A(\mathbb{T}_\triangle^2) = \mathbb{U}^A(\mathbb{T}_\triangle^2)$, for every $\chi \in \mathrm{Ind}_A^G(\underline{0}_A)$ and $\ell \in Z_1(\mathbb{T}_\triangle^2, \mathbb{Z})$, whereas Wilson lines generating the $\mathsf{Rep}(A)$ 1-form subsymmetry survive on it. In other words, even while acting with $\mathbb{U}^A(\mathbb{T}_\triangle^2)$, the model still possesses a faithful $\mathsf{Rep}(A)$ 1-form symmetry. Importantly, these are not the only topological line operators that can equip the topological surface operator $\mathbb{U}^A(\mathbb{T}_\triangle^2)$. Indeed, one can also decorate $\mathbb{U}^A(\mathbb{T}_\triangle^2)$ with networks of *'t Hooft lines* labelled by left cosets in $G/A$ so that the resulting surface operator projects onto charge sectors where holonomies along both $\gamma_1$ and $\gamma_2$ evaluate to any group element in $G$. Topological interfaces between condensation operators $\mathbb{U}^A(\mathbb{T}_\triangle^2)$ associated with different choices of $A \subseteq G$ can be constructed in a similar fashion [Del21, DT23].

Before introducing the remaining surface operators, it is very useful to provide a microscopic realisation of $\mathbb{U}^A(\mathbb{T}_\triangle^2)$, for every $A \subseteq G$, in terms of tensor networks, applying the general framework outlined in ref. [Del21]. First of all, notice that the condensation operator $\mathbb{U}^A(\mathbb{T}_\triangle^2)$ can be equivalently

---

[8]Since $G$ is abelian, every subgroup is normal and thus the quotient $G/A$ is always a group. As a matter of fact, since $G$ is finite abelian, $G$ admits a subgroup that is isomorphic to $G/A$, for any $A \subseteq G$.

[9]Given any subgroup $R \subseteq G$, there exists $A \subseteq G$ and a group isomorphism $\varphi : G/A \xrightarrow{\sim} R$. Let $g \in G$ such that $g \notin A$; it follows from $G = \bigsqcup_{rA \in G/A} rA$ that $g = ra$ with $a \in A$ and $r \neq \mathbb{1}$. Therefore, $\sum_{\chi \in \mathrm{Ind}_A^G(\underline{0}_A)} \chi(g) = \sum_{\chi \in \mathrm{Ind}_A^G(\underline{0}_A)} \chi(r) = \sum_{\chi \in \widehat{R}} \chi(\varphi(rA)) = 0$, as desired.

expressed as follows:

$$\mathbb{U}^A(\mathbb{T}^2_\triangle) = \sum_{\substack{\mathfrak{g} \in Z^1(\mathbb{T}^2_\triangle, G) \\ \mathfrak{m} \in C^0(\mathbb{T}^2_\triangle, G/A)}} \left( \prod_{(\mathsf{v}_1\mathsf{v}_2)} \delta_{\mathfrak{g}(\mathsf{v}_1\mathsf{v}_2) \triangleright \mathfrak{m}(\mathsf{v}_2), \mathfrak{m}(\mathsf{v}_1)} \right) |\mathfrak{g}\rangle\langle\mathfrak{g}| . \tag{3.19}$$

Indeed, since for every $\mathsf{e} \equiv (\mathsf{v}_1\mathsf{v}_2) \in \mathsf{E}(\mathbb{T}^2_\triangle)$, there is a Kronecker delta imposing $\mathfrak{g}(\mathsf{v}_1\mathsf{v}_2) \triangleright \mathfrak{m}(\mathsf{v}_2) = \mathfrak{m}(\mathsf{v}_1)$, it follows from the comments in sec. 2.5 that there exists a unique group element $a_{\mathfrak{g}(\mathsf{v}_1\mathsf{v}_2), \mathfrak{m}(\mathsf{v}_2)} \in A$ such that $\mathfrak{g}(\mathsf{v}_1\mathsf{v}_2) = \mathrm{rep}(\mathfrak{m}(\mathsf{v}_1)) \cdot a_{\mathfrak{g}(\mathsf{v}_1\mathsf{v}_2), \mathfrak{m}(\mathsf{v}_2)} \cdot \mathrm{rep}(\mathfrak{m}(\mathsf{v}_2))^{-1}$, where $\mathrm{rep}: G/A \to G$ assigns to every left coset in $G/A$ its representative in $G$. Moreover, by definition, we have $a_{g_1, g_2 \triangleright rA} \cdot a_{g_2, rA} = a_{g_1 g_2, rA}$, for every $g_1, g_2 \in G$ and $rA \in G/A$. Consequently, defining $\mathfrak{a}_{\mathfrak{g}, \mathfrak{m}} \in C^1(\mathbb{T}^2_\triangle, A)$ via $\mathfrak{a}_{\mathfrak{g}, \mathfrak{m}}(\mathsf{v}_1\mathsf{v}_2) := a_{\mathfrak{g}(\mathsf{v}_1\mathsf{v}_2), \mathfrak{m}(\mathsf{v}_2)}$, for every $(\mathsf{v}_1\mathsf{v}_2) \in \mathsf{E}(\mathbb{T}^2_\triangle)$ such that $\mathfrak{g}(\mathsf{v}_1\mathsf{v}_2) \triangleright \mathfrak{m}(\mathsf{v}_2) = \mathfrak{m}(\mathsf{v}_1)$, one finds $\mathfrak{a}_{\mathfrak{g}, \mathfrak{m}}(\mathsf{v}_1\mathsf{v}_2) \mathfrak{a}_{\mathfrak{g}, \mathfrak{m}}(\mathsf{v}_2\mathsf{v}_3) = \mathfrak{a}_{\mathfrak{g}, \mathfrak{m}}(\mathsf{v}_1\mathsf{v}_3)$, for every $(\mathsf{v}_1\mathsf{v}_2\mathsf{v}_3) \in \mathsf{P}(\mathbb{T}^2_\triangle)$, and thus $\mathfrak{a}_{\mathfrak{g}, \mathfrak{m}} \in Z^1(\mathbb{T}^2_\triangle, A)$. This confirms that $\mathbb{U}^A(\mathbb{T}^2_\triangle)$ acts as a projector onto the subspace spanned by states $|\mathfrak{a}\, \mathrm{d}^{(0)}\mathrm{rep}(\mathfrak{m})\rangle$, where $\mathfrak{a} \in Z^1(\mathbb{T}^2_\triangle, A)$ and $\mathfrak{m} \in C^0(\mathbb{T}^2_\triangle, G/A)$. Furthermore, we recover the fact that $\mathbb{U}^A(\mathbb{T}^2_\triangle) \circ \mathbb{U}^A(\mathbb{T}^2_\triangle) = |G/A| \cdot \mathbb{U}^A(\mathbb{T}^2_\triangle)$, as following from the fact that the Cartesian product $G/A \times G/A$ decomposes into a disjoint union of $|G/A|$ copies of $G/A$ under the diagonal action of $G$.

Invoking (3.19), one can explain how to construct the surface operator $\mathbb{U}^A(\mathbb{T}^2_\triangle)[\mathfrak{f}]$ that is decorated by a network $\mathfrak{f}$ of 't Hooft lines. Let us consider a concrete example. Let $\ell$ be an oriented 1-cycle along the Poincaré dual $\mathbb{T}^{2\,\vee}_\triangle$ of $\mathbb{T}^2_\triangle$. Consider the 't Hooft loop labelled by $fA \in G/A$ with support $\ell$. One defines the 1-cocycle $\mathfrak{f} \in Z^1(\mathbb{T}^2_\triangle, G/A)$ associated with this 't Hooft loop as follows:

$$\mathfrak{f}(\mathsf{e}) = \begin{cases} fA & \text{if } \iota(\mathsf{e}) \subset \ell \\ f^{-1}A & \text{if } \iota(-\mathsf{e}) \subset \ell \\ \mathbb{1} & \text{otherwise} \end{cases} , \tag{3.20}$$

for every oriented edge $\mathsf{e} \in \mathsf{E}(\mathbb{T}^2_\triangle)$, where $\iota: \mathsf{E}(\mathbb{T}^2_\triangle) \xrightarrow{\sim} \mathsf{E}(\mathbb{T}^{2\,\vee}_\triangle)$. When defining the corresponding surface operator $\mathbb{U}^A(\mathbb{T}^2_\triangle)[\mathfrak{f}]$, we replace the Kronecker deltas entering the definition (3.19) by $\delta_{\mathfrak{g}(\mathsf{v}_1\mathsf{v}_2) \triangleright \mathfrak{m}(\mathsf{v}_2), \mathfrak{f}(\mathsf{v}_1\mathsf{v}_2)\mathfrak{m}(\mathsf{v}_1)}$ for every $(\mathsf{v}_1\mathsf{v}_2) \in \mathsf{E}(\mathbb{T}^2_\triangle)$. Supposing that $\ell \equiv \gamma_1$ is one of the generators of $H_1(\mathbb{T}^{2\,\vee}_\triangle, \mathbb{Z})$, we depict such a configuration on the triangular lattice below

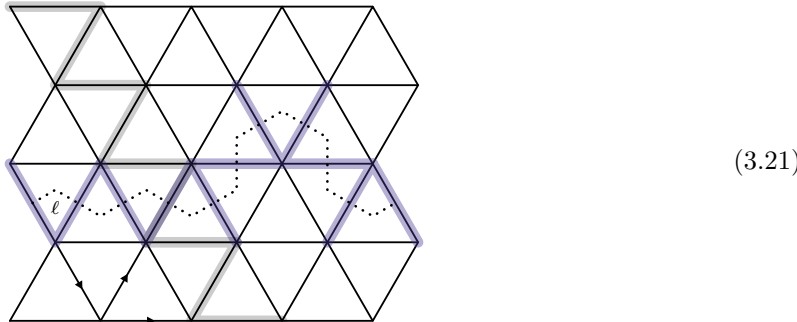

(3.21)

where the bold purple lines represent the only edges to which $\mathfrak{f}$ assigns non-trivial elements in $G/A$. There is a choice of $\mathfrak{m} \in C^0(\mathbb{T}^2_\triangle, G/A)$ such that a gauge field satisfying $\mathfrak{g}(\mathsf{v}_1\mathsf{v}_2) \triangleright \mathfrak{m}(\mathsf{v}_2) = \mathfrak{f}(\mathsf{v}_1\mathsf{v}_2)\mathfrak{m}(\mathsf{v}_1)$, at every edge $(\mathsf{v}_1\mathsf{v}_2)$, is non-vanishing along the bold gray lines only. It follows from the various definitions that $\mathrm{hol}_{\gamma_2}(\mathfrak{g}) \in fA$, for any continuous deformation of the 't Hooft loop.

Still specialising to the case where $\mathbb{T}^2_\triangle$ is the triangular lattice, let us invoke eq. (3.19) to express $\mathbb{U}^A(\mathbb{T}^2_\triangle)$ as a tensor network. One begins by introducing a rank-$(p+q)$ tensor in $\mathbb{C}[G/A]^{\otimes p} \to \mathbb{C}[G/A]^{\otimes q}$ of the form $\sum_{\{r_i A\}_i} \left( \prod_{i=1}^{p+q-1} \delta_{r_i A, r_{i+1} A} \right) \otimes_{i=p+1}^{p+q} |r_i A\rangle \otimes_{i=1}^p \langle r_i A|$, graphically depicted as

$$\equiv \sum_{\{r_i A\}_{i=1}^{p+q}} \bigotimes_{i=p+1}^{p+q} |r_i A\rangle \bigotimes_{i=1}^{p} \langle r_i A| \tag{3.22}$$

$$\equiv \sum_{\{r_i A\}_{i=1}^{p+q}} \left( \prod_{i=1}^{p+q-1} \delta_{r_i A, r_{i+1} A} \right) \bigotimes_{i=p+1}^{p+q} |r_i A\rangle \bigotimes_{i=1}^{p} \langle r_i A| \,,$$

where the sums are over left cosets in $G/A$. Henceforth, we refer to such a tensor as a 'Kronecker delta tensor'. We also require a tensor implementing the $G$-action on $G/A$:

$$\equiv \sum_{\substack{g \in G \\ r_1 A, r_2 A \in G/A}} |g\rangle\langle g| \otimes |g\rangle\langle g| \otimes |r_1 A\rangle\langle r_2 A| \tag{3.23}$$

$$\equiv \sum_{\substack{g \in G \\ rA \in G/A}} |g\rangle\langle g| \otimes |g\rangle\langle g| \otimes |g \triangleright rA\rangle\langle rA| \,,$$

as well as a tensor implementing the flatness of the gauge field:

$$\equiv \sum_{g_1, g_2, g_3 \in G} |g_1, g_2\rangle\langle g_3| \equiv \sum_{g_1, g_2 \in G} |g_1, g_2\rangle\langle g_1 g_2| \,, \tag{3.24}$$

$$\equiv \sum_{g_1, g_2, g_3 \in G} |g_3\rangle\langle g_1, g_2| \equiv \sum_{g_1, g_2 \in G} |g_1 g_2\rangle\langle g_1, g_2| \,.$$

Bringing everything together, we can realise $\mathbb{U}^A(\mathbb{T}_\triangle^2)$ as a tensor network whose unit cell is given by

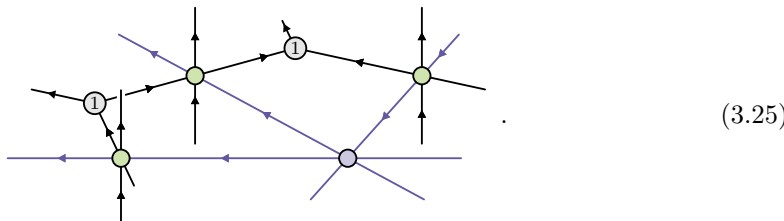

$$. \tag{3.25}$$

Let us dissect this tensor network. First of all, to every vertex of the triangular lattice, one assigns such a unit cell, in such a way that the Kronecker delta tensor coincides with the vertex, and the purple lines with the edges of the lattice. Comparing with eq. (3.19), the Kronecker delta tensors perform the summation over $C^0(\mathbb{T}_\triangle^2, G/A)$, tensors (3.23) impose the constraints encoded into the Kronecker delta tensors, whereas the remaining tensors enforce the flatness of the gauge field. Notice that when

realising $\mathbb{U}^{\{\mathbb{1}\}}(\mathbb{T}^2_\triangle)$, it follows from $\mathrm{d}^{(1)}(\mathrm{d}^{(0)}\mathfrak{m}) = 0$, for every $\mathfrak{m} \in C^0(\mathbb{T}^2_\triangle, G)$, that the tensors imposing the flatness of the gauge field are superfluous and can thus be omitted.

It turns out that the model with Hamiltonian $\mathbb{H}^\lambda$ hosts as many topological surface operators as there are (indecomposable) module categories over the category $\mathsf{Vec}_G$. We reviewed in sec. 2.5 that indecomposable $\mathsf{Vec}_G$-module categories are labelled by pairs $(A, \psi)$ consisting of a subgroup $A \subseteq G$ and a normalised representative $\psi$ of a cohomology class $[\psi] \in H^2(A, \mathrm{U}(1))$. Whenever the 2-cocycle $\psi$ is trival, the topological surface operator associated with the $\mathsf{Vec}_G$-module category $\mathcal{M}(A, 1)$ is the condendation operator $\mathbb{U}^A(\mathbb{T}^2_\triangle)$, as defined above [Del21]. A non-trivial 2-cocycle $\psi$ modifies it in the following way:

$$\mathbb{U}^{A_\psi}(\mathbb{T}^2_\triangle) := \sum_{\substack{\mathfrak{g} \in Z^1(\mathbb{T}^2_\triangle, G) \\ \mathfrak{m} \in C^0(\mathbb{T}^2_\triangle, G/A)}} \left( \prod_{(\mathsf{v}_1\mathsf{v}_2)} \delta_{\mathfrak{g}(\mathsf{v}_1\mathsf{v}_2)\triangleright\mathfrak{m}(\mathsf{v}_2), \mathfrak{m}(\mathsf{v}_1)} \prod_{(\mathsf{v}_1\mathsf{v}_2\mathsf{v}_3)} \psi\big(\mathfrak{a}_{\mathfrak{g},\mathfrak{m}}(\mathsf{v}_1\mathsf{v}_2), \mathfrak{a}_{\mathfrak{g},\mathfrak{m}}(\mathsf{v}_2\mathsf{v}_3)\big)^{\epsilon(\mathsf{v}_1\mathsf{v}_2\mathsf{v}_3)} \right) |\mathfrak{g}\rangle\langle\mathfrak{g}| ,$$

$$(3.26)$$

where we recognise the 2-cochain $\alpha^\triangleright_\psi \in C^2(G, \mathrm{Fun}(G/A, \mathrm{U}(1)))$ defined in eq. (2.43) the module associator of $\mathcal{M}(A, \psi)$ evaluates to. The commutation relation $[\mathbb{H}^\lambda, \mathbb{U}^{A_\psi}(\mathbb{T}^2_\triangle)] = 0$ follows from the 2-cocycle condition satisfied by $\psi$. Indeed, letting $\mathsf{v} \in \mathsf{V}(\mathbb{T}^2_\triangle)$ and $\mathfrak{x}_\mathsf{v} \in C^0(\mathbb{T}^2_\triangle, G)$ such that $\mathfrak{x}_\mathsf{v}(\mathsf{v}_1) = \mathbb{1}$, whenever $\mathsf{v}_1 \neq \mathsf{v}$, the 2-cocycle condition ensures that

$$\prod_{(\mathsf{v}_1\mathsf{v}_2\mathsf{v}_3)} \psi\big(\mathfrak{a}_{\mathfrak{g}\,\mathrm{d}^{(0)}\mathfrak{x}_\mathsf{v}, \mathfrak{x}_\mathsf{v}\triangleright\mathfrak{m}}(\mathsf{v}_1\mathsf{v}_2), \mathfrak{a}_{\mathfrak{g}\,\mathrm{d}^{(0)}\mathfrak{x}_\mathsf{v}, \mathfrak{x}_\mathsf{v}\triangleright\mathfrak{m}}(\mathsf{v}_2\mathsf{v}_3)\big)^{\epsilon(\mathsf{v}_1\mathsf{v}_2\mathsf{v}_3)} = \prod_{(\mathsf{v}_1\mathsf{v}_2\mathsf{v}_3)} \psi\big(\mathfrak{a}_{\mathfrak{g},\mathfrak{m}}(\mathsf{v}_1\mathsf{v}_2), \mathfrak{a}_{\mathfrak{g},\mathfrak{m}}(\mathsf{v}_2\mathsf{v}_3)\big)^{\epsilon(\mathsf{v}_1\mathsf{v}_2\mathsf{v}_3)} ,$$

$$(3.27)$$

for every $\mathfrak{g} \in Z^1(\mathbb{T}^2_\triangle, G)$ and $\mathfrak{m} \in C^0(\mathbb{T}^2_\triangle, G/A)$ such that $\mathfrak{g}(\mathsf{v}_1\mathsf{v}_2) \triangleright \mathfrak{m}(\mathsf{v}_2) = \mathfrak{m}(\mathsf{v}_1)$ at every edge $(\mathsf{v}_1\mathsf{v}_2) \in \mathsf{E}(\mathbb{T}^2_\triangle)$, and where $(\mathfrak{x}_\mathsf{v} \triangleright \mathfrak{m})(\mathsf{v}_1) := \mathfrak{x}_\mathsf{v}(\mathsf{v}_1) \triangleright \mathfrak{m}(\mathsf{v}_1)$ for every $\mathsf{v}_1 \in \mathsf{V}(\mathbb{T}^2_\triangle)$. Introducing the tensors

$$(3.28)$$

and

$$(3.29)$$

we can realise $\mathbb{U}^{A_\psi}(\mathbb{T}_\triangle^2)$ on the triangular lattice as a tensor network whose unit cell is given by[10]

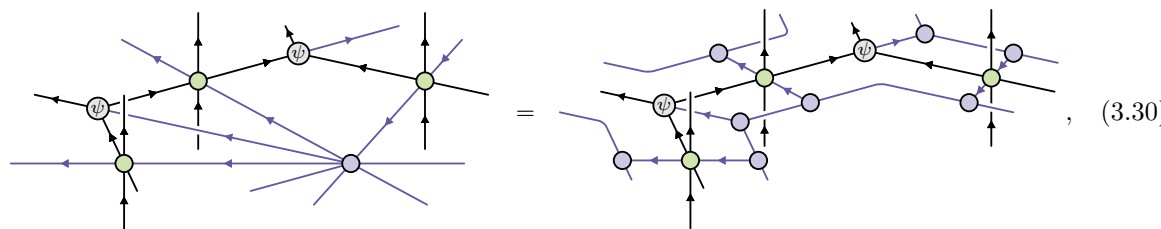

where, in order to obtain the r.h.s., we used the fact that contracting Kronecker delta tensors results in another Kronecker delta tensor. As a special case, the topological surface operator associated with $\mathcal{M}(G,\psi)$ reads

$$
\begin{aligned}
\mathbb{U}^{G_\psi}(\mathbb{T}_\triangle^2) &= \sum_{\mathfrak{g}\in Z^1(\mathbb{T}_\triangle^2,G)} \left( \prod_{(\mathsf{v}_1\mathsf{v}_2\mathsf{v}_3)} \psi\big(\mathfrak{g}(\mathsf{v}_1\mathsf{v}_2),\mathfrak{g}(\mathsf{v}_2\mathsf{v}_3)\big)^{\epsilon(\mathsf{v}_1\mathsf{v}_2\mathsf{v}_3)} \right) |\mathfrak{g}\rangle\langle\mathfrak{g}| \\
&= \sum_{\mathfrak{g}\in Z^1(\mathbb{T}_\triangle^2,G)} \frac{\psi\big(\mathrm{hol}_{\gamma_1}(\mathfrak{g}),\mathrm{hol}_{\gamma_2}(\mathfrak{g})\big)}{\psi\big(\mathrm{hol}_{\gamma_2}(\mathfrak{g}),\mathrm{hol}_{\gamma_1}(\mathfrak{g})\big)} |\mathfrak{g}\rangle\langle\mathfrak{g}| ,
\end{aligned}
$$

(3.31)

where we invoked the 2-cocycle condition of $\psi$ to go from the first line to the second line.

We commented earlier that the condensation operator (3.19) could be obtained by summing over networks of Wilson lines valued in a subgroup of $\widehat{G}$ that is isomorphic to $\widehat{G/A}$, which should be thought as topological lines decorating the identity surface operator $\mathbb{U}^G(\mathbb{T}_\triangle^2)$. Conversely, the same operator $\mathbb{U}^A(\mathbb{T}_\triangle^2)$ can be obtained, starting from $\mathbb{U}^{\{\mathbb{1}\}}(\mathbb{T}_\triangle^2)$, by summing over all possible networks of topological of lines labelled by group elements in $A \subseteq G$, in such a way that the group multiplication in $A$—or rather the multiplication rule of the group algebra $\mathbb{C}[A]$—is implemented at every junction. In this formulation, the operator $\mathbb{U}^{G_\psi}(\mathbb{T}_\triangle^2)$ is obtained starting from $\mathbb{U}^{\{\mathbb{1}\}}(\mathbb{T}_\triangle^2)$ by summing over all possible networks of topological lines in $G$ such that the multiplication rule of the twisted group algebra $\mathbb{C}[G]^\psi$ is implemented at every junction.

We have established that for every $\mathsf{Vec}_G$-module category, one can construct a topological surface operator. Moreover, these topological surfaces can be decorated with networks of topological lines. In particular, the surface operator $\mathbb{U}^{A_\psi}(\mathbb{T}_\triangle^2)$ associated with the $\mathsf{Vec}_G$-module category $\mathcal{M}(A,\psi)$ can be decorated with lines labelled by simple objects in the category $(\mathsf{Vec}_G)^\vee_{\mathcal{M}(A,\psi)}$ defined in sec. 2.5, which happens to be equivalent to $\mathsf{Rep}(A)^{\boxplus|G/A|}$ since the group $G$ is abelian, as expected. Bringing everything together, this indicates that the symmetry structure of Hamiltonians $\mathbb{H}^\lambda$ is provided by the *fusion 2-category* $\mathsf{Mod}(\mathsf{Vec}_G)$ of $\mathsf{Vec}_G$-module categories and $\mathsf{Vec}_G$-module functors. By analogy with $\mathsf{Rep}(G) \simeq \mathsf{Mod}(\mathbb{C}[G])$, it is enlightening to write $2\mathsf{Rep}(G) \equiv \mathsf{Mod}(\mathsf{Vec}_G)$, where $\mathsf{Vec}_G$ is thought as a categorification of $\mathbb{C}[G]$ obtained by promoting the ring $\mathbb{C}$ to the fusion (1-)category $\mathsf{Vec}$, and refer to objects in $2\mathsf{Rep}(G)$ as *2-representations* of $G$.

### 3.3  Symmetry twists

As mentioned earlier, the $2\mathsf{Rep}(G)$-symmetric Hamiltonians $\mathbb{H}^\lambda = \sum_\mathsf{v} \sum_n \mathbb{h}^\lambda_{\mathsf{v},n}$ defined in terms of local operators (3.3) assume periodic boundary conditions. Let us now consider symmetry-twisted closed

---

[10]Notice that we can unambiguously omit certain arrows for visual convenience.

boundary conditions obtained by inserting symmetry twists at various loci of the spatial manifold. The same way we distinguish two types of symmetry operators, namely topological surfaces and topological lines, we distinguish two types of symmetry twists. These are localised along specific spatial 1-cycles and 0-cycles, respectively, while extending in the time direction. Henceforth, we refer to the resulting boundary conditions as being 0-form symmetry-twisted or 1-form symmetry-twisted, respectively. When implementing such symmetry-twisted boundary conditions, the kinematical Hilbert space typically requires to be supplemented with appropriate degrees of freedom, but not always. In any case, the resulting symmetry-twisted boundary conditions preserve the translation invariance of the system, in the sense that there is a unitary isomorphism between Hilbert spaces associated with symmetry twists that are related by continuous deformations. Generally, the presence of 0-form and 1-form symmetry-twisted boundary conditions can be witnessed by the corresponding point-like and line-like charges by moving them along 1- and 2-cycles, respectively. Throughout this section, the focus will be on 1-form symmetry-twisted boundary conditions, since the 0-form ones can be deduced from them, the same way the topological surface operators descend from the topological line operators.

Starting from the kinematical Hilbert space $\mathrm{Span}_{\mathbb{C}}\{|\mathfrak{g}\rangle \,|\, \mathfrak{g} \in Z^1(\mathbb{T}^2_\triangle, G)\}$, consider inserting at a vertex $\mathsf{v}_0 \in \mathsf{V}(\mathbb{T}^2_\triangle)$ the topological line defect labelled by an irreducible representation $\eta : G \to \mathrm{U}(1)$. Concretely, we perform this operation by considering the tensor product between the kinematical Hilbert space and $\mathbb{C}[\widehat{G}]$ before enforcing the degree of freedom in $\mathbb{C}[\widehat{G}]$ to be $|\eta\rangle$. We denote the resulting kinematical space by $\mathcal{H}^\eta$, which is still isomorphic to $\mathrm{Span}_{\mathbb{C}}\{|\mathfrak{g}\rangle \,|\, \mathfrak{g} \in Z^1(\mathbb{T}^2_\triangle, G)\}$. Local symmetric operators acting at $\mathsf{v}_0$ are modified in the following way:

$$\mathbb{h}^{\lambda,\eta}_{\mathsf{v}_0,n} := \sum_{\mathfrak{g}\in Z^1(\mathbb{T}^2_\triangle,G)} \sum_{\mathfrak{r}_{\mathsf{v}_0}} h_n(\mathfrak{g},\mathfrak{r}_{\mathsf{v}_0})\,\eta(\mathfrak{r}_{\mathsf{v}_0})\,\lambda(\mathfrak{g},\mathfrak{r}_{\mathsf{v}_0})\,|\mathfrak{g}\,\mathrm{d}^{(0)}\mathfrak{r}_{\mathsf{v}_0}\rangle\langle\mathfrak{g}|\,, \tag{3.32}$$

where $\eta(\mathfrak{r}_{\mathsf{v}_0}) := \eta\big(\mathfrak{r}_{\mathsf{v}_0}(\mathsf{v}_0)\big)$. All the other local operators are left intact so that the resulting local Hamiltonians read

$$\mathbb{H}^{\lambda,\eta} = \sum_{\substack{\mathsf{v}\in \mathsf{V}(\mathbb{T}^2_\triangle)\\ \mathsf{v}\neq\mathsf{v}_0}} \sum_n \mathbb{h}^{\lambda}_{\mathsf{v},n} + \sum_n \mathbb{h}^{\lambda,\eta}_{\mathsf{v}_0,n}\,. \tag{3.33}$$

Since the group $G$ is abelian, the presence of the symmetry twist does not alter the symmetry structure of the Hamiltonian $\mathbb{H}^{\lambda,\eta}$, in the sense that symmetry operators are still organised in $2\mathsf{Rep}(G)$. Nonetheless, they need to be adapted so as to accommodate the presence of the symmetry twist. This is very simple in the case of topological surface operators $\mathbb{U}^{G_\psi}_\eta(\mathbb{T}^2_\triangle) : \mathcal{H}^\eta \to \mathcal{H}^\eta$ as it simply requires taking the tensor product of the vanilla operator (3.31) with $|\eta\rangle\langle\eta|$. However, specialising to the case where $\mathbb{T}^2_\triangle$ is the triangular lattice, the topological surface operator $\mathbb{U}^{\{\mathbb{1}\}}_\eta(\mathbb{T}^2_\triangle) : \mathcal{H}^\eta \to \mathcal{H}^\eta$ is obtained by replacing the unit cell of its tensor network parametrisation located at $\mathsf{v}_0$ by the following one:

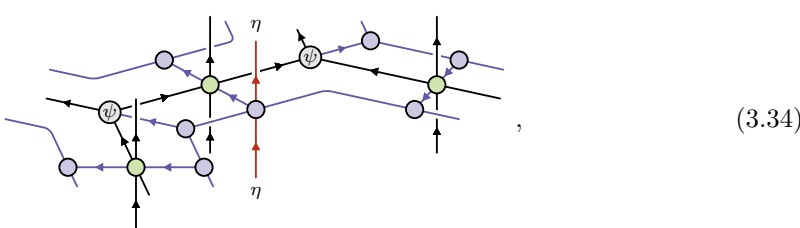

$$\tag{3.34}$$

where we introduced the tensor

$$
\begin{aligned}
&\equiv \sum_{\{g_i\}_{i=1}^{p+q}} \bigotimes_{i=p+1}^{p+q} |g_i\rangle \bigotimes_{i=1}^{p} \langle g_i| \\
&\equiv \sum_{\{g_i\}_{i=1}^{p+q}} \left( \prod_{i=1}^{p+q-1} \delta_{g_i, g_{i+1}} \right) \eta_1(g_1)\, \eta_2^\vee(g_1) \bigotimes_{i=p+1}^{p+q} |g_i\rangle \bigotimes_{i=1}^{p} \langle g_i| ,
\end{aligned}
\tag{3.35}
$$

for every $\eta_1, \eta_2 \in \widehat{G}$. More generally, for an arbitrary triangulation of $\mathbb{T}_\triangle^2$, it suffices to replace the tensor of the form (3.22) located at $\mathsf{v}_0$ by the corresponding one of the form (3.35). Importantly, tensors (3.35) satisfy the following invariance property:

$$
\quad = \quad \text{with} \quad
\begin{aligned}
&\boxed{x} \equiv \sum_{\eta \in \widehat{G}} \boxed{x}\, |\eta\rangle\!\langle\eta| \equiv \sum_{\eta \in \widehat{G}} \eta(x)\, |\eta\rangle\!\langle\eta| , \\[4pt]
&\boxed{x} \equiv \sum_{g \in G} |g\rangle\!\langle gx| ,
\end{aligned}
\tag{3.36}
$$

for every $x \in G$. Together with (3.27), this symmetry condition guarantees that the symmetry operator $\mathbb{U}^{\{1\}}(\mathbb{T}_\triangle^2)$ commutes with local operators $\mathbb{h}_{\mathsf{v}_0,n}^{\lambda,\eta}$ acting at the locus of the 1-form symmetry twist. The remaining symmetry operators can be treated in a similar fashion but the tensors (3.35) take a more complicated form. For any choice of 1-form symmetry-twisted boundary condition $\eta \in \widehat{G}$, it follows from the $2\mathsf{Rep}(G)$ symmetry—more specifically, the 1-form $\mathsf{Rep}(G)$ symmetry—that Hamiltonians $\mathbb{H}^{\lambda,\eta}$ still decompose into charge sectors labelled by $(g_1, g_2) \in G \times G$. Triples $(g_1, g_2, \eta) \in G \times G \times \widehat{G}$ label the *topological sectors* of the theory.

We mentioned above that we could also consider 0-form symmetry-twisted boundary conditions. For instance, consider inserting along a non-contractible cycle $\gamma_1 \in Z_1(\mathbb{T}_\triangle^2, \mathbb{Z})$ the topological surface defect labelled by simple 2-representations of the form $\mathcal{M}(G, \psi) \simeq \mathsf{Vec}^\psi \in 2\mathsf{Rep}(G)$, whose corresponding symmetry operator $\mathbb{U}^{G_\psi}(\mathbb{T}_\triangle^2)$ was defined in eq. (3.31). Similarly to the 1-form symmetry-twisted boundary conditions, this specific type of topological surface defect results in boundary conditions that do not alter the kinematical Hilbert space of the system. Local symmetric operators acting at vertices $\mathsf{v} \subset \gamma_1$ are modified in the following way:

$$
\mathbb{h}_{\mathsf{v},n}^{\lambda,\psi} := \sum_{\mathfrak{g} \in Z^1(\mathbb{T}_\triangle^2, G)} \sum_{\mathfrak{x}_\mathsf{v}} h_n(\mathfrak{g}, \mathfrak{x}_\mathsf{v})\, \psi(\mathfrak{g}, \mathfrak{x}_\mathsf{v})\, \lambda(\mathfrak{g}, \mathfrak{x}_\mathsf{v})\, |\mathfrak{g}\, \mathrm{d}^{(0)}\mathfrak{x}_\mathsf{v}\rangle\!\langle\mathfrak{g}| ,
\tag{3.37}
$$

where

$$
\psi(\mathfrak{g}, \mathfrak{x}_\mathsf{v}) := \prod_{(\mathsf{v}_1 \mathsf{v}) \subset \gamma_1} \psi\big(\mathfrak{g}(\mathsf{v}_1 \mathsf{v}) x^{-1}, x\big) \prod_{(\mathsf{v} \mathsf{v}_2) \subset \gamma_1} \psi\big(x, \mathfrak{g}(\mathsf{v} \mathsf{v}_2)\big)^{-1}, \quad \mathfrak{x}_\mathsf{v}(\mathsf{v}) \equiv x .
\tag{3.38}
$$

All the other local operators are left intact so that the resulting local Hamiltonians read

$$\mathbb{H}^{\lambda,\psi} = \sum_{\substack{\mathsf{v}\in\mathsf{V}(\mathbb{T}^2_\triangle) \\ \mathsf{v}\not\subset\gamma_1}} \sum_n \mathbb{h}^\lambda_{\mathsf{v},n} + \sum_{\substack{\mathsf{v}\in\mathsf{V}(\mathbb{T}^2_\triangle) \\ \mathsf{v}\subset\gamma_1}} \sum_n \mathbb{h}^{\lambda,\psi}_{\mathsf{v},n}\,. \tag{3.39}$$

The remaining symmetry twists can be implemented in a similar fashion.

## 3.4 Duality operators

Given a finite abelian group $G$, we constructed in sec. 3.1 families of two-dimensional quantum lattice models with $\mathsf{2Rep}(G)$ symmetry, which are parametrised by normalised representatives $\lambda$ of a cohomology class $[\lambda]$ in $H^3(G, \mathrm{U}(1))$ as well as sets of complex coefficients $\{h_n\}_n$. We claim that models constructed in this way that only differ in the choice of cohomology class $[\lambda]$ are dual to one another. For now, let us assume periodic boundary conditions and construct the operator transmuting Hamiltonians $\mathbb{H}$ and $\mathbb{H}^\lambda$ into each other. Postponing more formal justifications to sec. 3.7, let us first motivate the algebraic structure encoding such duality operators by analogy with the one-dimensional case. In $(1+1)$d, we showed that Hamiltonians of $\mathsf{Rep}(G)$-symmetric models only differing in a choice of 2-cocycle $\psi$ could be transmuted into each other by any duality operator encoded into the (1-)category $\mathsf{Rep}^\psi(G) \simeq \mathsf{Mod}(\mathbb{C}[G]^\psi)$ of $\psi$-projective representations of $G$. Thus we naively expect Hamiltonians of $\mathsf{2Rep}(G)$-symmetric models only differing in a choice of 3-cocycle $\lambda$ to be transmutable into each other by any duality operator encoded into the 2-category $\mathsf{2Rep}^\lambda(G)$ of '$\lambda$-projective 2-representations of $G$'. Consider the fusion category $\mathsf{Vec}_G$ of $G$-graded vector spaces, whose associator isomorphism, we recall, is given by the identity. Now, considering instead the associator

$$\lambda(g_1, g_2, g_3) \cdot 1_{\mathbb{C}_{g_1 g_2 g_3}} : (\mathbb{C}_{g_1} \otimes \mathbb{C}_{g_2}) \otimes \mathbb{C}_{g_3} \xrightarrow{\sim} \mathbb{C}_{g_1} \otimes (\mathbb{C}_{g_2} \otimes \mathbb{C}_{g_3})\,, \tag{3.40}$$

the resulting fusion category is denoted by $\mathsf{Vec}_G^\lambda$. We then define the fusion 2-category $\mathsf{2Rep}^\lambda(G)$ as the 2-category $\mathsf{Mod}(\mathsf{Vec}_G^\lambda)$ of $\mathsf{Vec}_G^\lambda$-module categories. Indecomposable module categories over $\mathsf{Vec}_G^\lambda$ are defined similarly to module categories over $\mathsf{Vec}_G$ with the following difference: Only subgroups $A$ such that $\lambda_{|A\times A\times A}$ is cohomologically trivial are admissible, and the module associator (2.42) now evaluates to a 2-cochain $\alpha_\phi^\triangleright \in C^2(G, \mathrm{Fun}(G/A, \mathrm{U}(1)))$, associated with a choice of 2-cochain $\phi \in C^2(A, \mathrm{U}(1))$ such that $\mathrm{d}^{(2)}\phi = \lambda^{-1}|_{A\times A\times A}$,[11] that satisfy

$$\mathrm{d}^{(2)}\alpha_\phi^\triangleright(g_1, g_2, g_3)(rA) := \frac{\alpha_\phi^\triangleright(g_2, g_3)(rA)\, \alpha_\phi^\triangleright(g_1, g_2 g_3)(rA)}{\alpha_\phi^\triangleright(g_1 g_2, g_3)(rA)\, \alpha_\phi^\triangleright(g_1, g_2)(g_3 \triangleright rA)} = \lambda(g_1, g_2, g_3)^{-1}\,, \tag{3.41}$$

for every $g_1, g_2, g_3 \in G$ and $rA \in G/A$. The condition (3.41) guarantees the pentagon axiom, which now involves the monoidal associator (3.40). We denote the resulting $\mathsf{Vec}_G^\lambda$-module category by $\mathcal{M}(A, \phi)$.

---

[11]Since we do not require it, we shall omit to write any explicit relation between the 2-cochains $\phi \in C^2(A, \mathrm{U}(1))$ and $\alpha_\phi^\triangleright \in C^2(G, \mathrm{Fun}(G/A, \mathrm{U}(1)))$. Nonetheless, it is important to keep in mind that even if $\phi$ is trivial, the corresponding 2-cochain $\alpha_\phi^\triangleright$ may not be. For instance, whenever $A = \{\mathbb{1}\}$, there is no choice of $\phi$, and yet we have $\alpha_{\mathbb{1}}^\triangleright(-,-)(-) = \lambda(-,-,-)$.

Given the data of a $\mathsf{Vec}_G^\lambda$-module category $\mathcal{M}(A,\phi)$, consider the tensors

$$\sum_{\substack{g_1,g_2,g_3\in G\\ rA\in G/A}} \;\bigg|\; |g_1,g_2\rangle\langle g_3|\otimes\langle rA| \equiv \sum_{\substack{g_1,g_2\in G\\ rA\in G/A}} \alpha_\phi^\rhd(g_1,g_2)(rA)\,|g_1,g_2\rangle\langle g_1 g_2|\otimes\langle rA|$$

$$\sum_{\substack{g_1,g_2,g_3\in G\\ rA\in G/A}} \;\bigg|\; |g_3\rangle\langle g_1,g_2|\otimes|rA\rangle \equiv \sum_{\substack{g_1,g_2\in G\\ rA\in G/A}} \alpha_\phi^\rhd(g_1,g_2)(rA)^{-1}\,|g_1 g_2\rangle\langle g_1,g_2|\otimes|rA\rangle,$$

and denote by $\mathbb{D}^{A_\phi}(\mathbb{T}_\triangle^2)$ the tensor network operator whose unit cell on the triangular lattice with periodic boundary conditions is given by

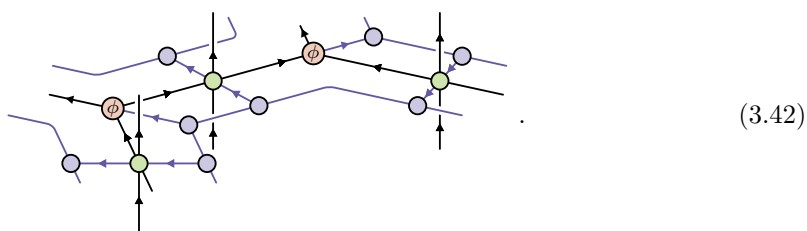

$$(3.42)$$

We claim that the operator $\mathbb{D}^{A_\phi}(\mathbb{T}_\triangle^2)$ transmutes $\mathbb{H}$ into $\mathbb{H}^\lambda$, i.e., $\mathbb{D}^{A_\phi}(\mathbb{T}_\triangle^2)\circ\mathbb{H}=\mathbb{H}^\lambda\circ\mathbb{D}^{A_\phi}(\mathbb{T}_\triangle^2)$. This computation is best carried out invoking the explicit expression

$$\mathbb{D}^{A_\phi}(\mathbb{T}_\triangle^2):=\sum_{\substack{\mathfrak{g}\in Z^1(\mathbb{T}_\triangle^2,G)\\ \mathfrak{m}\in C^0(\mathbb{T}_\triangle^2,G/A)}}\left(\prod_{(\mathsf{v}_1\mathsf{v}_2)}\delta_{\mathfrak{g}(\mathsf{v}_1\mathsf{v}_2)\rhd\mathfrak{m}(\mathsf{v}_2),\mathfrak{m}(\mathsf{v}_1)}\prod_{(\mathsf{v}_1\mathsf{v}_2\mathsf{v}_3)}\alpha_\phi^\rhd\big(\mathfrak{g}(\mathsf{v}_1\mathsf{v}_2),\mathfrak{g}(\mathsf{v}_2\mathsf{v}_3)\big)\big(\mathfrak{m}(\mathsf{v}_3)\big)^{\epsilon(\mathsf{v}_1\mathsf{v}_2\mathsf{v}_3)}\right)|\mathfrak{g}\rangle\langle\mathfrak{g}|.$$

$$(3.43)$$

Indeed, letting $\mathsf{v}\in\mathsf{V}(\mathbb{T}_\triangle^2)$ and $\mathfrak{x}_\mathsf{v}\in C^0(\mathbb{T}_\triangle^2,G)$ such that $\mathfrak{x}_\mathsf{v}(\mathsf{v}_1)=\mathbb{1}$, whenever $\mathsf{v}_1\neq\mathsf{v}$, it immediately follows from condition (3.41) that

$$\alpha_\phi^\rhd\big((\mathfrak{g}\,\mathrm{d}^{(0)}\mathfrak{x}_\mathsf{v})(\mathsf{v}_1\mathsf{v}_2),(\mathfrak{g}\,\mathrm{d}^{(0)}\mathfrak{x}_\mathsf{v})(\mathsf{v}_2\mathsf{v})\big)\big((\mathfrak{x}_\mathsf{v}\rhd\mathfrak{m})(\mathsf{v})\big)$$
$$=\frac{\alpha_\phi^\rhd\big(\mathfrak{g}(\mathsf{v}_2\mathsf{v})x^{-1},x\big)\big(\mathfrak{m}(\mathsf{v})\big)\,\alpha_\phi^\rhd\big(\mathfrak{g}(\mathsf{v}_1\mathsf{v}_2),\mathfrak{g}(\mathsf{v}_2\mathsf{v})\big)\big(\mathfrak{m}(\mathsf{v})\big)\,\lambda\big(\mathfrak{g}(\mathsf{v}_1\mathsf{v}_2),\mathfrak{g}(\mathsf{v}_2\mathsf{v})x^{-1},x\big)}{\alpha_\phi^\rhd\big(\mathfrak{g}(\mathsf{v}_1\mathsf{v})x^{-1},x\big)\big(\mathfrak{m}(\mathsf{v})\big)},\qquad(3.44)$$

$$\alpha_\phi^\rhd\big((\mathfrak{g}\,\mathrm{d}^{(0)}\mathfrak{x}_\mathsf{v})(\mathsf{v}_1\mathsf{v}),(\mathfrak{g}\,\mathrm{d}^{(0)}\mathfrak{x}_\mathsf{v})(\mathsf{v}\,\mathsf{v}_2)\big)\big((\mathfrak{x}_\mathsf{v}\rhd\mathfrak{m})(\mathsf{v}_2)\big)$$
$$=\frac{\alpha_\phi^\rhd\big(\mathfrak{g}(\mathsf{v}_1\mathsf{v}),\mathfrak{g}(\mathsf{v}\,\mathsf{v}_2)\big)\big(\mathfrak{m}(\mathsf{v}_2)\big)\,\alpha_\phi^\rhd\big(\mathfrak{g}(\mathsf{v}_1\mathsf{v})x^{-1},x\big)\big(\mathfrak{g}(\mathsf{v}\,\mathsf{v}_2)\rhd\mathfrak{m}(\mathsf{v}_2)\big)}{\alpha_\phi^\rhd\big(x,\mathfrak{g}(\mathsf{v}\mathsf{v}_2)\big)\big(\mathfrak{m}(\mathsf{v}_2)\big)\,\lambda\big(\mathfrak{g}(\mathsf{v}_1\mathsf{v})x^{-1},x,\mathfrak{g}(\mathsf{v}\,\mathsf{v}_2)\big)},\qquad(3.45)$$

$$\alpha_\phi^\rhd\big((\mathfrak{g}\,\mathrm{d}^{(0)}\mathfrak{x}_\mathsf{v})(\mathsf{v}\,\mathsf{v}_1),(\mathfrak{g}\,\mathrm{d}^{(0)}\mathfrak{x}_\mathsf{v})(\mathsf{v}_1\mathsf{v}_2)\big)\big((\mathfrak{x}_\mathsf{v}\rhd\mathfrak{m})(\mathsf{v})\big)$$
$$=\frac{\alpha_\phi^\rhd\big(\mathfrak{g}(\mathsf{v}\,\mathsf{v}_1),\mathfrak{g}(\mathsf{v}_1\mathsf{v}_2)\big)\big(\mathfrak{m}(\mathsf{v}_2)\big)\,\alpha_\phi^\rhd\big(x,\mathfrak{g}(\mathsf{v}\mathsf{v}_2)\big)\big(\mathfrak{m}(\mathsf{v}_2)\big)\,\lambda\big(x,g(\mathsf{v}\,\mathsf{v}_1),\mathfrak{g}(\mathsf{v}_1\mathsf{v}_2)\big)}{\alpha_\phi^\rhd\big(x,\mathfrak{g}(\mathsf{v}\,\mathsf{v}_1)\big)\big(\mathfrak{g}(\mathsf{v}_1\mathsf{v}_2)\rhd\mathfrak{m}(\mathsf{v}_2)\big)},\qquad(3.46)$$

where $\mathfrak{x}_{\mathsf{v}}(\mathsf{v}) \equiv x$. Bringing everything together, one obtains

$$\prod_{(\mathsf{v}_1\mathsf{v}_2\mathsf{v}_3)} \alpha_\phi^\triangleright \big((\mathfrak{g}\,\mathrm{d}^{(0)}\mathfrak{x}_{\mathsf{v}})(\mathsf{v}_1\mathsf{v}_2), (\mathfrak{g}\,\mathrm{d}^{(0)}\mathfrak{x}_{\mathsf{v}})(\mathsf{v}_2\mathsf{v}_3)\big)\big((\mathfrak{x}_{\mathsf{v}} \triangleright \mathfrak{m})(\mathsf{v}_3)\big)^{\epsilon(\mathsf{v}_1\mathsf{v}_2\mathsf{v}_3)}$$
$$= \lambda(\mathfrak{g},\mathfrak{x}_{\mathsf{v}}) \cdot \prod_{(\mathsf{v}_1\mathsf{v}_2\mathsf{v}_3)} \alpha_\phi^\triangleright \big(\mathfrak{g}(\mathsf{v}_1\mathsf{v}_2), \mathfrak{g}(\mathsf{v}_2\mathsf{v}_3)\big)\big(\mathfrak{m}(\mathsf{v}_3)\big)^{\epsilon(\mathsf{v}_1\mathsf{v}_2\mathsf{v}_3)}, \tag{3.47}$$

where $\lambda(\mathfrak{g},\mathfrak{x}_{\mathsf{v}})$ was defined in eq. (3.5). This guarantees that $\mathbb{D}^{A_\phi}(\mathbb{T}_\triangle^2) \circ \mathbb{H} = \mathbb{H}^\lambda \circ \mathbb{D}^{A_\phi}(\mathbb{T}_\triangle^2)$, as desired. Notice that the above derivation only relies on (3.41), which is verified for any $\mathsf{Vec}_G^\lambda$-module category, and thus does not depend on a specific choice $\mathcal{M}(A,\phi)$. From this case, it would be easy to infer the duality operators transmuting Hamiltonians $\mathbb{H}^\lambda$ and $\mathbb{H}^{\lambda'}$ into each other.

Suppose $\lambda$ is a non-trivial 3-cocycle. It follows from (3.41) that it is not possible to choose the whole group $G$ to define a $\mathsf{Vec}_G^\lambda$-module category, as otherwise it would require $\phi$ to be a 2-cochain in $C^2(G, \mathrm{U}(1))$ trivialising $\lambda$, which would be a contradiction. Therefore, a duality operator $\mathbb{D}^{A_\phi}(\mathbb{T}_\triangle^2)$ is necessarily associated with a $\mathsf{Vec}_G^\lambda$-module category $\mathcal{M}(A,\phi)$ with $A$ a *proper* subgroup of $G$. This implies in particular that any duality operator $\mathbb{D}^{A_\phi}(\mathbb{T}_\triangle^2)$ necessarily projects the Hamiltonians $\mathbb{H}$ and $\mathbb{H}^\lambda$ onto specific charge sectors. Consider for instance the duality operator $\mathbb{D}^{\{\mathbb{1}\}}(\mathbb{T}_\triangle^2)$ associated with the $\mathsf{Vec}_G^\lambda$-module category $\mathcal{M}(\{\mathbb{1}\}, 1) \simeq \mathsf{Vec}_G^\lambda$, whose module associator evaluates to $\lambda$ so that (3.41) merely follows from $\mathrm{d}^{(3)}\lambda = 1$. Similarly to the 0-form symmetry operator $\mathbb{U}^{\{\mathbb{1}\}}(\mathbb{T}_\triangle^2)$, the duality operator $\mathbb{D}^{\{\mathbb{1}\}}(\mathbb{T}_\triangle^2)$ enforces holonomies along both non-contractible cycles to be trivial, and thus acts as a projector onto the singlet sector of the 1-form $\mathsf{Rep}(G)$ symmetry, making this duality operator non-invertible, and a fortiori non-unitary. Nonetheless, upon restricting to the singlet sector, the operator $\mathbb{D}^{\{\mathbb{1}\}}(\mathbb{T}_\triangle^2)$ does become invertible and relates the spectra of the dual Hamiltonians. In order to access additional charge sectors, one option is to decorate the topological surface with topological lines labelled by elements in $G$, in the same vein as for the symmetry operator $\mathbb{U}^{\{\mathbb{1}\}}(\mathbb{T}_\triangle^2)$ (see sec. 3.5). However, this is insufficient as the duality operator typically imposes additional constraints on the charge sectors that depend on the choice of $\lambda$. In the same vein, employing a different duality operator associated with a distinct $\mathcal{M}(A,\phi)$ could a priori permit to access additional charge sectors, as it would not necessarily project onto the singlet sector, but there would still be additional constraints that depend on $\lambda$. Indeed, suppose we are restricting to charge sectors $(a_1, a_2) \in A \times A$. It follows from the cocycle condition (3.41) that the duality explicitly depends on the complex number[12]

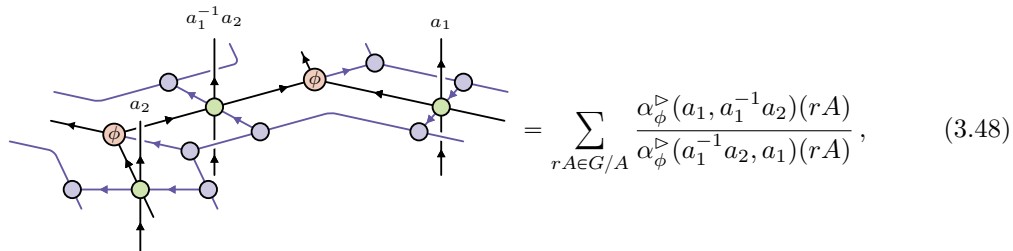

$$= \sum_{rA \in G/A} \frac{\alpha_\phi^\triangleright(a_1, a_1^{-1}a_2)(rA)}{\alpha_\phi^\triangleright(a_1^{-1}a_2, a_1)(rA)}, \tag{3.48}$$

where the l.h.s. should be interpreted as a single unit cell of the tensor network whose opposite indices are contracted so to as implement periodic boundary conditions. Consider now shifting the summation variable as follows: $rA \mapsto x \triangleright rA$, for any $x \in G$. But, repeatedly using the cocycle conditions (3.41)

---

[12]To establish this fact, we are employing a higher dimensional version of the trick presented in eq. (2.15).

and $\mathrm{d}^{(3)}\lambda = 1$, as well as the fact that $a \triangleright rA = rA$ for every $a \in A$ and $rA \in G/A$, one finds that

$$\frac{\alpha_\phi^\triangleright(a_1, a_1^{-1}a_2)(x \triangleright rA)}{\alpha_\phi^\triangleright(a_1^{-1}a_2, a_1)(x \triangleright rA)} = \frac{\mathsf{t}_x(\lambda)(a_1, a_1^{-1}a_2)}{\mathsf{t}_x(\lambda)(a_1^{-1}a_2, a_1)} \cdot \frac{\alpha_\phi^\triangleright(a_1, a_1^{-1}a_2)(rA)}{\alpha_\phi^\triangleright(a_1^{-1}a_2, a_1)(rA)}, \tag{3.49}$$

where $\mathsf{t}_x : Z^3(G, \mathrm{U}(1)) \to Z^2(G, \mathrm{U}(1))$ is defined via [Wil05]

$$\mathsf{t}_x(\lambda)(g_1, g_2) := \frac{\lambda(x, g_1, g_2)\,\lambda(g_1, g_2, x)}{\lambda(g_1, x, g_2)}, \tag{3.50}$$

for every $g_1, g_2, x \in G$. Additional manipulations reveal that

$$\frac{\mathsf{t}_x(\lambda)(a_1, a_1^{-1}a_2)}{\mathsf{t}_x(\lambda)(a_1^{-1}a_2, a_1)} = \mathsf{t}^2_{a_1, a_1^{-1}a_2}(\lambda)(x) = \mathsf{t}^2_{a_1, a_2}(\lambda)(x), \tag{3.51}$$

where $\mathsf{t}^2_{x,y} : Z^3(G, \mathrm{U}(1)) \to Z^1(G, \mathrm{U}(1))$ is defined via

$$\mathsf{t}^2_{x_1, x_2}(\lambda)(g) := \frac{\mathsf{t}_{x_1}(\lambda)(x_2, g)}{\mathsf{t}_{x_1}(\lambda)(g, x_2)}, \tag{3.52}$$

for every $g, x_1, x_2 \in G$. Bringing everything together, we showed that within the charge sector $(a_1, a_2) \in A \times A$, $\mathbb{D}^{A_\phi}(\mathbb{T}^2_\triangle) = \mathsf{t}^2_{a_1, a_2}(\lambda)(x) \cdot \mathbb{D}^{A_\phi}(\mathbb{T}^2_\triangle)$, for every $x \in G$. Since it is true for every $x \in G$ and $\mathsf{t}^2_{a_1, a_2}(\lambda) \in \widehat{G}$, we further have

$$\mathbb{D}^{A_\phi}(\mathbb{T}^2_\triangle) = \left(\frac{1}{|G|}\sum_{x \in G} \mathsf{t}^2_{a_1, a_2}(\lambda)(x)\right)\mathbb{D}^{A_\phi}(\mathbb{T}^2_\triangle) = \begin{cases} \mathbb{D}^{A_\phi}(\mathbb{T}^2_\triangle) & \text{if } \mathsf{t}^2_{a_1, a_2}(\lambda) = 1 \\ 0 & \text{otherwise} \end{cases}, \tag{3.53}$$

which means that charge sectors $(a_1, a_2) \in A \times A$, for which $\mathsf{t}^2_{a_1, a_2}(\lambda)$ is not the trivial representation are projected out. In order to access all the charge sectors—regardless of the choice of duality operator—it is required to allow for the duality operator to modify the closed boundary conditions. We compute in the following the relevant permutation of topological sectors.

### 3.5  Permutation of topological sectors

As alluded to above, in order to lift a duality operator $\mathbb{D}^{A_\phi}(\mathbb{T}^2_\triangle)$ to a unitary transformation, it is required to characterise the interplay between boundary conditions and charge sectors. First, we need to modify the duality operators so as to accommodate the presence of symmetry twists, the same way we modified the symmetry operators. Since for every group $G$ and $\lambda$, the only duality operator whose existence we can guarantee is $\mathbb{D}^{\{1\}}(\mathbb{T}^2_\triangle)$, we shall focus on this case throughout the remainder of this section. Suppose a 1-form symmetry twist is inserted at the vertex $\mathsf{v}_0$. Specialising to the triangular lattice, we define the duality operator $\mathbb{D}^{\{1\}}_{\eta_1 \to \eta_2}(\mathbb{T}^2_\triangle) : \mathcal{H}^{\eta_1} \to \mathcal{H}^{\eta_2}$ by replacing the unit cell (3.42) of its tensor network parametrisation located at $\mathsf{v}_0$ by the following one:

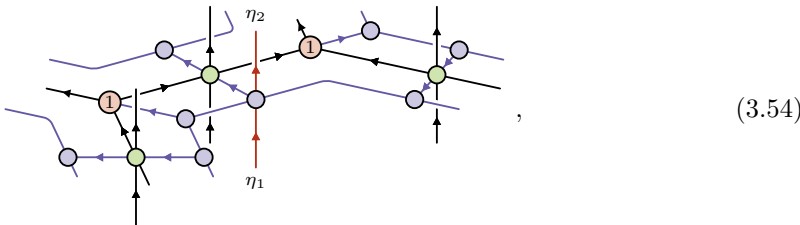

$$\tag{3.54}$$

where we are employing the tensors defined in eq. (3.35). It is clear from the previous derivations that $\mathbb{D}^{\{\mathbb{1}\}}_{\eta_1 \to \eta_2}(\mathbb{T}^2_\triangle) \circ \mathbb{H}^{1,\eta_1} = \mathbb{H}^{\lambda,\eta_2} \circ \mathbb{D}^{\{\mathbb{1}\}}_{\eta_1 \to \eta_2}(\mathbb{T}^2_\triangle)$. Now, recall that the duality operator $\mathbb{D}^{\{\mathbb{1}\}}_{\eta_1 \to \eta_2}(\mathbb{T}^2_\triangle)$ projects out states $|\mathfrak{g}\rangle$ such that $\mathrm{hol}_{\gamma_1}(\mathfrak{g})$ and $\mathrm{hol}_{\gamma_2}(\mathfrak{g})$ are not the identity element. We can lift this limitation by decorating $\mathbb{D}^{\{\mathbb{1}\}}_{\eta_1 \to \eta_2}(\mathbb{T}^2_\triangle)$ by appropriate topological lines in $G$, thereby allowing to consider any charge sector $(g_1, g_2) \in G \times G$. More formally, the duality operator $\mathbb{D}^{\{\mathbb{1}\}}(\mathbb{T}^2_\triangle)$ being associated with the $\mathsf{Vec}_G^\lambda$-module category $\mathcal{M}(\{\mathbb{1}\}, 1) \simeq \mathsf{Vec}_G^\lambda$, it can be decorated with 't Hooft lines labelled by simple objects in the category $(\mathsf{Vec}_G^\lambda)^\vee_{\mathcal{M}(\{\mathbb{1}\},1)} \simeq \mathsf{Vec}_G^\lambda$. Therefore, although the 't Hooft lines are effectively labelled by group elements in $G$, their action is more subtle than in (3.21) and depends in particular on $\lambda$. Correspondingly, it is not enough to implement the group multiplication at junctions in order to ensure topological invariance of network of lines. Since any network of 't Hooft lines boils down to a collection of single 't Hooft loops, by virtue of the group $G$ being abelian, let us focus for now on a single 't Hooft loop labelled by $f \in G$. Consider an oriented closed loop in $Z_1(\mathbb{T}^2_\triangle, \mathbb{Z})$. For every edge in this closed loop, we modify the corresponding local patch of the tensor network as follows

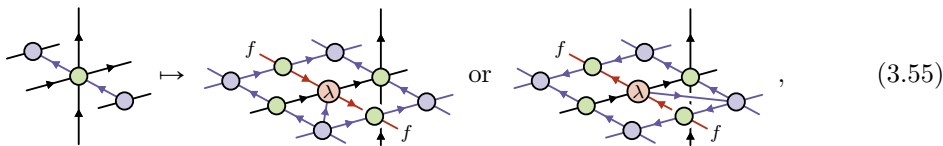

$$\tag{3.55}$$

depending on the orientation of the edge relative to that of the loop, where we are using tensors adapted from eq. (3.23) in an obvious way and

$$\vcenter{\hbox{\includegraphics{tensor1}}} \equiv \sum_{\substack{f_1,f_2,g_1 \\ g_2,x \in G}} f_1 \to \vcenter{\hbox{\includegraphics{tensor1b}}} \to f_2 \; |f_2\rangle\langle f_1| \otimes |g_2\rangle\langle g_1| \otimes \langle x| \equiv \sum_{f,g,x \in G} \lambda(g,x,f) \, |f\rangle\langle f| \otimes |g\rangle\langle g| \otimes \langle x| \, ,$$

$$\vcenter{\hbox{\includegraphics{tensor2}}} \equiv \sum_{\substack{f_1,f_2,g_1 \\ g_2,x \in G}} f_1 \to \vcenter{\hbox{\includegraphics{tensor2b}}} \to f_2 \; |f_1\rangle\langle f_2| \otimes |g_2\rangle\langle g_1| \otimes |x\rangle \equiv \sum_{f,g,x \in G} \lambda(g,x,f)^{-1} \, |f\rangle\langle f| \otimes |g\rangle\langle g| \otimes |x\rangle \, .$$

One can check that the cocycle condition $\mathrm{d}^{(3)}\lambda = 1$ ensures that the resulting duality operator is left invariant under continuous deformations of the closed loop. Although the definition of these operators may seem somewhat ad hoc, it is actually a reinterpretation of another tensor network calculus that is closely related to that of fusion 2-categories [Del21]. For instance, given the data of a $\mathsf{Vec}_G^\lambda$-module endofunctor of $\mathcal{M}(A, \phi)$, there is a systematic way of constructing a tensor network operator, which boil down to that defined in eq. (3.55) whenever $\mathcal{M}(A, \phi) \simeq \mathsf{Vec}_G^\lambda$. In order to construct the operator $\mathbb{D}^{\mathbb{1}}_{\eta_1 \to \eta_2}(\mathbb{T}^2_\triangle)[g_1, g_2]$ projecting onto a charge sector labelled by $[\mathfrak{f}] \in H^1(G, \mathbb{T}^2_\triangle)$ with holonomies $\mathrm{hol}_{\gamma_1}(\mathfrak{f}) = g_1$ and $\mathrm{hol}_{\gamma_2}(\mathfrak{f}) = g_2$, it suffices to insert two loop operators as defined above along both non-contractible cycle of the torus. However, as mentioned above, the junction of these two loops operators requires some care so as to be topologically invariant. Immediately specialising to our configuration of interest, one finds that the duality operator $\mathbb{D}^{\{\mathbb{1}\}}_{\eta_1 \to \eta_2}(\mathbb{T}^2_\triangle)[g_1, g_2] : \mathcal{H}^{\eta_1} \to \mathcal{H}^{\eta_2}$

explicitly depends on the complex number

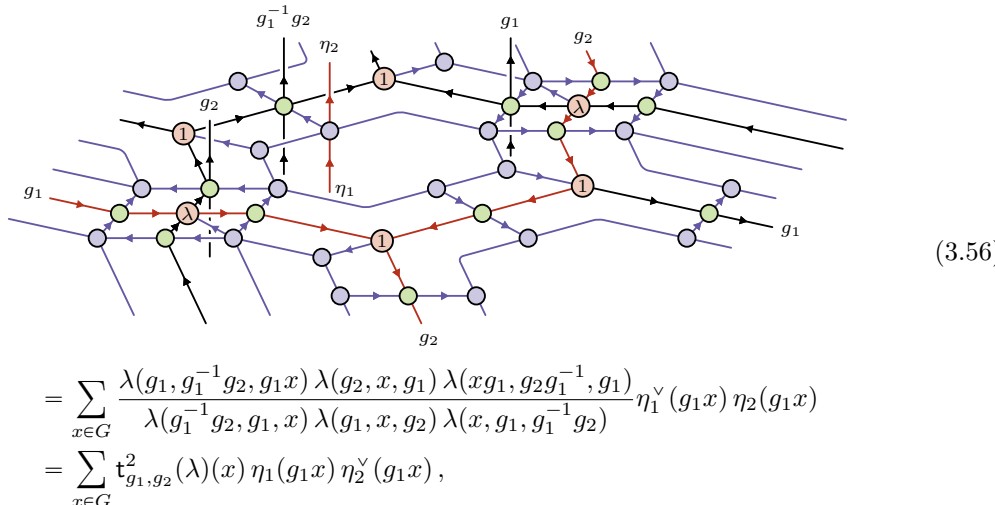

$$= \sum_{x \in G} \frac{\lambda(g_1, g_1^{-1}g_2, g_1 x)\,\lambda(g_2, x, g_1)\,\lambda(x g_1, g_2 g_1^{-1}, g_1)}{\lambda(g_1^{-1}g_2, g_1, x)\,\lambda(g_1, x, g_2)\,\lambda(x, g_1, g_1^{-1}g_2)}\eta_1^\vee(g_1 x)\,\eta_2(g_1 x)$$

$$= \sum_{x \in G} \mathsf{t}^2_{g_1,g_2}(\lambda)(x)\,\eta_1(g_1 x)\,\eta_2^\vee(g_1 x)\,,$$

where, as in eq. (3.48), the l.h.s. is interpreted as a single unit cell of the tensor network whose opposite indices are contracted so to as implement periodic boundary conditions. In addition to the unit cell (3.54) and tensors of the form (3.55), one introduced the following tensors that implement the fusion of topological lines in $\mathsf{Vec}_G^\lambda$:

$$\equiv \sum_{\substack{f_1, f_2 \\ f_3, x \in G}} \quad |f_1, f_2\rangle\langle f_3| \otimes \langle x| \equiv \sum_{f_1, f_2, x \in G} \lambda(x, f_2, f_1)\,|f_1, f_2\rangle\langle f_1 f_2| \otimes \langle x|\,,$$

$$\equiv \sum_{\substack{f_1, f_2 \\ f_3, x \in G}} \quad |f_3\rangle\langle f_1, f_2| \otimes |x\rangle \equiv \sum_{f_1, f_2, x \in G} \lambda(x, f_2, f_1)^{-1}\,|f_1 f_2\rangle\langle f_1, f_2| \otimes |x\rangle\,.$$

It simply follows from the orthogonality of representations in $G$ that, in order for the operator $\mathbb{D}^{\{\mathbb{1}\}}_{\eta_1 \to \eta_2}(\mathbb{T}^2_\triangle)[g_1, g_2]$ not to vanish, we must have $\eta_2(-) = \mathsf{t}^2_{g_1,g_2}(\lambda)(-)\,\eta_1(-)$. Bringing everything together, one obtains the following mapping of topological sectors:

$$(g_1, g_2, \eta) \mapsto (g_1, g_2, \eta^\lambda_{g_1,g_2}) \quad \text{with} \quad \eta^\lambda_{g_1,g_2}(-) := \mathsf{t}^2_{g_1,g_2}(\lambda)(-)\,\eta(-)\,. \tag{3.57}$$

In particular, it follows from the normalisation of $\lambda$ that the topological sectors of the form $(\eta, \mathbb{1}, \mathbb{1})$ are mapped to themselves under $\mathbb{D}^{A_\phi}(\mathbb{T}^2_\triangle)$ for any $\eta \in \widehat{G}$. Finally, promoting the 1-form symmetry twist to a dynamical degree of freedom, the invertible duality operators between compatible topological sectors constructed above organise into a unitary operator onto the resulting total Hilbert space.

### 3.6 Examples

Let illustrate our formalism with a couple of examples. Let $G = \mathbb{Z}_2$. As in sec. 2.4, we write the multiplication rule of $\mathbb{Z}_2 = \{0, 1\}$ as addition modulo 2. A normalised representative of the cohomology class $[\lambda]$ generating $H^3(\mathbb{Z}_2, \mathrm{U}(1)) \cong \mathbb{Z}_2$ is provided by $\lambda(g_1, g_2, g_3) = (-1)^{g_1 g_2 g_3}$, for every $g_1, g_2, g_3 \in \mathbb{Z}_2$. Given a two-dimensional quantum lattice model with a 0-form $\mathbb{Z}_2$ symmetry, we

have two gauging procedures at our disposal, namely the untwisted one and the one twisted by the representative $\lambda$ defined above. The two Hamiltonians resulting from these two gauging procedures possess a $2\mathsf{Rep}(\mathbb{Z}_2)$ symmetry. Moreover, since they only differ in the choice of 3-cocycle $\lambda$, the corresponding duality operator is encoded into the unique simple object in the 2-category $2\mathsf{Rep}^\lambda(\mathbb{Z}_2)$, which is $\mathsf{Vec}^\lambda_{\mathbb{Z}_2}$ as a module category over itself. Let us consider the special Hamiltonian whose local operators (3.3) are obtained by choosing the non-vanishing complex coefficients $h(\mathfrak{g}, \mathfrak{x}_\mathsf{v})$ to be equal to $-1$, for every $\mathfrak{g} \in Z^1(\mathbb{T}^2_\triangle, \mathbb{Z}_2)$ and $\mathfrak{x}_\mathsf{v} \in C^0(\mathbb{T}^2_\triangle, \mathbb{Z}_2)$ such that $\mathfrak{x}_\mathsf{v}(\mathsf{v}) = 1$, i.e.,

$$\mathbb{H} = -\sum_\mathsf{v} \prod_{\mathsf{e} \supset \mathsf{v}} X_\mathsf{e}\,. \tag{3.58}$$

Recall that the kinematical Hilbert space is spanned by states $|\mathfrak{g}\rangle$, where $\mathfrak{g} \in Z^1(\mathbb{T}^2_\triangle, \mathbb{Z}_2)$. Making the kinematical constraint $\ker \mathrm{d}^{(1)}\mathfrak{g} = 0$ dynamical would result in the celebrated $(2+1)$d *toric code* model [Kit03]. Let us briefly comment on the $2\mathsf{Rep}(\mathbb{Z}_2)$ symmetry structure. By definition, there are two topological surface operators, namely the identity operator and the condensation operator[13]

$$\mathbb{U}^{\{\mathbb{1}\}}(\mathbb{T}^2_\triangle) = \sum_{\substack{\mathfrak{g} \in Z^1(\mathbb{T}^2_\triangle, \mathbb{Z}_2) \\ \mathfrak{m} \in C^0(\mathbb{T}^2_\triangle, \mathbb{Z}_2)}} \delta_{\mathfrak{g}, \mathrm{d}^{(0)}\mathfrak{m}} |\mathfrak{g}\rangle\langle\mathfrak{g}| = \frac{1}{2^{\mathsf{P}(\mathbb{T}^2_\triangle)}} \Big( \bigotimes_\mathsf{v} \langle+| \Big) \prod_\mathsf{e} \big( \mathbb{1} + Z_{\partial_-\mathsf{e}} Z_\mathsf{e} Z_{\partial_+\mathsf{e}} \big) \Big( \bigotimes_\mathsf{v} |+\rangle \Big)\,, \tag{3.59}$$

where $|+\rangle := \frac{1}{\sqrt{2}}(|0\rangle + |1\rangle)$. Decorating the trivial surface operator $\mathbb{U}^{\mathbb{Z}_2}(\mathbb{T}^2_\triangle)$ with the network of Wilson lines encoded into the 1-cocycle $\mathfrak{X} \in Z^1(\mathbb{T}^2_\triangle, \mathbb{Z}_2)$ results into the surface operator[14]

$$\mathbb{U}^{\mathbb{Z}_2}(\mathbb{T}^2_\triangle)[\mathfrak{X}] \equiv \mathfrak{X}(\mathbb{T}^2_\triangle) = \sum_{\mathfrak{g} \in Z^1(\mathbb{T}^2_\triangle, \mathbb{Z}_2)} \exp\Big( i\pi \int_{\mathbb{T}^2_\triangle} \mathfrak{X} \smile \mathfrak{g} \Big) |\mathfrak{g}\rangle\langle\mathfrak{g}|\,. \tag{3.60}$$

Decorating $\mathbb{U}^{\{\mathbb{1}\}}(\mathbb{T}^2_\triangle)$ with the network of 't Hooft lines encoded into the 1-cocycle $\mathfrak{f} \in Z^1(\mathbb{T}^2_\triangle, \mathbb{Z}_2)$ results into the surface operator

$$\mathbb{U}^{\{\mathbb{1}\}}(\mathbb{T}^2_\triangle)[\mathfrak{f}] = \sum_{\substack{\mathfrak{g} \in Z^1(\mathbb{T}^2_\triangle, \mathbb{Z}_2) \\ \mathfrak{m} \in C^0(\mathbb{T}^2_\triangle, \mathbb{Z}_2)}} \delta_{\mathfrak{g}, \mathrm{d}^{(0)}\mathfrak{m}+\mathfrak{f}} |\mathfrak{g}\rangle\langle\mathfrak{g}| = \frac{1}{2^{\mathsf{P}(\mathbb{T}^2_\triangle)}} \Big( \bigotimes_\mathsf{v} \langle+| \Big) \prod_\mathsf{e} \big( \mathbb{1} + (-1)^{\mathfrak{f}(\mathsf{e})} Z_{\partial_-\mathsf{e}} Z_\mathsf{e} Z_{\partial_+\mathsf{e}} \big) \Big( \bigotimes_\mathsf{v} |+\rangle \Big)\,.$$

Replacing in the definition of the local operators the trivial 3-cocycle by $\lambda$, while keeping everything else the same, yields the Hamiltonian $\mathbb{H}^\lambda$. Note that $\mathbb{H}^\lambda$ does not admit a particularly compact expression. However, specialising to the scenario where $\mathbb{T}^2_\triangle$ is the triangular lattice, one can perform the unitary transformation [SPHMD19]

$$U := \sum_\mathfrak{g} \int_{\mathbb{T}^2_\triangle} i^\mathfrak{g} |\mathfrak{g}\rangle\langle\mathfrak{g}| \tag{3.61}$$

so as to obtain

$$U\,\mathbb{H}^\lambda\,U^\dagger = \sum_\mathsf{v} \prod_{\mathsf{e} \supset \mathsf{v}} X_\mathsf{e} \prod_{\substack{(\mathsf{v}\,\mathsf{v}_1\mathsf{v}_2) \\ (\mathsf{v}_1\,\mathsf{v}\,\mathsf{v}_2) \\ (\mathsf{v}_1\mathsf{v}_2\mathsf{v})}} S_{(\mathsf{v}_1\mathsf{v}_2)}\,, \tag{3.62}$$

where $S = |0\rangle\langle0| + i|1\rangle\langle1|$. At this point, making the kinematical constraint $\ker \mathrm{d}^{(1)}\mathfrak{g} = 0$ dynamical would result in the Hamiltonian of the so-called *double semion* model [LW04, LG12], which is the

---

[13]Recall that $\mathsf{P}(\mathbb{T}^2_\triangle) = \mathsf{E}(\mathbb{T}^2_\triangle) - \mathsf{V}(\mathbb{T}^2_\triangle)$ since the Euler characteristic of $\mathbb{T}^2$ equals 0.

[14]Here, we are exploiting the isomorphism $\mathbb{Z}_2 \cong \widehat{\mathbb{Z}_2}$ in order to simplify the definition of the cup product (3.11).

Hamiltonian realisation of the Turaev–Viro–Barrett–Westbury state-sum invariant with input fusion category $\mathsf{Vec}^\lambda_{\mathbb{Z}_2}$ [TV92, BW93]. Assuming periodic boundary conditions for both $\mathbb{H}$ and $\mathbb{H}^\lambda$, the duality operator $\mathbb{D}^{\{\mathbb{1}\}}(\mathbb{T}^2_\triangle)$ transmuting the Hamiltonians $\mathbb{H}$ and $\mathbb{H}^\lambda$ into each other takes the following form:

$$
\begin{aligned}
\mathbb{D}^{\{\mathbb{1}\}}(\mathbb{T}^2_\triangle) &= \sum_{\substack{\mathfrak{g} \in Z^1(\mathbb{T}^2_\triangle, \mathbb{Z}_2) \\ \mathfrak{m} \in C^0(\mathbb{T}^2_\triangle, \mathbb{Z}_2)}} \delta_{\mathfrak{g}, \mathrm{d}^{(0)}\mathfrak{m}} \left( \prod_{(\mathsf{v}_1 \mathsf{v}_2 \mathsf{v}_3)} \lambda\big(\mathfrak{g}(\mathsf{v}_1\mathsf{v}_2), \mathfrak{g}(\mathsf{v}_2\mathsf{v}_3), \mathfrak{m}(\mathsf{v}_3)\big)^{\epsilon(\mathsf{v}_1\mathsf{v}_2\mathsf{v}_3)} \right) |\mathfrak{g}\rangle\langle\mathfrak{g}| \\
&= \sum_{\mathfrak{m} \in C^0(\mathbb{T}^2_\triangle, \mathbb{Z}_2)} \left( \prod_{(\mathsf{v}_1\mathsf{v}_2\mathsf{v}_3)} (-1)^{\mathfrak{m}(\mathsf{v}_1)\mathfrak{m}(\mathsf{v}_3)(\mathfrak{m}(\mathsf{v}_2)+1)} \right) |\mathrm{d}^{(0)}\mathfrak{m}\rangle\langle\mathrm{d}^{(0)}\mathfrak{m}| \,.
\end{aligned}
\tag{3.63}
$$

One can now verify that $\mathbb{D}^{\{\mathbb{1}\}}(\mathbb{T}^2_\triangle) \circ \mathbb{H} = \mathbb{H}^\lambda \circ \mathbb{D}^{\{\mathbb{1}\}}(\mathbb{T}^2_\triangle)$. What about the mapping of topological sectors? Since $\lambda(g_1, g_2, g_3) = \lambda(g_2, g_3, g_1) = \lambda(g_3, g_1, g_2)$, for every $g_1, g_2, g_3 \in \mathbb{Z}_2$, it turns out that $\mathsf{t}^2_{g_1,g_2}(x) = 1$, for every $x \in \mathbb{Z}_2$. Therefore, any 1-form symmetry twisted boundary condition is exceptionally preserved under the duality transmuting $\mathbb{H}$ into $\mathbb{H}^\lambda$. Naturally, $\mathbb{D}^{\{\mathbb{1}\}}(\mathbb{T}^2_\triangle)$ is not a unitary operator on the total kinematical Hilbert space, since it projects both $\mathbb{H}$ and $\mathbb{H}^\lambda$ onto the singlet sector of the 1-form $\mathsf{Rep}(\mathbb{Z}_2)$-symmetry. In order to obtain the unitary operator relating the full spectra of $\mathbb{H}$ and $\mathbb{H}^\lambda$, it is required to decorate $\mathbb{D}^{\{\mathbb{1}\}}(\mathbb{T}^2_\triangle)$ with networks of 't Hooft lines, following the procedure described in sec. 3.5.

Since the previous example is not accompanied with a non-trivial permutation of topological sectors, let us briefly consider a slightly more complicated example. Let $G = \mathbb{Z}_2 \times \mathbb{Z}_2 \times \mathbb{Z}_2 \ni (g^{(1)}, g^{(2)}, g^{(3)}) \equiv g$. There is a normalised representative of a cohomology class in $H^3(\mathbb{Z}_2^3, \mathrm{U}(1))$ defined by

$$
\lambda(g_1, g_2, g_3) = \exp\left(i\pi g_1^{(1)} g_2^{(2)} g_3^{(3)}\right),
\tag{3.64}
$$

for every $g_1, g_2, g_3 \in \mathbb{Z}_2^3$. It follows from the general derivation that upon restricting to the charge sector $(g_1, g_2) \in \mathbb{Z}_2^3 \times \mathbb{Z}_2^3$, the duality operator $\mathbb{D}^{\{\mathbb{1}\}}_{1 \to \eta}(\mathbb{T}^2_\triangle)[g_1, g_2]$ would be non-vanishing if and only if $\eta(-) = \mathsf{t}^2_{g_1,g_2}(-)$ with

$$
\mathsf{t}^2_{g_1,g_2}(x) = \exp\left(i\pi \begin{vmatrix} g_1^{(1)} & g_2^{(1)} & x^{(1)} \\ g_1^{(2)} & g_2^{(2)} & x^{(2)} \\ g_1^{(3)} & g_2^{(3)} & x^{(3)} \end{vmatrix}\right),
\tag{3.65}
$$

for every $x \in \mathbb{Z}_2^3$. Therefore, for a generic charge sector, the 1-form symmetry twisted boundary condition of the Hamiltonian $\mathbb{H}^\lambda$ would need to be non-trivial.

## 3.7 Higher category theoretic underpinnings

Transposing sec. 2.5, let us sketch the mathematical formalism underlying the results presented above, following the approach of ref. [DT23]. Consider a two-dimensional quantum lattice model with a non-anomalous abelian symmetry group $G$, which admits topological surfaces valued in $G$. The requirement that we should be able to construct topological interfaces between surfaces and topological junctions of interfaces invites us to rather consider a higher category theoretic structure, namely the *fusion 2-category* $2\mathsf{Vec}_G$ of $G$-graded 2-vector spaces, where a 2-vector space is defined to be a finite semisimple $\mathbb{C}$-linear (1-)category. In particular, simple objects in $2\mathsf{Vec}_G$ are 2-vector spaces $\mathsf{Vec}_g$, for every $g \in G$, such that $(\mathsf{Vec}_{g_1})_{g_2} = \delta_{g_1,g_2}\mathsf{Vec}$ with hom-categories $\mathsf{Hom}_{2\mathsf{Vec}_G}(\mathsf{Vec}_{g_1}, \mathsf{Vec}_{g_2}) \cong \delta_{g_1,g_2}\mathsf{Vec}$, while the monoidal structure reads $\mathsf{Vec}_{g_1} \odot \mathsf{Vec}_{g_2} = \mathsf{Vec}_{g_1 g_2}$. Moreover, the monoidal associator evaluates to the identity 1-morphism, and it satisfies the pentagon axiom up to an invertible modification, known

as the *monoidal pentagonator*, which evaluates to the identity 2-morphism. Given a non-anomalous symmetry $G$ in (2+1)d—or rather, a symmetry $2\mathsf{Vec}_G$—it is always possible to gauge a subsymmetry $A$, where $A$ is a subgroup of $G$. As in (1+1)d, we identify several ways to perform such a gauging, which are now labelled by cohomology classes in $H^3(A, \mathrm{U}(1))$. Whenever the whole symmetry is gauged, the resulting theory possesses a $2\mathsf{Rep}(G)$-symmetry, i.e., a symmetry structure encompassing both 0-form symmetry operators labelled by 2-representations of the group $G$, which compose according to the fusion of $\mathsf{Vec}_G$-module categories [Gre10], as well as 1-form symmetry operators labelled by $\mathsf{Vec}_G$-module functors. Crucially, even though the group is abelian, $2\mathsf{Vec}_G$ and $2\mathsf{Rep}(G)$ are clearly not equivalent, even as 2-categories.

Mathematically, a pair $(A, \lambda)$ consisting of a subgroup $A \subseteq G$ and a normalised representative $\lambda$ of a cohomology $[\lambda] \in H^3(A, \mathrm{U}(1))$ specifies a so-called *module 2-category* over $2\mathsf{Vec}_G$ [Dé21, Del21]. Let $\mathcal{M}(A, \lambda)$ be the (finite semisimple) 2-category whose simple objects are 2-vector spaces $\mathsf{Vec}_{rA}$, for every $rA \in G/A$. It is equipped with the (left) $2\mathsf{Vec}_G$-module structure provided by the $G$-action of left cosets in $G/A$, i.e. $\mathsf{Vec}_g \rhd \mathsf{Vec}_{rA} := \mathsf{Vec}_{g \rhd rA} \simeq \mathsf{Vec}_{(gr)A}$, such that the module associator evaluates to the identity 1-morphisms, and it satisfies the pentagon axiom up to an invertible modification $\pi_\lambda^\rhd$ referred as the *module pentagonator*. Components of the module pentagonator $\pi_\lambda^\rhd$ evaluate to the 3-cochain $\pi_\lambda^\rhd \in C^3(G, \mathrm{Fun}(G/A, \mathrm{U}(1)))$ defined in terms of the 3-cocycle $\lambda$ via

$$\pi_\lambda^\rhd(g_1, g_2, g_3)(rA) := \lambda(a_{g_1, g_2 g_3 \rhd rA}, a_{g_2, g_3 \rhd rA}, a_{g_3, rA}), \tag{3.66}$$

for every $g_1, g_2, g_3 \in G$ and $rA \in G/A$. It follows from the 3-cocycle condition satisfied by $\lambda$ that

$$\frac{\pi_\lambda^\rhd(g_2, g_3, g_4)(rA)\, \pi_\lambda^\rhd(g_1, g_2 g_3, g_4)(rA)\, \pi_\lambda^\rhd(g_1, g_2, g_3)(g_4 \rhd rA)}{\pi_\lambda^\rhd(g_1 g_2, g_3, g_4)(rA)\, \pi_\lambda^\rhd(g_1, g_2, g_3 g_4)(rA)} = 1, \tag{3.67}$$

for every $g_1, g_2, g_3, g_4 \in G$ and $rA \in G/A$. Identity (3.67) guarantees the so-called *associahedron axiom* of $\mathcal{M}(A, \lambda)$ [Dé21, Del21].

Performing the $\lambda$-twisted gauging of a symmetry $2\mathsf{Vec}_G$ amounts to picking the $2\mathsf{Vec}_G$-module 2-category $\mathcal{M}(G, \lambda) \equiv 2\mathsf{Vec}^\lambda$, which is equivalent to $2\mathsf{Vec}$ as a 2-category but whose $2\mathsf{Vec}_G$-module structure depends on $\lambda$. The same data encodes a choice of *fiber 2-functor* $2\mathsf{Vec}_G \to 2\mathsf{Vec}$. The symmetry structure of the theory resulting from the $\lambda$-twisted gauging of the subsymmetry $A$ is provided by the fusion 2-category $(2\mathsf{Vec}_G)^\vee_{\mathcal{M}(A, \lambda)} := 2\mathsf{Fun}_{2\mathsf{Vec}_G}(\mathcal{M}(A, \lambda), \mathcal{M}(A, \lambda))$ defined as the category of $2\mathsf{Vec}_G$-module 2-functors from $\mathcal{M}(A, \lambda)$ to itself, with fusion structure provided by the composition of $2\mathsf{Vec}_G$-module 2-functors [Del21, BBSNT22, BBFP22, DT23]. In particular, we have $(2\mathsf{Vec}_G)^\vee_{2\mathsf{Vec}^\lambda} \simeq 2\mathsf{Rep}(G)$, for any $[\lambda] \in H^3(G, \mathrm{U}(1))$, as expected.

We can think of the $2\mathsf{Rep}(G)$-symmetric models defined in sec. 3.1 as resulting from the $\lambda$-twisted gauging of $G$-symmetric models. Therefore, two models with Hamiltonians $\mathbb{H}$ and $\mathbb{H}^\lambda$—only differing in a choice of 3-cocycle $\lambda$—are associated with the $2\mathsf{Vec}_G$-module categories $2\mathsf{Vec}$ and $2\mathsf{Vec}^\lambda$, respectively. This can be made very explicit employing the formalism of ref. [DT23], from which also follows that a duality operator transmuting $\mathbb{H}$ into $\mathbb{H}^\lambda$ corresponds to a choice of simple object in $2\mathsf{Fun}_{2\mathsf{Vec}_G}(2\mathsf{Vec}, 2\mathsf{Vec}^\lambda)$, which can be shown to be equivalent to $2\mathsf{Rep}^\lambda(G)$, as expected.

### 3.8 Self-duality and lattice higher gauge theories

As in (1+1)d, a particularly attractive feature of the dualities considered in this section is that they preserve the symmetry structure, namely $2\mathsf{Rep}(G)$ for some finite abelian group $G$. As a matter of fact, it seems that all non-trivial dualities of bosonic two-dimensional quantum lattice models preserving symmetry structures are of this flavour, i.e., dualities between models only differing in a choice of fiber

2-functor over some input fusion 2-category [DY23]. This property suggests that it may be possible to define a model that is self-dual with respect to such a duality. Given such a self-dual model, one could then consider promoting the self-duality to a genuine internal symmetry, which mathematically amounts to considering an extension of the symmetry fusion 2-category.

Let us illustrate this principle with the first duality studied in sec. 3.6. There, we constructed two manifestly $2\mathsf{Rep}(\mathbb{Z}_2)$-symmetric Hamiltonians $\mathbb{H}$ and $\mathbb{H}^\lambda$, which are associated with fiber 2-functors $2\mathsf{Vec}_{\mathbb{Z}_2} \to 2\mathsf{Vec}$ and $2\mathsf{Vec}_{\mathbb{Z}_2} \to 2\mathsf{Vec}^\lambda$, respectively. Clearly, the Hamiltonian $\mathbb{H} + \mathbb{H}^\lambda$ is left invariant under the action of the duality operator (3.63), making $\mathbb{H} + \mathbb{H}^\lambda$ self-dual. We claim that promoting this self-duality to a genuine internal symmetry results in the fusion 2-category $2\mathsf{Rep}(\mathbb{Z}_2^{[0]}\bullet_\mu\mathbb{Z}_2^{[1]})$ of 2-representations of a *2-group* $\mathbb{Z}_2^{[0]}\bullet_\mu\mathbb{Z}_2^{[1]}$ with homotopy group $\mathbb{Z}_2$ both in degree one and two. Generally, a 2-group is defined as a monoidal category such that every object admits a weak inverse and every morphism is invertible. In particular, given two finite abelian groups $G$ and $H$, and a normalised representative 3-cocycle $\mu$ of a cohomology class in $H^3(G, H)$, the 2-group $G^{[0]}\bullet_\mu H^{[1]}$ is the monoidal category such that the objects form the group $G$, automorphisms of the monoidal unit forms the group $H$, while the monoidal associator is provided by $\mu$.[15] The fusion 2-category $2\mathsf{Rep}(G^{[0]}\bullet_\mu H^{[1]})$ is then defined as the fusion 2-category of pseudofunctors from the so-called delooping of the 2-group to $2\mathsf{Vec}$.[16] The underlying 2-category was studied in detail in ref. [Elg07]:

$$2\mathsf{Rep}(G^{[0]}\bullet_\mu H^{[1]}) \simeq \boxplus_{\chi\in\widehat{H}} 2\mathsf{Rep}^{\chi\circ\mu}(G). \tag{3.68}$$

We are now ready to show that the self-dual Hamiltonian $\mathbb{H} + \mathbb{H}^\lambda$ has a $2\mathsf{Rep}(\mathbb{Z}_2^{[0]}\bullet_\mu\mathbb{Z}_2^{[1]})$ symmetry structure, where $\mu(g_1, g_2, g_3) = 1 \in \mathbb{Z}_2$ if $g_1 = g_2 = g_3 = 1$, and 0 otherwise. Our strategy is to demonstrate that $\mathbb{H} + \mathbb{H}^\lambda$ has a lattice *higher gauge theory* interpretation [BH10, KT13, BD19, DT18, DT19, BD19, ZLW19] and results from gauging a two-dimensional quantum lattice model with a $2\mathsf{Vec}_{\mathbb{Z}_2^{[0]}\bullet_\mu\mathbb{Z}_2^{[1]}}$ symmetry [DT23].

We begin by constructing a family of two-dimensional quantum lattice models with a lattice higher gauge theory interpretation that possess a $2\mathsf{Rep}(G^{[0]}\bullet_\mu H^{[1]})$ symmetry, before specialising to $2\mathsf{Rep}(\mathbb{Z}_2^{[0]}\bullet_\mu\mathbb{Z}_2^{[1]})$. The kinematical Hilbert space is spanned by vectors $|\mathfrak{g}, \mathfrak{h}\rangle$, where $\mathfrak{g} \in Z^1(\mathbb{T}_\triangle^2, G)$ and $\mathfrak{h} \in C^2(\mathbb{T}_\triangle^2, H)$. Let $\mu$ be a normalised representative 3-cocycle of a cohomology class $[\mu] \in H^3(G, H)$. Given a vertex $\mathsf{v} \in \mathsf{V}(\mathbb{T}_\triangle^2)$, we denote by $\mathfrak{x}_\mathsf{v} \in C^0(\mathbb{T}_\triangle^2, G)$ a function $\mathsf{V}(\mathbb{T}_\triangle^2) \to G$ such that $\mathfrak{x}_\mathsf{v}(\mathsf{v}_1) = \mathbb{1}$ whenever $\mathsf{v}_1 \neq \mathsf{v}$, and $\mathfrak{x}_\mathsf{v}(\mathsf{v}) = x$, for some $x \in G$. Moreover, we denote by $\mathfrak{z}_\mathsf{v} \in C^1(\mathbb{T}_\triangle^2, H)$ a function $\mathsf{E}(\mathbb{T}_\triangle^2) \to H$ such that $\mathfrak{z}_\mathsf{v}(\mathsf{v}_1\mathsf{v}_2) = \mathbb{1}$ whenever $\mathsf{v} \notin \partial(\mathsf{v}_1\mathsf{v}_2)$. We consider the following local operators acting on the kinematical Hilbert space:

$$\mathbb{h}_{\mathsf{v},n} := \sum_{\substack{\mathfrak{g}\in Z^1(\mathbb{T}_\triangle^2, G) \\ \mathfrak{h}\in C^0(\mathbb{T}_\triangle^2, H)}} \sum_{\mathfrak{x}_\mathsf{v}, \mathfrak{z}_\mathsf{v}} h_n(\mathfrak{g}, \mathfrak{h}, \mathfrak{x}_\mathsf{v}, \mathfrak{z}_\mathsf{v}) |\mathfrak{g}\, \mathrm{d}^{(0)}\mathfrak{x}_\mathsf{v}, \mathfrak{h}\, \mathrm{d}^{(1)}\mathfrak{z}_\mathsf{v}\, \mu(\mathfrak{g}, \mathfrak{x}_\mathsf{v})^{-1}\rangle\langle\mathfrak{g}, \mathfrak{h}|\,, \tag{3.69}$$

where $n$ is some label, $h_n(\mathfrak{g}, \mathfrak{h}, \mathfrak{x}_\mathsf{v}, \mathfrak{z}_\mathsf{v})$ are arbitrary complex coefficients, $\mathfrak{h}\, \mathrm{d}^{(1)}\mathfrak{z}_\mathsf{v}\, \mu(\mathfrak{g}, \mathfrak{x}_\mathsf{v})^{-1}$ is by defini-

---

[15]More generally, the group $H$ can be endowed with the structure of a $G$-module, but we choose the module structure to be trivial.

[16]Similarly, the fusion 2-category of 2-representations of a finite group $G$ can be defined as the fusion 2-category of pseudofunctors from the delooping of the group to $2\mathsf{Vec}$.

tion the function $\mathsf{P}[\mathbb{T}^2_\triangle] \to G$ such that

$$(\mathfrak{h}\,\mathrm{d}^{(1)}\mathfrak{z}_\mathsf{v}\,\mu(\mathfrak{g},\mathfrak{x}_\mathsf{v})^{-1})(\mathsf{p}) = \begin{cases} \mathfrak{h}(\mathsf{p})\,\mathfrak{z}_\mathsf{v}(\mathsf{v}_2\mathsf{v})\,\mathfrak{z}_\mathsf{v}(\mathsf{v}_1\mathsf{v})^{-1}\mu\big(\mathfrak{g}(\mathsf{v}_1\mathsf{v}_2),\mathfrak{g}(\mathsf{v}_2\mathsf{v})x^{-1},x\big)^{-1} & \text{if } \mathsf{p} \equiv (\mathsf{v}_1\mathsf{v}_2\mathsf{v}) \\ \mathfrak{h}(\mathsf{p})\,\mathfrak{z}_\mathsf{v}(\mathsf{v}\,\mathsf{v}_2)\,\mathfrak{z}_\mathsf{v}(\mathsf{v}_1\mathsf{v})\,\mu\big(\mathfrak{g}(\mathsf{v}_1\mathsf{v})x^{-1},x,\mathfrak{g}(\mathsf{v}\,\mathsf{v}_2)\big) & \text{if } \mathsf{p} \equiv (\mathsf{v}_1\mathsf{v}\,\mathsf{v}_2) \\ \mathfrak{h}(\mathsf{p})\,\mathfrak{z}_\mathsf{v}(\mathsf{v}\,\mathsf{v}_1)\,\mathfrak{z}_\mathsf{v}(\mathsf{v}\,\mathsf{v}_2)^{-1}\mu\big(x,\mathfrak{g}(\mathsf{v}\,\mathsf{v}_1),\mathfrak{g}(\mathsf{v}_1\mathsf{v}_2)\big)^{-1} & \text{if } \mathsf{p} \equiv (\mathsf{v}\,\mathsf{v}_1\mathsf{v}_2) \\ \mathfrak{h}(\mathsf{p}) & \text{otherwise} \end{cases} , \quad (3.70)$$

where we employed the shorthand $\mathfrak{x}_\mathsf{v}(\mathsf{v}) \equiv x \in G$, and defined the 2-cochain $\mu(\mathfrak{g},\mathfrak{x}_\mathsf{v})$ analogously to the phase factor (3.5).

We claim that any Hamiltonian built from local operators (3.69) possesses a $2\mathsf{Rep}(G^{[0]}\bullet_\mu H^{[1]})$ symmetry. Let us focus on the 0-form symmetry operators. It follows from eq. (3.68) that simple objects $2\mathsf{Rep}(G^{[0]}\bullet_\mu H^{[1]})$ are labelled by pairs $\big(\chi \in \widehat{H}, \mathcal{M}(A,\phi) \in 2\mathsf{Rep}^{\chi\circ\mu}(G)\big)$. In particular, whenever $\chi$ is the trivial representation of $H$, the topological surface operator is labelled by a 2-representation of $G$, acts as in eq. (3.26), and commutes with local operators $\mathbb{h}_{\mathsf{v},n}$ for the same reasons as before. More generally, let us denote by $\mathbb{U}^{A_\phi,\chi}(\mathbb{T}^2_\triangle)$ the topological surface operator associated with the pair $(\chi, \mathcal{M}(A,\phi))$:

$$\mathbb{U}^{A_\phi,\chi}(\mathbb{T}^2_\triangle) := \sum_{\substack{\mathfrak{g}\in Z^1(\mathbb{T}^2_\triangle,G) \\ \mathfrak{m}\in C^0(\mathbb{T}^2_\triangle,G/A)}} \left( \prod_{(\mathsf{v}_1\mathsf{v}_2)} \delta_{\mathfrak{g}(\mathsf{v}_1\mathsf{v}_2)\triangleright\mathfrak{m}(\mathsf{v}_2),\mathfrak{m}(\mathsf{v}_1)} \right.$$
$$\left. \cdot \prod_{(\mathsf{v}_1\mathsf{v}_2\mathsf{v}_3)} \Big[\chi(\mathfrak{h})\,\alpha_\phi^\triangleright\big(\mathfrak{g}(\mathsf{v}_1\mathsf{v}_2),\mathfrak{g}(\mathsf{v}_2\mathsf{v}_3)\big)\big(\mathfrak{m}(\mathsf{v}_3)\big)\Big]^{\epsilon(\mathsf{v}_1\mathsf{v}_2\mathsf{v}_3)} \right) |\mathfrak{g},\mathfrak{h}\rangle\!\langle\mathfrak{g},\mathfrak{h}| . \tag{3.71}$$

Letting $\mathfrak{x}_\mathsf{v} \in C^0(\mathbb{T}^2_\triangle,G)$ and $\mathfrak{z}_\mathsf{v} \in C^1(\mathbb{T}^2_\triangle,H)$ defined as above, it follows from eq. (3.47) as well as $\prod_{(\mathsf{v}_1\mathsf{v}_2\mathsf{v}_3)} \chi(\mathrm{d}^{(1)}\mathfrak{z}_\mathsf{v}) = 1$ that

$$\prod_{(\mathsf{v}_1\mathsf{v}_2\mathsf{v}_3)} \Big[\chi\big(\mathfrak{h}\,\mathrm{d}^{(1)}\mathfrak{z}_\mathsf{v}\,\mu(\mathfrak{g},\mathfrak{x}_\mathsf{v})^{-1}\big)\,\alpha_\phi^\triangleright\big((\mathfrak{g}\,\mathrm{d}^{(0)}\mathfrak{x}_\mathsf{v})(\mathsf{v}_1\mathsf{v}_2),(\mathfrak{g}\,\mathrm{d}^{(0)}\mathfrak{x}_\mathsf{v})(\mathsf{v}_2\mathsf{v}_3)\big)\big((\mathfrak{x}_\mathsf{v}\triangleright\mathfrak{m})(\mathsf{v}_3)\big)\Big]^{\epsilon(\mathsf{v}_1\mathsf{v}_2\mathsf{v}_3)}$$
$$= \prod_{(\mathsf{v}_1\mathsf{v}_2\mathsf{v}_3)} \Big[\chi(\mathfrak{h})\,\alpha_\phi^\triangleright\big(\mathfrak{g}(\mathsf{v}_1\mathsf{v}_2),\mathfrak{g}(\mathsf{v}_2\mathsf{v}_3)\big)\big(\mathfrak{m}(\mathsf{v}_3)\big)\Big]^{\epsilon(\mathsf{v}_1\mathsf{v}_2\mathsf{v}_3)} , \tag{3.72}$$

for every $\mathfrak{g} \in Z^1(\mathbb{T}^2_\triangle,G)$ and $\mathfrak{h} \in C^2(\mathbb{T}^2_\triangle,H)$, thereby confirming that $\mathbb{U}^{A_\phi,\chi}(\mathbb{T}^2_\triangle)$ commutes with local operators $\mathbb{h}_{\mathsf{v},n}$.

Let us now specialise to the 2-group $\mathbb{Z}_2^{[0]}\bullet_\mu\mathbb{Z}_2^{[1]}$. By definition, the Hamiltonian $\mathbb{H}+\mathbb{H}^\lambda$ constructed from eq. (3.58) and eq. (3.62) is made of local operators that acts as $-\prod_{\mathsf{e}\supset\mathsf{v}}X_\mathsf{e}$, for every $\mathsf{v} \in \mathsf{V}(\mathbb{T}^2_\triangle)$, while projecting out states $|\mathfrak{g}\rangle$ satisfying $\lambda(\mathfrak{g},\mathfrak{x}_\mathsf{v}) = -1$. We argue that there is a choice of complex coefficients $h_1(\mathfrak{g},\mathfrak{h},\mathfrak{x}_\mathsf{v},\mathfrak{z}_\mathsf{v})$ such that local operators of the form (3.69) effectively act in the same way. Similarly to the example of sec. 2.6, the crux is to restrict the kinematical Hilbert space to states $|\mathfrak{g},\mathfrak{h}\rangle$, for which the 2-cochain $\mathfrak{h} \in C^2(\mathbb{T}^2_\triangle,\mathbb{Z}_2)$ assigns the identity group element to every plaquette. This can be enforced via a choice of coefficients $h_1(\mathfrak{g},\mathfrak{h},\mathfrak{x}_\mathsf{v},\mathfrak{z}_\mathsf{v})$. One can further choose these coefficients such that the only non-vanishing ones are equal to $-1$, for every $\mathfrak{g} \in Z^1(\mathbb{T}^2_\triangle,\mathbb{Z}_2)$, $\mathfrak{z}_\mathsf{v} \in C^1(\mathbb{T}^2_\triangle,\mathbb{Z}_2)$, and $\mathfrak{x}_\mathsf{v}(\mathsf{v}) = 1$. Proceeding as such, $(\mathfrak{h}\,\mathrm{d}^{(1)}\mathfrak{z}_\mathsf{v}\,\mu(\mathfrak{g},\mathfrak{x}_\mathsf{v})^{-1})(\mathsf{p}) = 0$ enforces that $(\mathrm{d}^{(1)}\mathfrak{z}_\mathsf{v}\,\mu(\mathfrak{g},\mathfrak{x}_\mathsf{v})^{-1})(\mathsf{p}) = 0$, for every $\mathsf{p} \in \mathsf{P}(\mathbb{T}^2_\triangle)$. For any $\mathfrak{z}_\mathsf{v} \in C^1(\mathbb{T}^2_\triangle,\mathbb{Z}_2)$, the previous condition enforces that $\mu(\mathfrak{g},\mathfrak{x}_\mathsf{v})(\mathsf{p})$ is distinct from the identity element in $\mathbb{Z}_2$ for an even number of plaquettes, precisely constraining $\mathfrak{g}$ to be such that $\lambda(\mathfrak{g},\mathfrak{x}_\mathsf{v}) = 1$, as desired. Bringing everything together, this confirms that promoting the self-duality of $\mathbb{H}+\mathbb{H}^\lambda$ to an internal symmetry does result in a $2\mathsf{Rep}(\mathbb{Z}_2^{[0]}\bullet_\mu\mathbb{Z}_2^{[1]})$ symmetry.

Naturally, one can construct much more general lattice higher gauge theory with a $2\mathsf{Rep}(\mathbb{Z}_2^{[0]}\bullet_\mu\mathbb{Z}_2^{[1]})$ symmetry than the one discussed above. Given such a theory, one can then ask for a duality relation of the same type as those considered so far, i.e. combining twisted and untwisted gauging operations. This is possible since the fusion 2-category $2\mathsf{Vec}_{\mathbb{Z}_2^{[0]}\bullet_\mu\mathbb{Z}_2^{[1]}}$ admits two inequivalent fiber 2-functors, which differ in a choice of 2-group 3-cocycle. Indeed, representatives of cohomology classes in $H^3(\mathbb{Z}_2^{[0]}\bullet_\mu\mathbb{Z}_2^{[1]},\mathrm{U}(1))$ are labelled by pairs $(\epsilon,\gamma)$ consisting of a 3-cochain $\gamma\in C^3(\mathbb{Z}_2,\mathrm{U}(1))$ and a homomorphism $\epsilon:\mathbb{Z}_2\to\widehat{\mathbb{Z}_2}$ such that $\mathrm{d}\gamma=\langle\epsilon,\smile\mu\rangle$ [KT13]. The unique non-trivial solution is provided by choosing $\epsilon$ to be the non-trivial character of $\mathbb{Z}_2$ and $\gamma(g_1,g_2,g_3)=i^{g_1g_2g_3}$. Given a model that would happen to be self-dual under the resulting duality, lifting the self-duality to an internal symmetry would result in one of the Tambara–Yamagami fusion 2-categories considered in ref. [DY23].

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
