# Peer review of "Twisted gauging and topological sectors in (2+1)d abelian lattice gauge theories"

_SciPost Physics_

## Round 1 · Referee Report · Anonymous (Referee 2) · 2025-5-22

Report

In this paper, the authors develop a systematic formulation of duality on quantum lattice models with an invertible symmetry. In 1+1 dimension, the original lattice model is assumed to have a conventional Abelian invertible (0-form) symmetry. Then the duality is defined by an untwisted gauging of the symmetry followed by a twisted gauging of the dual symmetry, which also has a tensor-network representation. For the $\mathbb{Z}_p \times \mathbb{Z}_p$ symmetry, the duality thus defined reproduces the well known Kennedy-Tasaki transformation. The authors then extend the formulation to lattice gauge theories in 2+1 dimensions, to formulate the duality as a untwisted gauging of the 1-form symmetry (of the original lattice gauge model) followed by a twisted gauging of the dual 0-form symmetry. As in 1+1 dimensions, there is a nontrivial interplay between the symmetry-twisted boundary conditions and charge sectors under the duality. This paper gives a general perspective for the Kennedy-Tasaki duality in 1+1 dimension and further provides interesting novel results on duality on lattice gauge theories in 2+1 dimensions. I recommend publication of this paper in SciPost Physics. However I would like to ask the authors to consider the following points.

1) The notation $\mathfrak{x}_v$ is confusing as it actually depends on $x$ but it is not easy to relate $x$ with $\mathfrak{x}$. Maybe you can introduce another symbol and put $x$ in superscript (if this does not cause other problems). 2) (this is optional) Do you have any insight/comment on systems on 1d lattice other than simple chains (such as 2-leg ladders)? Essentially they are 1+1 dimensional but in some sense they are between 1+1 and 2+1 dimensions ($n$-leg ladder in $n \to \infty$ limit is the 2d square lattice).

Recommendation

Ask for minor revision

---

## Round 1 · Referee Report · Anonymous (Referee 3) · 2025-5-24

Report

The authors investigate twisted gauging of both ordinary and generalized symmetries in 2+1D lattice models. They focus on models connected through sequences of gauging transformations and the associated duality or non-invertible operators that relate them. The paper presents a tensor network formulation of these non-invertible operators and also explores models that are self-dual under such gaugings, leading to enhanced non-invertible symmetries.

I recommend the draft for publication, pending the following minor revision:

While the paper is technically precise, it is difficult to follow, particularly for readers not already familiar with the authors’ notation. I suggest including a summary or roadmap for Sections 2 and 3. For example, it would be helpful to clearly outline the starting point (e.g., the Hamiltonian and its global symmetry), the gauging operations or duality transformations applied, and the resulting theory (with its new Hamiltonian and symmetry). This overview could be presented in a concise table or descriptive paragraph, either in the introduction or at the end of each relevant subsection.

Recommendation

Publish (meets expectations and criteria for this Journal)

---

## Editorial Decision

resubmitted